# Exact Community Recovery under Side Information: Optimality of Spectral Algorithms

**Julia Gaudio**
Northwestern University
julia.gaudio@northwestern.edu

**Nirmit Joshi**
Toyota Technological Institute at Chicago
nirmit@ttic.edu

## Abstract

We study the problem of exact community recovery in general, two-community block models, in the presence of node-attributed *side information*. We allow for a very general side information channel for node attributes, and for pairwise (edge) observations, consider both Bernoulli and Gaussian matrix models, capturing the Stochastic Block Model, Submatrix Localization, and $\mathbb{Z}_2$-Synchronization as special cases. A recent work of Dreveton et al. (2024) characterized the information-theoretic limit of a very general exact recovery problem with side information. In this paper, we show algorithmic achievability in the above important cases by designing a simple but optimal spectral algorithm that incorporates side information (when present) along with the eigenvectors of the pairwise observation matrix. Using the powerful tool of entrywise eigenvector analysis (Abbe et al., 2020), we show that our spectral algorithm can mimic the so called *genie-aided estimators*, where the $i^{\text{th}}$ genie-aided estimator optimally computes the estimate of the $i^{\text{th}}$ label, when all remaining labels are revealed by a genie. This perspective provides a unified understanding of the optimality of spectral algorithms for various exact recovery problems in a recent line of work.

## 1 Introduction

In this paper, we consider inference problems of the following form: there is an unknown partition of the set $[n] := \{1, 2, \ldots, n\}$ into two *communities*, denoted $(C_+, C_-)$ with $(\rho, 1 - \rho)$ fraction of vertices respectively for $\rho \in (0, 1)$. The community assignments are encoded by a vector $\sigma^* \in \{\pm 1\}^n$. We observe a symmetric matrix $A \in \mathbb{R}^{n \times n}$, which is specified by three distributions: $\mathcal{P}_+$, $\mathcal{P}_-$, and $\mathcal{Q}$. The entries of $A$ are independent (up to symmetry), such that $A_{ij} \sim \mathcal{P}_+$ if $i, j \in C_+$, $A_{ij} \sim \mathcal{P}_-$ if $i, j \in C_-$, and $A_{ij} \sim \mathcal{Q}$ otherwise. One famous example is the Stochastic Block Model (SBM), where $\mathcal{P}_+ \equiv \mathrm{Bern}(p_1)$, $\mathcal{P}_- \equiv \mathrm{Bern}(p_2)$, and $\mathcal{Q} \equiv \mathrm{Bern}(q)$. Other prominent examples include $\mathbb{Z}_2$-synchronization and submatrix localization, in which $\mathcal{P}_+, \mathcal{P}_-, \mathcal{Q}$ are Gaussian distributions. Given the observation $A$, the goal is to *exactly* recover the unknown $\sigma^*$.

After a long line of research (Decelle et al., 2011; Mossel et al., 2015; Abbe et al., 2016; Hajek et al., 2016; Abbe, 2017; Bandeira et al., 2017; Javanmard et al., 2016; Cai et al., 2017), a fairly complete picture of the fundamental limits of exact recovery and algorithmic achievability is known. In recent years, with practical considerations, exact recovery has also been studied in different variants such as node-attributed side information (Saad & Nosratinia, 2018; 2020; Deshpande et al., 2018; Abbe et al., 2022; Dreveton et al., 2024), partially censored edges (Abbe et al., 2014; Hajek et al., 2015; Moghaddam et al., 2022; Dhara et al., 2022a; 2023), multiple correlated networks (Racz & Sridhar, 2021; Gaudio et al., 2022), spatially embedded networks (Abbe et al., 2021; Gaudio et al., 2024b;a), etc. See Appendix A.3 for different variants considered more broadly in community detection literature beyond exact recovery.

In this paper, similar to Dreveton et al. (2024), we consider the setting of node-attributed side information, where we are given a side information vector $y \in \mathcal{Y}^n$, in addition to the pairwise observation matrix $A \in \mathbb{R}^{n \times n}$. The side information is drawn according to a pair of distributions $(\mathcal{S}_+, \mathcal{S}_-)$, where for each $i \in [n]$, we obtain

$$y_i \sim \mathcal{S}_+ \text{ if } i \in C_+, \quad \text{or} \quad y_i \sim \mathcal{S}_- \text{ if } i \in C_-,$$

where the observations are independent, conditioned on the communities. The setup captures several interesting special cases:

1. **Gaussian Features** (GF)**:** For each node, we have a vector of $d$ real-valued attributes. This is modeled as $\mathcal{Y} = \mathbb{R}^d$ and both $\mathcal{S}_+$ and $\mathcal{S}_-$ are parameterized multivariate Gaussians in $d$ dimensions.

2. **Binary Erasure Channel** (BEC)**:** The true community assignment of some subset of vertices are known (partially *observed* labels), modeled as $\mathcal{Y} = \{-1, 0, +1\}$ and $y$ is formed by passing $\sigma^*$ through a binary erasure channel.

3. **Binary Symmetric Channel** (BSC)**:** We have a "guess" on community assignment of each vertex (partially *correct* labels) and interested in recovering the true assignment. This is modeled as passing $\sigma^*$ through a binary symmetric channel with $\mathcal{Y} \in \{-1, +1\}$.

Exact recovery under BEC and BSC side information channels was first studied by Saad & Nosratinia (2018; 2020) for two special cases of Bernoulli pairwise observations; symmetric SBM ($\mathcal{P}_+ \equiv \mathcal{P}_-, \rho = 1/2$) and Planted Dense Subgraph (PDS) ($\mathcal{P}_- \equiv \mathcal{Q}$). The work of Dreveton et al. (2024) recently studied a very general recovery problem, allowing generic distributions ($\mathcal{P}_+, \mathcal{P}_-, \mathcal{Q}$) and side information laws ($\mathcal{S}_+, \mathcal{S}_-$), satisfying certain technical assumptions. They derived sharp information theoretic thresholds for exact recovery under side information in a unified way, and showed that the optimal Maximum A Posterior (MAP) estimator succeeds down to this threshold. However, naïve MAP estimation requires a brute-force computation, and thus it is not efficiently (poly-time) computable in the worst case. Therefore, it remains important to design efficient algorithms. We note that Dreveton et al. also proposed an efficient, iterative likelihood maximization algorithm when node and edge observations are from an exponential family of distributions, though without theoretical guarantees on the performance. In this work, our primary goal is:

*Objective 1: To design efficient algorithms that are provably optimal i.e. succeed down to the information theoretic threshold.*

We note that the work of Saad & Nosratinia (2018; 2020) designed provably optimal algorithms only for the two special cases they studied (symmetric SBM and PDS), and only under BEC and BSC channels. They use a *two-stage* strategy that first determines *almost exactly correct* labels, followed by a refinement step achieving exact recovery (see Appendix A.3). In this work, our focus will be on designing a *single-stage* spectral algorithm without any refinement step that directly achieves exact recovery and in much greater generality. In particular, we consider any side information channel with distributions ($\mathcal{S}_+, \mathcal{S}_-$) but restrict to the following pairwise distribution laws[1]:

1. Rank One Spike (ROS): The distributions $\mathcal{P}_+$, $\mathcal{P}_-$, and $\mathcal{Q}$ are Gaussian distributions, capturing $\mathbb{Z}_2$-synchronization and submatrix localization as special cases.

2. SBM: The general two community stochastic block model where $\mathcal{P}_+$, $\mathcal{P}_-$, and $\mathcal{Q}$ are any Bernoulli distributions.

**Background on spectral algorithms for exact recovery.** Spectral algorithms were popularized by classical works such as McSherry's algorithm for community detection (McSherry, 2001) and the planted clique recovery algorithm of Alon, Krivilevich, and Sudakov (Alon et al., 1998). In recent years, attention has shifted to spectral algorithms without the need of a combinatorial cleanup phase. This line of work was initiated by the influential work of Abbe, Fan, Wang, and Zhong (Abbe et al., 2020), who showed that in the symmetric, balanced SBM, simply thresholding the second leading eigenvector $u_2$ of the adjacency matrix at $0$ gives the correct community partition with high probability. In order to analyze the vector $u_2$, Abbe et al. developed the technique of *entrywise eigenvector analysis*, which characterizes the entrywise behavior of eigenvectors of matrices whose expectations are low-rank.

Following Abbe et al. (2020), a series of papers used the entrywise eigenvector technique to give strong guarantees for spectral algorithms. For example, Deng et al. (2021) showed that, in the symmetric SBM, using the Laplacian instead of the adjacency matrix also yields an optimal algorithm. Another line of work Dhara et al. (2022a; 2023) considered the censored variant of the problem, where the status of some edges is unknown. Even for the censored variant, they show that

---

[1]While our side information channel laws are general, we note that the information theoretic limits of Dreveton et al. (2024) are in even more generality. Namely, they consider (i) more general distribution families ($\mathcal{P}_+, \mathcal{P}_-, \mathcal{Q}$), and (ii) the case of more than two communities. In Appendix A.2, we shall describe how our spectral algorithms can be further generalized to (i) any ($\mathcal{P}_+, \mathcal{P}_-, \mathcal{Q}$) coming from exponential families (with additional requirements), and (ii) more than two communities.

spectral algorithms are optimal, where the encoding of the unknown edges is chosen carefully to achieve optimaliy. Perhaps surprisingly, Dhara et al. (2023) showed that for this censored variant of the problem, to handle cases beyond the symmetric SBM ($\mathcal{P}_+ \equiv \mathcal{P}_-, \rho = 1/2$) and the PDS ($\mathcal{P}_- \equiv \mathcal{Q}$), any clustering algorithm based on a single adjacency matrix does not succeed down to the information-theoretic threshold. Instead, the authors devised a spectral algorithm which forms two matrix representations of the same network and takes a carefully chosen linear combination of their eigenvectors to achieve optimaliy. All these results at their core rely on the entrywise behavior of eigenvectors (Abbe et al., 2020).

This raises several closely related questions: what governs the optimality of these seemingly different problem specific choices? Is there some principle behind designing new algorithms in related settings? For example, for the standard uncensored variant as we study here, the spectral algorithm for general two community SBM is unknown even when there is no side information. Do we need two matrices like its censored counterpart (Dhara et al., 2023)? To answer these questions, another auxiliary goal of this work:

*Objective 2: To develop a unified perspective on the optimality of these spectral algorithms.*

## 1.1 OUR CONTRIBUTION

**Spectral Algorithms.** We propose a simple spectral strategy of computing the top eigenvectors (top one for ROS and top two for SBM) of the observation matrix $A$ and take an appropriate linear combination. When side information $y$ is also available, we incorporate it by shifting the eigenvector combination by the log-likelihood ratio vector of side information:

$$\log\left(\frac{\mathcal{S}_+}{\mathcal{S}_-}(y)\right) \in (\mathbb{R} \cup \{\pm\infty\})^n \text{ such that } \left[\log\left(\frac{\mathcal{S}_+}{\mathcal{S}_-}(y)\right)\right]_i := \log\left(\frac{\mathcal{S}_+(y_i)}{\mathcal{S}_-(y_i)}\right), \qquad (1)$$

followed by a prescribed thresholding.

The main technical novelty lies in establishing a rigorous connection between the spectral estimator and the so-called *genie-aided estimators*, where $i^{\text{th}}$ genie-aided estimator optimally computes the $i^{\text{th}}$ label, where the rest of them, denoted by $\sigma^*_{-i}$, are revealed by a genie. Using the entrywise eigenvector technique (Abbe et al., 2020), we show that taking an appropriately weighted sum of the leading eigenvectors of $A$ (along with side information use prescribed above when they are available) produces a vector whose $i^{\text{th}}$ entry is well-approximated by the statistic computed by these genie-aided estimators in each of the settings. Thus, the spectral algorithm without any clean-up step is able to mimic the genie, and therefore achieves exact recovery down to the information-theoretic limits. Moreover, our algorithm is highly efficient with *nearly linear runtime* for the respective setting (see Remark 5.3).

**Models without Side Information.** We note that even in the case of no side information, determining when a single stage spectral algorithm without any clean up stage can succeed is of interest to the learning theory community. Our results fill the complete picture in important remaining cases such as the general SBM (beyond the symmetric case with $\mathcal{P}_+ \equiv \mathcal{P}_-, \rho = 1/2$ (Abbe et al., 2020) and the Planted Dense Subgraph (PDS) with $\mathcal{P}_- \equiv \mathcal{Q}$ (Dhara et al., 2022a)). Most notably, unlike its censored counterpart (Dhara et al., 2023), one does not need two matrix representations to achieve optimality. In fact, the strategies of Abbe et al. (2020); Deng et al. (2021); Dhara et al. (2022a;b; 2023) are essentially mimicking the genie in their respective settings. See the discussion in Appendix A.2 for more details.

**Unsupervised vs Semi-supervised Learning.** The presence of side information turns community recovery from an unsupervised learning problem into a semi-supervised learning problem. Therefore, another related perspective is to investigate the effectiveness semi-supervised learning approaches over unsupervised ones e.g. (Jiang & Ke, 2023; Wu et al., 2022; Ni et al., 2024). We note that, from this perspective, investigating other relaxed goals such as *weak* or *almost exact* recovery could provide a more satisfying picture, as the exact recovery is a very demanding criterion. The primary message of our work is that, for the exact recovery goal, there are simple extensions of the spectral algorithm that optimally combine the signal from $A$ and the side information $y$, achieving the new sharp information-theoretic limit provably using an efficient algorithm.

**Organization.** Section 2 contains our models and other preliminary setup. Our main results are stated in Section 3. The genie-aided estimators are introduced in Section 4, which we show how to

mimic using a spectral strategy in Section 5. Future directions are proposed in Section 6. The proofs are postponed to the appendices.

## 2 PRELIMINARIES

### 2.1 MODELS

We first introduce the General Two Community Block Model (GBM), which captures the two special cases that we consider.

**Definition 2.1** (General Two Community Block Model (GBM)). *For any $\rho \in (0,1)$ and distributions $\mathcal{P}_+, \mathcal{P}_-$, and $\mathcal{Q}$, we say that $(A, \sigma^*) \sim \mathsf{GBM}_n(\rho, \mathcal{P}_+, \mathcal{P}_-, \mathcal{Q})$, where $A \in \mathbb{R}^{n \times n}$ and $\sigma^* \in \{\pm 1\}^n$ are sampled as follows. Each coordinate of $\sigma^*$ is sampled i.i.d. such that $\mathbb{P}(\sigma_i^* = +1) = \rho$ and $\mathbb{P}(\sigma_i^* = -1) = 1 - \rho$. Moreover, we will use the notation $C_+ := \{i : \sigma_i^* = +1\}$ and $C_- := \{i : \sigma_i^* = +1\}$. Conditioned on $\sigma^*$, we sample $A$, a zero diagonal symmetric matrix with independent entries, such that for $1 \le i < j \le n$, we have*

$$A_{ij} \sim \mathcal{P}_+, \text{ if } i, j \in C_+; \quad A_{ij} \sim \mathcal{P}_-, \text{ if } i, j \in C_-; \quad A_{ij} \sim \mathcal{Q}, \text{ otherwise.}$$

In the above definition, we restrict to distributions which are either (i) a continuous distribution or (ii) a finite, discrete distribution. Thus, $\mathcal{P}_+(\cdot), \mathcal{P}_-(\cdot), \mathcal{Q}(\cdot)$ also denote the corresponding probability density function or probability mass function, respectively. We will consider special cases where the distributions $(\mathcal{P}_+, \mathcal{P}_-, \mathcal{Q})$ are either all Gaussian or Bernoulli distributions. The specialized definitions are given below.

**Definition 2.2** (Rank One Spike (ROS)). *Fix any $\rho \in (0,1)$ and $a, b \in \mathbb{R}$ such that $\max\{|a|, |b|\} > 0$. We say that $(A, \sigma^*) \sim \mathsf{ROS}_n(\rho, a, b)$ if they are sampled as follows. First sample $\sigma^*$ as mentioned for $\mathsf{GBM}$. Conditioned on $\sigma^*$ consider the vector $v^* \in \{a, b\}^n$ such that for $i \in [n]$*

$$v_i^* = a \cdot \mathbf{1}[\sigma_i^* = +1] + b \cdot \mathbf{1}[\sigma_i^* = -1]. \tag{2}$$

*Finally, conditioned on $\sigma^*$, we get independent noisy measurements for every $i < j$ of the following form.*

$$A_{ij} = v_i^* v_j^* \sqrt{\frac{\log n}{n}} + W_{ij}, \quad \text{where } W_{ij} \overset{\text{i.i.d.}}{\sim} \mathcal{N}(0,1).$$

Note that the model $\mathsf{ROS}_n(\rho, a, b)$ is a special case of $\mathsf{GBM}_n(\rho, \mathcal{P}_+, \mathcal{P}_-, \mathcal{Q})$, by taking

$$\mathcal{P}_+ \equiv \mathcal{N}\left(a^2 \sqrt{\frac{\log n}{n}}, 1\right), \mathcal{P}_- \equiv \mathcal{N}\left(b^2 \sqrt{\frac{\log n}{n}}, 1\right), \text{ and } \mathcal{Q} \equiv \mathcal{N}\left(ab \sqrt{\frac{\log n}{n}}, 1\right).$$

Taking $b = 0$ yields a version of the Gaussian submatrix localization problem Hajek et al. (2018), for which the goal is to recover a submatrix of elevated mean (corresponding to the entries in $C_+$). Taking $a = -b$ yields the $\mathbb{Z}_2$-synchronization problem Bandeira et al. (2017) after rescaling; see Remark A.2. Our scaling choice allows both submatrix localization and $\mathbb{Z}_2$-synchronization to be handled under a unified model. We also consider the Stochastic Block Model (SBM) in the logarithmic-degree regime, which is the relevant regime for exact recovery.

**Definition 2.3** (Stochastic Block Model (SBM)). *Fix any $\rho \in (0,1)$ and $a_1, a_2, b > 0$. Then the model $\mathsf{SBM}_n(\rho, a_1, a_2, b)$ is a special case of $\mathsf{GBM}_n(\rho, \mathcal{P}_+, \mathcal{P}_-, \mathcal{Q})$ with*

$$\mathcal{P}_+ \equiv \mathsf{Bern}\left(\frac{a_1 \log n}{n}\right), \mathcal{P}_- \equiv \mathsf{Bern}\left(\frac{a_2 \log n}{n}\right), \text{ and } \mathcal{Q} \equiv \mathsf{Bern}\left(\frac{b \log n}{n}\right).$$

Finally, we consider a generic side information channel.

**Definition 2.4** (Side Information (SI)). *For any domain $\mathcal{Y}$, distributions $(\mathcal{S}_+, \mathcal{S}_-)$ supported on $\mathcal{Y}$, and $\sigma^* \in \{\pm 1\}^n$, we say that $y \sim \mathsf{SI}(\sigma^*, \mathcal{S}_+, \mathcal{S}_-)$, where $y \in \mathcal{Y}^n$ such that for any $i \in [n]$,*

$$y_i \sim \mathcal{S}_+, \text{ if } \sigma_i^* = +1 \quad \text{and} \quad y_i \sim \mathcal{S}_- \text{ if } \sigma_i^* = -1.$$

*The entries $\{y_i\}_{i \in [n]}$ are independent conditional on $\sigma^*$. We assume that the likelihoods $\mathcal{S}_+(y_i)$ and $\mathcal{S}_-(y_i)$ are computable in $O(1)$ time.*

**Definition 2.5** (Gaussian Features (GF)). *The model $y \sim \mathsf{GF}(\sigma^*, \upsilon_+, \upsilon_-, \sigma^2)$ for $\upsilon_+, \upsilon_- \in \mathbb{R}^d$ and $\sigma^2 > 0$ is a special case of $\mathsf{SI}$ with $\mathcal{Y} = \mathbb{R}^d$ and $\mathcal{S}_+ \equiv \mathcal{N}(\upsilon_+, \sigma^2 I_d)$ and $\mathcal{S}_- \equiv \mathcal{N}(\upsilon_-, \sigma^2 I_d)$.*

**Definition 2.6** (Binary Erasure Channel (BEC)). *For any $\sigma^* \in \{\pm 1\}^n$ and $\epsilon \in (0, 1]$, we say $y \sim \mathsf{BEC}(\sigma^*, \epsilon)$ where each entry of $\sigma^*$ is erased to 0, independently with probability $\epsilon$, to form $y \in \{-1, 0, +1\}^n$.*

**Definition 2.7** (Binary Symmetric Channel (BSC)). *For any $\sigma^* \in \{\pm 1\}^n$ and $\alpha \in (0, 1/2]$, we say $y \sim \mathsf{BSC}(\sigma^*, \alpha)$ where each entry of $\sigma^*$ is flipped independently with probability $\alpha$, to form $y \in \{\pm 1\}^n$.*

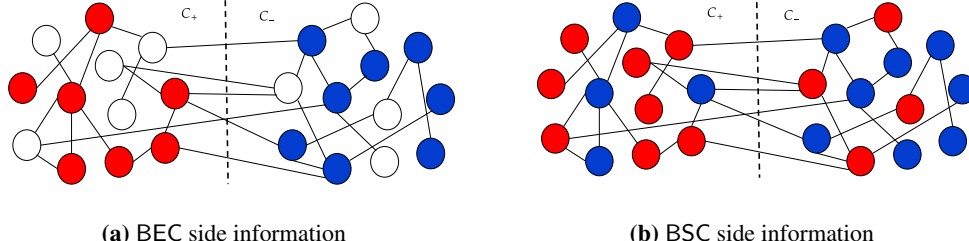

**(a)** BEC side information                     **(b)** BSC side information

**Figure 1:** Visualization of BSC and BEC side information. The red-colored, blue-colored, and uncolored vertices have side information labels of $+1$, $-1$, and 0 respectively.

## 2.2 EXACT RECOVERY.

Our goal is to exactly recover the community labels $\sigma^*$ given the observation matrix $A$ and the side information $y$ when available, as formalized below.

**Definition 2.8** (Exact Recovery). *We say that an estimator $\hat{\sigma}$ succeeds if*

*(i) $\hat{\sigma} \in \{\pm\sigma^*\}$ when $\mathcal{P}_+ \equiv \mathcal{P}_-, \rho = 1/2$ and there is no side information (the symmetric case);*

*(ii) $\hat{\sigma} = \sigma^*$, when $\mathcal{P}_+ \not\equiv \mathcal{P}_-$ or $\rho \neq 1/2$ or side information is present (non-symmetric case).*

*Otherwise, we say $\hat{\sigma}$ fails. We say that $\hat{\sigma}$ achieves exact recovery if $\lim_{n \to \infty} \mathbb{P}(\hat{\sigma} \text{ succeeds}) = 1$.*

Note that in the symmetric setting described in Definition 2.8, it is not possible to recover $\sigma^*$ with high probability, and we can only hope to recover the partition. In all other cases, we wish to recover the labels and not just the partition. All our positive results will demonstrate recovery in this strong sense. The optimal predictor for any model and side information is the one which has maximum posterior probability given the observation matrix $A$ and the side information $y$, when it is present.

**Definition 2.9** (MAP Estimator). *Consider the observation matrix $A$ and the side information $y$ from some side information channel. We define the Maximum A Posteriori (MAP) estimator as*

$$\hat{\sigma}_{\mathrm{MAP}} = \arg \max_{\sigma \in \{\pm 1\}^n} \mathbb{P}(\sigma^* = \sigma \mid A, y).$$

*When no side information is present, define $\hat{\sigma}_{\mathrm{MAP}} = \arg \max_{\sigma \in \{\pm 1\}^n} \mathbb{P}(\sigma^* = \sigma \mid A)$.*

## 2.3 INFORMATION THEORETIC THRESHOLD FROM DREVETON ET AL. (2024)

We now discuss the information-theoretic limits of exact recovery derived in Dreveton et al. (2024) in our notation. Define the *Chernoff coefficient* across the pair of communities as

$$\mathrm{CH}_t(+, -) = (1 - t)\left[\rho \, \mathrm{D}_t(\mathcal{P}_+ \| \mathcal{Q}) + (1 - \rho)\mathrm{D}_t(\mathcal{Q} \| \mathcal{P}_-) + \frac{1}{n}\mathrm{D}_t(\mathcal{S}_+ \| \mathcal{S}_-).\right] \quad (3)$$

Here $\mathrm{D}_t(\mathcal{A} \| \mathcal{B})$ is the Rényi divergence of order $t$ between any two laws $(\mathcal{A}, \mathcal{B})$, such that they are either both continuous or discrete, is given by

$$\mathrm{D}_t(\mathcal{A} \| \mathcal{B}) := \frac{1}{(t-1)} \log \mathbb{E}_{x \sim \mathcal{B}}\left[\left(\frac{\mathcal{A}(x)}{\mathcal{B}(x)}\right)^t\right]. \quad (4)$$

Define the limit $L : (0, 1) \to \mathbb{R}_{\geq 0} \cup \{+\infty\}$ by

$$L(t) = \lim_{n \to \infty} \frac{n}{\log n} \mathrm{CH}_t(+, -). \quad (5)$$

We restrict ourselves to the laws such that $L(t)$ is well defined[2]. Then the information theoretic limit is characterized by

$$I^* = \sup_{t \in (0,1)} L(t), \qquad (6)$$

where by convention, we consider the supremum to be $+\infty$ if $L(t)$ is unbounded in $t \in (0,1)$ or there exists $t \in (0,1)$ such that $L(t) = +\infty$. Then the following proposition is a minor variant of the two community case of (Dreveton et al., 2024, Theorem 1), characterizing the fundamental limit for exact recovery for the GBM.

**Proposition 2.10** (Dreveton et al. (2024)). *Let $\rho \in (0,1)$ and $(\mathcal{P}_+, \mathcal{P}_-, \mathcal{Q})$ be probability laws. Let $(A, \sigma^*) \sim \mathsf{GBM}_n(\rho, \mathcal{P}_+, \mathcal{P}_-, \mathcal{Q})$ and we observe $A$. Optionally, for side information laws $(\mathcal{S}_+, \mathcal{S}_-)$, we observe $y \sim \mathsf{SI}(\sigma^*, \mathcal{S}_+, \mathcal{S}_-)$. We assume the laws are such that $I^*$ in (6) is well-defined. Then*

1. *If $I^* > 1$, then the Maximum A Posteriori estimator $\hat\sigma_{\mathrm{MAP}}$ achieves exact recovery.*

2. *Additionally, if $I^* < 1$ and the function $L(t)$ is strictly concave then no estimator $\hat\sigma$ achieves exact recovery.*

This variant follows from (Dreveton et al., 2024, Theorem 1) by observing that the success of MAP proof does not rely on the strict concavity of $L(t)$ and the proof goes through even if $I^* = +\infty$. The impossibility result is completely equivalent to theirs. We discuss the necessary proof changes in Appendix A.4. We note that the case of no side information can be realized by considering side information laws $(\mathcal{S}_+, \mathcal{S}_-)$ that always deterministically output 0; in this case $\mathrm{D}_t(\mathcal{S}_+ \| \mathcal{S}_-) = 0$. Finally, this information theoretic limit in (6) can also be interpreted as a signal-to-noise ratio; we discuss this interpretation in Appendix A.1 for the resulting expressions of both ROS and SBM, and for each side information channel BEC, BSC, and GF.

## 3 MAIN RESULTS: ALGORITHMIC ACHIEVABILITY FOR ROS AND SBM

The main results of the paper are to design an optimal spectral algorithm for ROS and SBM.

**Theorem 1** (Optimality for ROS). *Fix $\rho \in (0,1)$ and $a, b \in \mathbb{R}$ such that $\max\{|a|, |b|\} > 0$. Let $(A, \sigma^*) \sim \mathsf{ROS}_n(\rho, a, b)$. Optionally, consider a side information channel $y \sim \mathsf{SI}(\sigma^*, \mathcal{S}_+, \mathcal{S}_-)$ such that $I^*$ in (6) is well-defined. Then there is a spectral algorithm (Algorithm 2) that returns the estimator $\hat\sigma_{\mathrm{spec}}$ which achieves exact recovery whenever $I^* > 1$.*

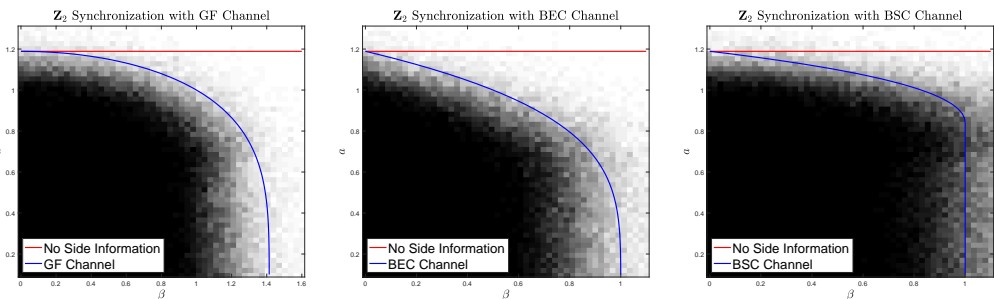

**Figure 2:** We consider the $\mathbb{Z}_2$-Synchronization model which is $\mathsf{ROS}_n(1/2, a, -a)$. With $n = 300$, for each type of side information channel of "strength" $\beta$ (see Appendix A.1), we validate the performance of the spectral algorithm over $N = 50$ trails over a grid of values for $(\beta, a)$. Lighter pixels correspond to higher rate of success. The blue and red curves are theoretical thresholds with and without side information respectively.

**Theorem 2** (Optimality for SBM). *Let $\rho \in (0,1)$ and $a_1, a_2, b > 0$. Let $(A, \sigma^*) \sim \mathsf{SBM}_n(\rho, a_1, a_2, b)$. Optionally, consider a side information channel $y \sim \mathsf{SI}(\sigma^*, \mathcal{S}_+, \mathcal{S}_-)$ such that $I^*$ in (6) is well-defined. Then there is a spectral algorithm (Algorithms 5 and 6) that returns the estimator $\hat\sigma_{\mathrm{spec}}$ which achieves exact recovery, whenever $I^* > 1$.*

In Appendix C, we will verify that for both SBM and ROS, the function $L(t)$ (and thus also $I^*$) is well-defined and $L(t)$ is strictly concave for each (i) no side information (ii) GF, BEC, and BSC

---

[2]We expect $L(t)$ to be well-defined for most sequences of laws, e.g. $\mathsf{ROS}_n$ or $\mathsf{SBM}_n$. We mention this explicitly just for the formalism, and to rule out pathological sequences of laws that do not converge as $n \to \infty$.

channels. This along with Theorems 1, 2 and Proposition 2.10 establishes that the spectral algorithm succeeds up to the information-theoretic limit in these settings. We also run simulations to verify our theoretical results (see Figure 2 and Appendix A.1). The main novelty of our analysis is to establish the rigorous connection between the spectral estimator and the so-called genie-aided estimators, which we now discuss.

## 4 GENIE-AIDED ESTIMATORS

In the genie-aided setting (see e.g., Abbe (2017)), we suppose that all labels but the $i^{\text{th}}$ are known, and the goal is to determine the $i^{\text{th}}$ label. More formally, let $\sigma^*_{-i}$ denote the true labels, apart from $\sigma^*_i$. The optimal estimator for the $i^{\text{th}}$ label is given by

$$\hat{\sigma}_{\text{Gen},i} = \begin{cases} +1, & \text{if } \mathbb{P}(\sigma^*_i = +1 \mid A, y, \sigma^*_{-i}) \geq \mathbb{P}(\sigma^*_i = -1 \mid A, y, \sigma^*_{-i}). \\ -1, & \text{otherwise.} \end{cases} \tag{7}$$

(where the conditioning on $y$ is omitted when there is no side information present). Moreover, we say that $\hat{\sigma}_{\text{Gen},i}$ fails on an instance if the posterior probability of the incorrect label is *strictly* greater than the posterior probability of the correct label. The following lemma rigorously establishes the intuitive claim that the failure of some genie-aided estimator implies the global MAP also fails.

**Lemma 4.1.** *Let* $\rho \in (0,1)$ *and* $\mathcal{P}_+, \mathcal{P}_-, \mathcal{Q}$ *be any distributions. Let* $(A, \sigma^*) \sim$ $\mathsf{GBM}_n(\rho, \mathcal{P}_+, \mathcal{P}_-, \mathcal{Q})$. *Optionally, let* $y \sim \mathsf{SI}(\sigma^*, \mathcal{S}_+, \mathcal{S}_-)$ *in* $\mathcal{Y}^n$ *where* $y \in \mathcal{Y}^n$ *is the side information for any laws* $(\mathcal{S}_+, \mathcal{S}_-)$ *over* $\mathcal{Y}$. *Define the genie-aided estimators* $\{\hat{\sigma}_{\text{Gen},i} : i \in [n]\}$ *and* $\hat{\sigma}_{\text{MAP}}$ *for the respective model. Then*

$$\mathbb{P}(\exists i \in [n] : \hat{\sigma}_{\text{Gen},i} \text{ fails}) \leq \mathbb{P}(\hat{\sigma}_{\text{MAP}} \text{ fails}).$$

**Genie scores.** We form a vector $z^* \in (\mathbb{R} \cup \{\pm\infty\})^n$, called the *genie score* vector, where $z^*_i$ records the log of the ratio of posterior probabilities of the label $\sigma^*_i$ given $\sigma^*_{-i}$ by a genie. That is,

$$z^*_i = \log \left( \frac{\mathbb{P}(\sigma^*_i = +1 \mid A, y, \sigma^*_{-i})}{\mathbb{P}(\sigma^*_i = -1 \mid A, y, \sigma^*_{-i})} \right) \quad \text{or} \quad z^*_i = \log \left( \frac{\mathbb{P}(\sigma^*_i = +1 \mid A, \sigma^*_{-i})}{\mathbb{P}(\sigma^*_i = -1 \mid A, \sigma^*_{-i})} \right), \tag{8}$$

in the cases of side information and no side information respectively. Then the optimal genie-based estimator corresponds to $\hat{\sigma}_{\text{Gen},i} = \text{sgn}(z^*_i)$. Throughout the paper, we treat log as function from $[0, \infty]$ to the extended real line and implicitly use standard conventions of extended real line algebra. See Appendix B.1. The following lemma gives the form of the genie scores without or with node-attributed side information. Here we define $C^{-i}_+ := C_+ \setminus \{i\}$ and $C^{-i}_- := C_- \setminus \{i\}$.

**Lemma 4.2.** *Consider any* $\rho \in (0,1)$ *and laws* $\mathcal{P}_+, \mathcal{P}_-$ *and* $\mathcal{Q}$. *Let* $(A, \sigma^*) \sim$ $\mathsf{GBM}_n(\rho, \mathcal{P}_+, \mathcal{P}_-, \mathcal{Q})$. *Optionally, let* $y \sim \mathsf{SI}(\sigma^*, \mathcal{S}_+, \mathcal{S}_-)$ *where* $y \in \mathcal{Y}^n$ *is the side information for any laws* $(\mathcal{S}_+, \mathcal{S}_-)$ *over* $\mathcal{Y}$. *Then for any* $i \in [n]$, *the genie score for the* $i^{\text{th}}$ *label is given by*

- *No side information:*

$$z^*_i = \sum_{j \in C^{-i}_+} \log \left( \frac{\mathcal{P}_+(A_{ij})}{\mathcal{Q}(A_{ij})} \right) + \sum_{j \in C^{-i}_-} \log \left( \frac{\mathcal{Q}(A_{ij})}{\mathcal{P}_-(A_{ij})} \right) + \log \left( \frac{\rho}{1-\rho} \right).$$

- *Side information:*

$$z^*_i = \sum_{j \in C^{-i}_+} \log \left( \frac{\mathcal{P}_+(A_{ij})}{\mathcal{Q}(A_{ij})} \right) + \sum_{j \in C^{-i}_-} \log \left( \frac{\mathcal{Q}(A_{ij})}{\mathcal{P}_-(A_{ij})} \right) + \log \left( \frac{\rho}{1-\rho} \right) + \log \left( \frac{\mathcal{S}_+(y_i)}{\mathcal{S}_-(y_i)} \right).$$

**Motivation behind our Spectral Strategy.** Due to the independence of $A$ and $y$, conditioned on $\sigma^*$, the genie score $z^*_i$ under side information is simply the score without side information with the addition of the log-likelihood ratio of the node attribute $y_i$. Intuitively, we devise a principled method for determining the weights of the eigenvector combination and threshold value in our spectral algorithm such that $i^{\text{th}}$ entry of the vector formed approximates the the genie score $z^*_i$ under no side information. Remarkably, the spectral algorithm will approximate this statistic from eigenvectors of $A$, despite not knowing $C^{-i}_+$ and $C^{-i}_-$. In light of Lemma 4.2, it is also immediate to see the motivation behind the use of side information prescribed in (1). In particular, the genie scores undergo exactly the same transformation when the side information becomes available according to Lemma 4.2. We derive the closed-form expressions for these additive factors in (1) for our examples of side information channels.

**Lemma 4.3.** *For any $i \in [n]$, the log-likelihood ratio of side information label is given by*

- GF *(Gaussian features): When $y \sim \mathsf{GF}(\sigma^*, v_+, v_-, \sigma^2)$ for $v_+, v_- \in \mathbb{R}^d$ and $\sigma^2 > 0$,*

$$\log\left(\frac{\mathcal{S}_+(y_i)}{\mathcal{S}_-(y_i)}\right) = \frac{\|y_i - v_-\|_2^2 - \|y_i - v_+\|_2^2}{2\sigma^2}.$$

- BEC *channel: When $y \sim \mathsf{BEC}(\sigma^*, \epsilon)$ for $\epsilon \in (0, 1]$,*

$$\log\left(\frac{\mathcal{S}_+(y_i)}{\mathcal{S}_-(y_i)}\right) = \begin{cases} +\infty, & \text{if } \sigma_i^* = +1; \\ -\infty, & \text{if } \sigma_i^* = -1; \\ 0, & \text{otherwise.} \end{cases}$$

- BSC *channel: When $y \sim \mathsf{BSC}(\sigma^*, \alpha)$ for $\alpha \in (0, 0.5]$,*

$$\log\left(\frac{\mathcal{S}_+(y_i)}{\mathcal{S}_-(y_i)}\right) = \log\left(\frac{1-\alpha}{\alpha}\right) y_i$$

**Genie Success with Margin above the IT Limit:** We start by a crucial observation about the genie scores, which will be important in analyzing the spectral algorithm. We show that above the information-theoretic limit, the genie score have the correct sign corresponding to the true label, but with a *sufficient margin*. We show the for any model GBM and optionally a side information channel, whenever $I^* > 1$, there exists a constant $\delta > 0$ such that with high probability

$$\min_{i \in [n]} \sigma_i^* z_i^* > \delta \log n, \tag{9}$$

formalized in Lemma D.1.

## 5 Genie to Spectral Algorithm (for ROS and SBM)

In light of Eq. (9), if an algorithm can approximate the genie score up to an additive error of $o(\log n)$, then sign thresholding will correctly recover all the labels. As ROS and SBM entries are from exponential families, the log-likelihood is affine function of the observations $A_{ij}$, and thus, the genie score vector is an affine function of $A$. I.e.

$$z^* \approx Aw^* + \gamma \mathbf{1}_n \quad \text{(in } \ell_\infty \text{ norm)}, \tag{10}$$

for a certain $\gamma \in \mathbb{R}$ and $w^* \in \mathbb{R}^n$ with entries $(w_+, w_-) \in \mathbb{R}^2$ in the locations of $C_+$ and $C_-$ respectively. See Lemmas F.1 (ROS) and G.2 (SBM) for precise statements. Note that $(w_+, w_-, \gamma)$ are just scalars that can be calculated from the model parameters and do not depend on $\sigma^*$. The main power of the genie lies in forming the vector $w^*$, which requires knowing the locations $C_+$ and $C_-$. Then the question that remains is how one may come up with a proxy for $w^*$ without knowing $C_+$ and $C_-$ such that the genie score is well-approximated.

**Warm-up: Degree Profiling Algorithm for** BEC **and** BSC **Channels.** Under BEC and BSC channels, a natural question is then what if we simply trust the side information labels $y \in \{-1, 0, +1\}^n$ and $y \in \{-1, +1\}^n$ respectively. In particular, we use the proxy for $w^*$ where $w_i^{\mathrm{dp}} = w_+ \mathbf{1}[y_i = +1] + w_- \mathbf{1}[y_i = -1]$ where we just trust the side information on the face value and use the locations of $S_+ := \{i \in [n] : y_i = +1\}$ and $S_- := \{i \in [n] : y_i = -1\}$ instead of $C_+$ and $C_-$.

It is known that the asymptotic information-theoretic threshold does not shift for the BEC and BSC channels unless $\epsilon = O(n^{-\beta})$ and $\alpha = O(n^{-\beta})$ for some $\beta > 0$ (Dreveton et al., 2024; Saad & Nosratinia, 2018). Therefore, the side information $y$ already satisfies almost exact recovery criterion, recovering $(1 - o(1))$ labels correctly. Hence, we have $|C_+ \Delta S_+|, |C_- \Delta S_-| = o(n)$ with high probability. We then show that for this proxy choice of scores $z^{\mathrm{dp}}$, we indeed have $\|z^{\mathrm{dp}} - z^*\|_\infty = o(\log n)$, and thus, the degree-profiling algorithm succeeds down to the shifted threshold. The formal theorems and algorithms can be found in Appendix F.3 and G.3.

**Remark 5.1.** *We emphasize that the degree profiling algorithm has an important caveat that it would fail to recover labels exactly from a tuple $(A, y)$, when side information strength is "weak"*

*or completely absent, even though the recovery was possible just from A. This is because just using $y$ as a proxy to mimic the genie is detrimental if the primary signal is present in A. To overcome this, one has to get preliminary almost exactly correct labels utilizing A, and then mimic the genie using this preliminary labeling as a proxy. This exactly corresponds to the two-stage strategies described in Appendix A.3. Remarkably, our spectral algorithm in just one stage recovers all the labels correctly from $(A, y)$ (including no side information), whenever it is possible to do so, and that too for any side information channel.*

## 5.1 SPECTRAL ALGORITHM

**Spectral Algorithm.** The spectral algorithm primarily uses the signal from eigenvectors of $A$ to approximate $z^*$, rather than relying on side information. Therefore, the spectral algorithm affords significantly more versatility than the degree-profiling algorithm, and succeeds without any clean-up step with more general or even no side information. The design of our spectral algorithm is informed by the entrywise eigenvector analysis result of Abbe et al. (2020), which allows us to say that the leading $K$ eigenvectors of $A$ satisfy

$$u_i \approx \frac{Au_i^*}{\lambda_i^*}, \tag{11}$$

where $(\lambda_i^*, u_i^*)$ is the corresponding eigenvector of the expectation matrix $\mathbb{E}[A \mid \sigma^*]$, and the approximation is in the $\ell_\infty$ norm. Here $K = 1$ for ROS and $K = 2$ for SBM. For both models, the matrix $\mathbb{E}[A \mid \sigma^*]$ has a "block" structure, so do its top eigenvectors as well. Thus, we can find an appropriate linear combination coefficients $(c_i)_{i \in [K]}$ such that the optimal genie combination vector

$$w^* \approx \sum_{i=1}^{K} \frac{c_i}{\lambda_i^*} u_i^*. \tag{12}$$

It is important to note that computing $(c_i)_{i \in [K]}$ does not require knowing the ground-truth $\sigma^*$. This is because the block structure of $w^*$ and $\{u_i^*\}_{i \in [K]}$ align leaving the coefficients always the same.

---

**Algorithm 1** An informal sketch of the spectral algorithm

---

**Input:** An observation matrix $A \in \mathbb{R}^{n \times n}$ and the model parameters[3]. Optionally, the side information $y \in \mathcal{Y}^n$ and the associated parameters.

**Output:** An estimate of community assignments $\hat{\sigma}_{\text{spec}}$.

1: Compute coefficients $(c_i)_{i \in [K]}$ from the model parameters such that $w^* \approx \sum_{i=1}^{K} \frac{c_i}{\lambda_i^*} u_i^*$.
2: Compute the top $K$ eigenpairs of $A$ denoted by $(u_i, \lambda_i)$.
3: Form the spectral score vector:
   - No side information:

   $$z^{\text{spec}} = \gamma \mathbf{1}_n + \sum_{i=1}^{K} c_i u_i.$$

   - Side information: If side information is present, further update

   $$z^{\text{spec}} = z^{\text{spec}} + \log\left(\frac{\mathcal{S}_+}{\mathcal{S}_-}(y)\right).$$

4: $\hat{\sigma}_{\text{spec}} = \text{sgn}(z^{\text{spec}})$.

---

Then the spectral score vector, when the side information is absent, is given by

$$z^{\text{spec}} = \gamma \mathbf{1}_n + \sum_{i=1}^{K} c_i u_i \stackrel{(11)}{\approx} \gamma \mathbf{1}_n + \sum_{i=1}^{K} c_i \frac{Au_i^*}{\lambda_i^*} \stackrel{(12)}{\approx} \gamma \mathbf{1}_n + Aw^* \stackrel{(10)}{\approx} z^*.$$

This achieves an $\ell_\infty$ approximation of the genie score under no side information using eigenvectors of $A$. When side information is present, due to Lemma 4.2 , it suffices to add the log-likelihood ratio vector from (1). The approximations are tight enough such that we achieve

$$\|z^{\text{spec}} - z^*\|_\infty = o(\log n).$$

---

[3]As common in the exact recovery literature, we assume that model parameters are already given. In Remark A.1, we discuss how to estimate them if they are unknown, and in what cases the spectral algorithm is robust enough to continue to succeed when we use these estimates as proxies.

Recalling that the genie scores succeed with a margin of $\Omega(\log n)$ by (9) above the IT threshold immediately gives us the optimality of the spectral algorithm.

**Remark 5.2.** *We remark that, to show the impossibility of recovery, due to the optimality of MAP, Dreveton et al. (2024) shows that the MAP estimator (Definition 2.9) fails below the threshold. However, their proof indeed shows an even stronger statement that below the threshold, even the genie-aided estimators (7) fail. Both their impossibility and our achievability of spectral algorithm results are driven by the genie-aided estimators, so it comes as no surprise that there is a certain threshold collapse: the genie-aided estimators, spectral estimator, and MAP estimator all achieve the same recovery threshold. The entire discussion can be summarized in Figure 3.*

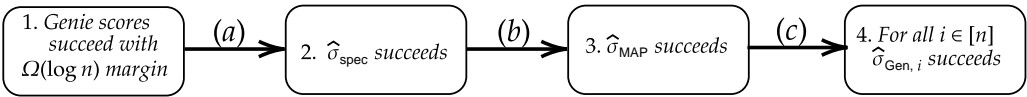

**Figure 3:** Summary of our unified proof framework. Our spectral algorithm is designed such that $(a)$ holds. Note that: $(b)$ follows from the optimality of $\hat{\sigma}_{\mathrm{MAP}}$ and $(c)$ follows from Lemma 4.1. Finally, we get the threshold collapse phenomenon because below the IT threshold, the event in the fourth block happens with probability $o(1)$, and above the IT threshold, the event in the first block happens with high probability by (9).

**Remark 5.3** (Runtime). *Our spectral algorithm has nearly linear runtime in the input size (# of non-zero entries of A). That is for* SBM *and* ROS*, the runtime is $\tilde{O}(n)$ (due to a sparse network) and $\tilde{O}(n^2)$ respectively. The implementation details are already laid out in (Gaudio & Joshi, 2023, Section 4) and (Dhara et al., 2022b, Remark 2.1); the idea is to compute top eigenpairs sequentially using the faster eigenvector computation of Garber et al. (2016). The use of side information requires only $O(n)$ additional computation.*

## 6 DISCUSSION AND FUTURE WORK

In this paper, we provide a systematic treatment of designing optimal spectral algorithms for two-community matrix inference problems under side information, focused on the Bernoulli and Gaussian cases. From a technical standpoint, our work makes a rigorous connection between spectral algorithms and genie-aided estimators, characterizing their effectiveness in achieving sharp thresholds for various exact recovery problems in a recent line of work. We refer the reader to Appendix A.2 for a detailed discussion on this. Understanding the capabilities of such vanilla spectral algorithms, without any clean-up stage, is of fundamental interest; we hope this perspective will guide the design and analysis of spectral algorithms for exact community recovery problems moving forward. Some directions for future work include:

- **Exact recovery in Gaussian Mixture Block Model:** In a recent work, Li & Schramm (2023) proposed an alternative model to better capture real-world networks and sketched out the general landscape for recovery by studying almost exact recovery. What about exact recovery? Interestingly, Li & Schramm (2023) proposed exactly the same vanilla spectral algorithm and showed it achieves almost exact recovery. Does it also succeed for exact recovery?

- **More general degree-profiling:** A natural analogue of degree-profiling algorithm for more general side information beyond BEC and BSC is to sign-threshold the log-likelihood ratio vector $\log\left(\frac{S_+}{S_-}(y)\right)$ to compute a preliminary labeling. We conjecture that whenever such side information is sufficient to shift the exact recovery threshold, the preliminary assignment will already correctly compute a $(1 - o(1))$-faction of labels. Emulating the genie then using this labeling as a proxy should succeed in exact recovery (though with the limitation detailed in Remark 5.1).

- **More general settings:** To design spectral algorithms for more than two communities and more general observation distributions $(\mathcal{P}_+, \mathcal{P}_-, \mathcal{Q})$ from a class of exponential families (also see details in Appendix A.2).

**Acknowledgements.** J.G. was supported in part by NSF CCF-215410, and N.J. was supported in part by the Institute for Data, Econometric, Algorithms, and Learning (NSF TRIPODS HDR). The authors thank Raghav Sinha from the Illinois Mathematics and Science Academy for a preliminary empirical study.

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

# Appendices

## A   DEFERRED DISCUSSIONS

**Remark A.1** (Estimating Model Parameters when Unknown). *In our spectral algorithm design, we assumed that the model parameters are already given as the input. If the distribution parameters for $\mathcal{P}_+$, $\mathcal{P}_-$, and $\mathcal{Q}$ are unknown, one can perform almost exact recovery to find a preliminary labeling and use the preliminary labeling to estimate the distributional parameters. One can then use these estimates as a proxy in the spectral algorithm or even. As a result, the practical advantage of the spectral algorithm over two-stage approaches is not clear if the model parameters are not known, however, the spectral algorithm's analysis has still been of great interest due to its elegance and for theoretical investigations. The plug-in estimates created this way for SBM and ROS parameters suffice for the spectral algorithm to continue to succeed. We can estimate the parameters for $\mathcal{S}_+$ and $\mathcal{S}_-$ similarly from preliminary labelings if they are parameterized distributions like BEC, BSC, GF However, whether the estimates would be good enough for the spectral algorithm to continue to succeed depends on the channel. For our examples:*

- GF*: the estimated means from preliminary labelings suffice for the spectral algorithm to continue to work.*
- BEC*: the algorithm does not use $\varepsilon$ to compute $\log\left(\frac{\mathcal{S}_+}{\mathcal{S}_-}(y)\right)$ (Lemma 4.3), so there is no need to estimate.*
- BSC*: it remains unclear whether the estimated $\alpha$ would be good enough.*

**Remark A.2** ($\mathbb{Z}_2$-synchronization as rescaled ROS). *We remark that the $\mathbb{Z}_2$-synchronization problem is typically formulated as $A_{ij} = x_i^* x_j^* + \sigma W_{ij}$, where $x^*$ is an unknown vector is chosen uniformly at random from the set $\{\pm 1\}^n$, $W$ is a zero-diagonal symmetric matrix with independent entries sampled from $\mathcal{N}(0,1)$ (Bandeira et al., 2017). In that case, the relevant parametrization of $\sigma$ is $\sigma = c\sqrt{\frac{n}{\log n}}$, as $\sigma = \sqrt{\frac{n}{2\log n}}$ is the threshold value for exact recovery Bandeira et al. (2017). Thus, taking $a = 1/\sqrt{c}, b = -1/\sqrt{c}$ and $\rho = 1/2$ in our ROS model (Definition 2.2) produces a matrix $A$ such that*

$$A_{ij} = \frac{1}{c}\sqrt{\frac{\log n}{n}}\, x_i^* x_j^* + W, \quad x^* \sim \mathrm{Uniform}(\{\pm 1\}^n).$$

*After scaling $A$ by $c\sqrt{n/\log n}$, we achieve the standard model $x^* x^{*\top} + \sigma W$ (with zero diagonal).*

### A.1   ADDITIONAL SIMULATIONS AND INTERPRETING INFORMATION THEORETIC LIMITS

**Empirical Validations.**   In Figure 2, we consider the $\mathbb{Z}_2$-Synchronization setting, which corresponds to $\mathrm{ROS}(\rho, a, b)$ for $b = -a$ and $\rho = 1/2$. We considered our three examples of side information whose parameters are further parameterized by a strength parameter $\beta \geq 0$.

- Gaussian Features (Definition 2.5): We consider $\sigma = 1$ and $v_+ = \beta\sqrt{\log n} \in \mathbb{R}$ and $v_- = -\beta\sqrt{\log n} \in \mathbb{R}$.
- BEC Channel (Definition 2.6): We set $\epsilon = n^{-\beta}$.
- BSC Channel (Definition 2.7): We set $\alpha = 1/(n^\beta + 1) = \Theta(n^{-\beta})$.

In addition to $\mathbb{Z}_2$-Synchronization (Gaussian edge observations), we also verify that the spectral algorithm achieves the information-theoretic limits under side information also for Bernoulli edge observations (Theorem 2). In particular, we will consider Symmetric SBM for which $a_1 = a_2 = a$ and $\rho = 1/2$ in Definition 2.3. As there are now more than two parameters ($a, b$, and side information), we fix the side information strength parameter $\beta$ and vary $(a, b)$. See Figures 4 and 5 for BEC and BSC channels respectively.

**Symmetric SBM Threshold.**   In this case, as we shall derive in Lemma C.6, the function $L(t)$ in (5) is given by

$$L(t) = \frac{1}{2}\left(a + b - a^t b^{1-t} - b^t a^{1-t}\right).$$

The function $L(t)$ has the unique minimizer at $t^* = 1/2$.

$$I^* = \sup_{t \in (0,1)} L(t) = \frac{1}{2}(\sqrt{a} - \sqrt{b})^2.$$

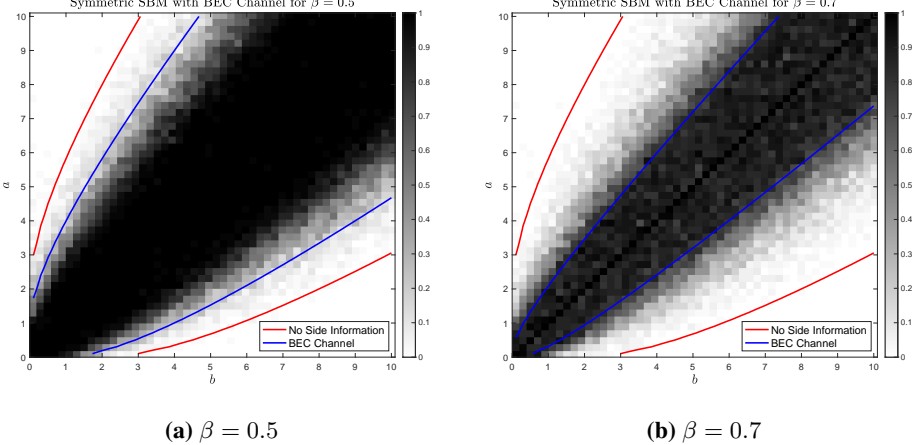

**(a)** $\beta = 0.5$              **(b)** $\beta = 0.7$

**Figure 4:** (BEC Channel). For the symmetric SBM, i.e. $\mathsf{SBM}_n(1/2, a, b)$. We let $n = 300$ and consider the BEC side information channel with $\epsilon = n^{-\beta}$ for $\beta \in \{0, 5, 0.7\}$ in Figures 4a and 4b respectively. We run the spectral algorithm (Algorithm 5) over $N = 50$ independent trials and plot its success rate for the exact recovery. Lighter pixels correspond to higher probability of success. Finally, the blue and red curves are the theoretical information-theoretic thresholds.

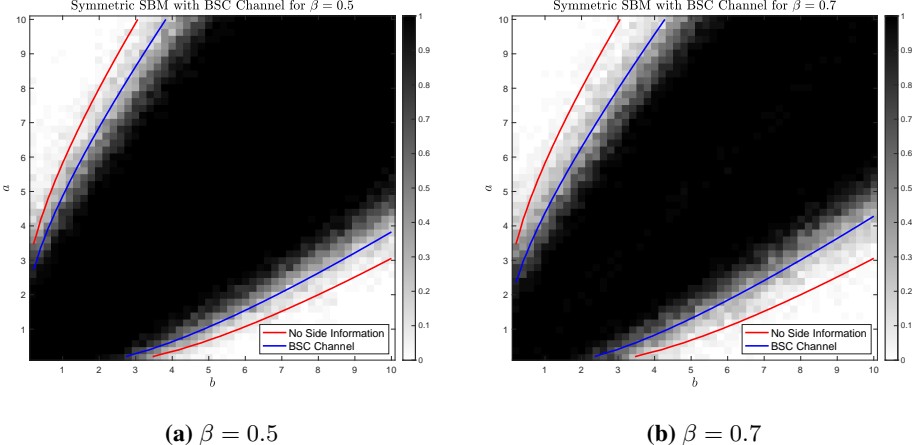

**(a)** $\beta = 0.5$              **(b)** $\beta = 0.7$

**Figure 5:** (BSC Channel). The setup is identical to the one in Figure 4, but now with the BSC channel with a flipping probability $\alpha = 1/(n^{\beta} + 1) = \Theta(n^{-\beta})$ for $\beta \in \{0, 5, 0.7\}$.

This is the well-known information-theoretic for the symmetric SBM (Abbe, 2017; Abbe et al., 2016), which can be interpreted as a signal strength.

**Rank One Spike Threshold.** For $\mathsf{ROS}(\rho, a, b)$ (Definition 2.2), let us define

$$\Psi := \Psi(\rho, a, b) = (a - b)^2 (\rho a^2 + (1 - \rho) b^2).$$

This quantity $\Psi$ can be interpreted as the signal-to-noise ratio, and controls the information-theoretic limit in $\mathsf{ROS}(\rho, a, b)$. In particular, in the absence of side information, we have:

$$L(t) = \frac{t(1 - t)}{2} \Psi(\rho, a, b), \text{ and thus, } I^* = \sup_{t \in (0,1)} L(t) = \frac{\Psi(\rho, a, b)}{8}.$$

**Side Information.** In the presence of side information, the function $L(t)$ shifts by a function of $t$ due to an additive term $\mathsf{D}_t(\mathcal{S}_+ \| \mathcal{S}_-)$ from (3). The final limits $I^*$ are then calculated by taking the $\sup_{t \in (0,1)}$, and thus, they do not necessarily have simple closed-form expressions. However, the additive shifts before taking the supremum have simplified expressions which we specify for each of the three channels.

- Gaussian Features (Definition 2.5): In GF with $\upsilon_+ = \boldsymbol{\beta}_+\sqrt{\log n}$ and $\upsilon_- = \boldsymbol{\beta}_-\sqrt{\log n}$

$$L_{\mathsf{GF}}(t) = L_{\text{no-SI}}(t) + \frac{t(1-t)\left\|\boldsymbol{\beta}_+ - \boldsymbol{\beta}_-\right\|_2^2}{2\sigma^2}\,.$$

  See Lemma C.2 for the derivation. The shifts depends on the signal strength of side information $\left\|\boldsymbol{\beta}_+ - \boldsymbol{\beta}_-\right\|_2^2$.

- BEC Channel (Definition 2.6): When $\epsilon = \Theta(n^{-\beta})$ for a parameter $\beta \geq 0$, we have
$$L_{\mathsf{BEC}}(t) = L_{\text{no-SI}}(t) + \beta\,.$$
  See Lemma C.3. The signal strength $\beta$ determines the shift.

- BSC Channel (Definition 2.7): When $\alpha = \Theta(n^{-\beta})$ for a signal strength parameter $\beta \geq 0$, by Lemma C.4, we have
$$L_{\mathsf{BSC}}(t) = L_{\text{no-SI}}(t) + \beta \min\{t, 1-t\}\,.$$

In Figure 6, we compare the final IT thresholds for BEC and BSC channels against the threshold without side information for ROS.

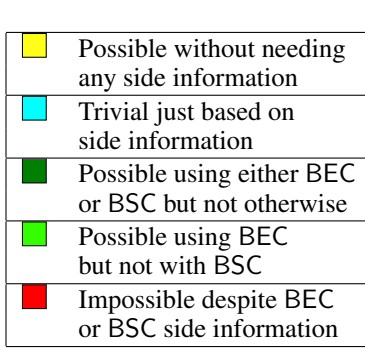

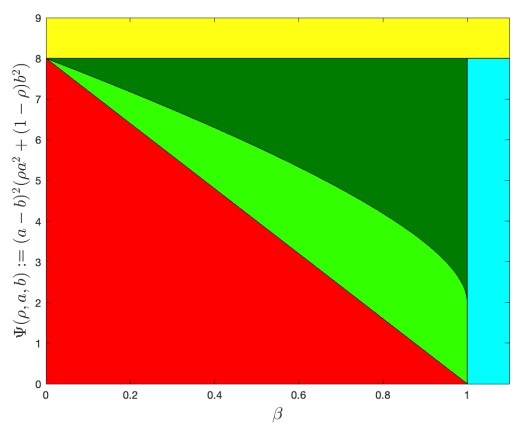

**Figure 6:** Visualization of Information Theoretic Thresholds for ROS. The colored regions in the above $\Psi$ vs $\beta$ plot indicate how BEC or BSC side information of strength $\beta$ helps.

## A.2 FUTURE EXTENSIONS AND PRIOR WORKS AND ON SPECTRAL ALGORITHMS

**Future Extensions.** We now describe, how we can generalize the spectral algorithm to more than two communities and more general distribution families. The entire genie-aided framework generalizes to the $K$-community case where each vertex now has $K$ score values $\{z_{i,(k)}^* : k \in [K], i \in [n]\}$.

The score for the $k^{\text{th}}$ community for the vertex $i$ is

$$z_{i,(k)}^* = \log \mathbb{P}(\sigma_i^* = k \mid A, y, \sigma_{-i}^*).$$

The optimal genie-based estimator is then simply given by $\hat{\sigma}_{\text{Gen},i} = \arg\max_{k \in [K]} z_{i,(k)}^*$. Mimicking the genie, the spectral algorithm now also computes $K$ different score vectors to approximate the statistics of the genie scores and takes the entrywise arg max to come up with the spectral estimator of the labeling.

We note that the entrywise eigenvectors behaviors of Abbe et al. (2020) for top eigenvectors continue to hold. Despite this, (Abbe et al., 2020, Appendix C.4) noted difficulties in designing spectral algorithms for more than two blocks due to the multiplicity of eigenvalues and eigenvectors being only computed up to a rotation. But we note that, except for these degenerate cases (some measure zero subset of parameters where exact recovery is possible), the algorithm should be able to emulate the genie-aided estimation and achieve optimality. It remains an interesting open question to handle these degenerate cases by a spectral strategy, which may require new ideas.

We expect that more general distribution families for pairwise observation matrix could be possible to handle. In particular, we first summarize the key technical properties we relied on for designing the spectral algorithms for ROS and SBM.

1. The genie-score vector takes a special form where it is an affine function of the entries of the matrices.
$$z^* \approx Aw^* + \gamma \mathbf{1}_n.$$

2. The optimal genie linear combination vector $w^*$ is in the span of eigenvectors of $\mathbb{E}[A \mid \sigma^*]$.

3. Lastly, the eigenvector $u$ of $A$, and its corresponding eigenvector $u^*$ of $\mathbb{E}[A \mid \sigma^*]$, by Abbe et al. (2016) we have
$$u \approx \frac{Au^*}{\lambda^*},$$
which allows us to emulate the genie using eigenvectors of $A$.

Property 1 holds more generally for any exponential family of distributions. The linear combinations in Property 2 needs to be designed for specific distribution families, but we expect this to be possible in great generality due to the block structure of the expectation matrices. Finally, one needs to verify the entrywise behavior of eigenvector $A$ holds. Abbe et al. has general conditions under which this behavior holds, however, verifying the technical conditions for the distribution families remains to be the main challenge.

**Prior Works.** We note that prior works on spectral algorithms mentioned in Section 1 are all essentially emulating the genie, and the models satisfy the aforementioned three properties. For example, prior works have used weighted adjacency matrices (Dhara et al., 2022a), and even multiple adjacency matrices (Dhara et al., 2023; Gaudio & Liu, 2024) in order to bring out "helpful" eigenvectors that can be used to mimic the genie estimators.

Interestingly, for the exact recovery problem in the hypergraph SBM from the similarity matrix $W$ (counting the hyperedges involving each pair of vertices), Gaudio & Joshi (2023) devised the same spectral strategy of computing the signs of the second eigenvector in the symmetric case and analyzed it by proving the $\ell_\infty$ behavior of eigenvectors. However, this strategy is known to be strictly suboptimal (Bresler et al., 2024) for this setting. This is because the genie scores are not affine functions of the entries of the matrices $W$ due to internal dependencies. However, it is interesting to note that the algorithm achieves the threshold of the min-bisection estimator (also not efficiently computable), which is the first order (linear) approximation of MAP. In other words, the eigenvector approximated the statistics that correspond to the linear terms of the genie scores. This suggests that even when the likelihood is not a linear function of the entries of the observation matrix, while the spectral algorithm may be suboptimal, using our framework it could still be possible to get a non-trivial performance guarantee.

### A.3 ADDITIONAL RELATED WORK

**Local to global amplification and two stage algorithms.** The SBM is known to exhibit *local-to-global amplification*, in the sense that whenever (local) recovery of a single vertex label given the labels of all other vertices is possible with probability $1 - o(1/n)$, then (global) exact recovery is possible with probability $1 - o(1)$ (see e.g., Abbe (2017)). Two-stage algorithms, which are prevalent in the literature (Abbe et al., 2016; Coja-Oghlan, 2006; Vu, 2018; Yun & Proutiere, 2014; 2016; Gaudio et al., 2024a), essentially leverage this statistical property.

**Community recovery under side information.** Community detection problems with side information have been studied in numerous settings. Saad and Nosratinias Saad & Nosratinia (2018) considered exact recovery in the symmetric, balanced SBM, under the BEC, BSC, and more general side information with $K$ features, where $K$ may grow with $n$. Additionally, an efficient two-stage exact recovery algorithm was proposed. Vector-valued side information was also studied in Saad & Nosratinia (2020), in the recovery of a planted dense subgraph of size $o(n)$. Community detection in the sparse setting under side information has received significant attention- see for example Mossel & Xu (2016); Cai et al. (2016); Kadavankandy et al. (2017); Kanade et al. (2016); Deshpande et al. (2018); Braun et al. (2022); we note that Deshpande et al. (2018) considers Gaussian side information with either Bernoulli or Gaussian pairwise observations. See also Zhang et al. (2014) which includes statistical physics conjectures for recovery thresholds derived from the cavity method. Numerous approaches for clustering have been proposed in the network science literature, such as Newman & Clauset (2016); Zanghi et al. (2010); Yang et al. (2009); Xu et al. (2012); Yang et al. (2013); Zhang et al. (2016); Gibert et al. (2012); Zhou et al. (2009); Günnemann et al. (2013); Cheng et al. (2011); Binkiewicz et al. (2017); see Bothorel et al. (2015) for a survey.

**Other inference problems with side information**    Related problems in the literature include document classification Krithara et al. (2008); Chang & Blei (2010) and text classification Balasubramanyan & Cohen (2011). A recent line of work studies the problem of community detection from correlated graphs Racz & Sridhar (2021); Gaudio et al. (2022), so that the additional graph plays the role of side information. See also Zhang et al. (2021), which considers attributed graph alignment. More broadly, inference with side information falls under the area of semi-supervised learning (see e.g. Van Engelen & Hoos (2020); Chapelle et al. (2002); Bair (2013); Basu et al. (2004); Newman & Clauset (2016)).

### A.4    PROOF REVIEW OF DREVETON ET AL. (2024) FOR PROPOSITION 2.10

We first argue that the impossibility result in Proposition 2.10 is just a special case of the corresponding result in (Dreveton et al., 2024)[Theorem 1]. Note that we have only one pair of communities $(C_+, C_-)$, which is responsible for determining the threshold. Exactly as in Dreveton et al. (2024), we require the limit $L(t) = \lim_{n \to \infty} \frac{n}{\log n} \mathrm{CH}_t(+, -)$ to exist and be strictly concave.

For the MAP's success, we note that the only change is that we relax the requirement in two ways: (i) we do not require $L(t)$ is strictly concave, (ii) we do not require that $L(t)$ always exists, and are fine when the limit does not exist while diverging to $+\infty$. Dreveton et al. prove the MAP success in two parts, by first bounding the probability of the event that an assignment vector $\sigma$ with hamming distance of $m$ has higher likelihood than the ground truth $\sigma^*$ (see their Lemma 3). In the second step (see their Appendix A.4), they do a union bound argument over all such possible vectors with the Hamming distance of $m$, and all varying $1 \le m \le n/2$.

We first note that that their Lemma 3 statement as well as its proof do not make use of the condition that $L(t)$ is strictly concave. In Appendix A.4, the union bound argument also does not require $L(t)$ is strictly concave (it only uses $\sup_{t \in (0,1)} L(t) > 1$), and the strict concavity was only required to show the impossibility direction. To justify the second relaxation where we allow the cases of infinite limit $L(t)$, we first note that their Lemma 3 is non-asymptotic anyway and derived for any finite $n$. Later, in Appendix A.4, we observe that the existence of the constants $\epsilon > 0$ and $\kappa > 0$ holds true, even when the limit $L(t) = +\infty$. In particular, the purpose behind the requirement was to exclude the cases when $L(t)$ does not converge but neither approaches $+\infty$.

## B    NOTATION AND PROBABILITY FACTS

### B.1    NOTATION

Throughout the paper, we extensively use the standard extended real line algebra on $\bar{\mathbb{R}} := \mathbb{R} \cup \{-\infty, +\infty\}$, where $+\infty$ and $-\infty$ are respectively greater and less than any other real number. Moreover, for any two $a, b \in \bar{\mathbb{R}}$, we will define $a - b = 0$ when $a = b$. We also view $\log : \bar{\mathbb{R}}_{\ge 0} \to \bar{\mathbb{R}}$ where $\log(0) = -\infty$ and $\log(+\infty) = +\infty$, and use $0 \log 0 = 0 \log(+\infty) = 0$, following the convention in the information-theory literature.

For any two vectors of random variables $X$ and $Y$, we write $X \perp Y$ if entries of one are independent from the others. For any real numbers $a, b \in R$, we denote $a \vee b = \max\{a, b\}$ and $a \wedge b = \min\{a, b\}$. Let $\mathrm{sgn} : \mathbb{R} \to \{\pm 1\}$ be the function defined by $\mathrm{sgn}(x) = 1$ if $x \ge 0$ and $\mathrm{sgn}(x) = -1$ if $x < 0$. We also extend the definition to vectors; let $\mathrm{sgn} : \mathbb{R}^n \to \{\pm 1\}^n$ be the map defined by applying the sign function componentwise. We define $\mathbb{R}_+ = [0, \infty)$. For $n \in \mathbb{N}$, we write $[n] = \{1, 2, \dots, n\}$. We use the standard notation $o(.), O(.), \omega(.), \Omega(.), \Theta(.)$ etc. throughout the paper. For non-negative sequences $(a_n)_{n \ge 1}$ and $(b_n)_{n \ge 1}$, we write $a_n \lesssim b_n$ to mean $a_n \le C b_n$ for some constant $C > 0$. The notation $\asymp$ is similar, hiding two constants in upper and lower bounds. Moreover, we denote $a_n \approx b_n$ as a shorthand for $\lim_{n \to \infty} \frac{a_n}{b_n} = 1$.

For a vector $x \in \mathbb{R}^n$, we define $\|x\|_2 = (\sum_{i=1}^n x_i^2)^{1/2}$, $\|x\|_1 = \sum_{i=1}^n |x_i|$, and $\|x\|_\infty = \max_i |x_i|$. Additionally, for any $i \in [n]$, we define $x_{-i}$ as the vector in $\mathbb{R}^n$ such that $(x_{-i})_j = x_j$ for $j \ne i$ and $(x_{-i})_i = 0$. For any matrix $M \in \mathbb{R}^{n \times n}$, $M_{i \cdot}$ refers to its $i^{\text{th}}$ row, which is a row vector, and $M_{\cdot i}$ refers to its $i^{\text{th}}$ column, which is a column vector. The matrix spectral norm is $\|M\|_2 = \sup_{\|x\|_2 = 1} \|Mx\|_2$, the matrix $2 \to \infty$ norm is $\|M\|_{2 \to \infty} = \sup_{\|x\|_2 = 1} \|Mx\|_\infty = \sup_i \|M_{i \cdot}\|_2$. Let $\mathrm{zd} : \mathbb{R}^{n \times n} \to \mathbb{R}^{n \times n}$ be the zero-diagonal mapping, where for any $A \in \mathbb{R}^{n \times n}$, $\mathrm{zd}(A)_{ij} = A_{ij}$ if $i \ne j$ and 0 otherwise.

## B.2 STANDARD PROBABILITY LEMMAS

**Lemma B.1** (Rényi Divergence Formula). *For any two probability laws $(\mathcal{A}, \mathcal{B})$, either both discrete or continuous, we have that*

$$e^{(t-1)D_t(\mathcal{A}\|\mathcal{B})} = \mathop{\mathbb{E}}_{x\sim\mathcal{B}}\left[\left(\frac{\mathcal{A}(x)}{\mathcal{B}(x)}\right)^t\right].$$

*Proof.* The proof is immediate by rearranging the terms from the definition in (4). $\quad\square$

**Lemma B.2.** *Let $Z \sim \mathcal{N}(0,1)$. The for any $t > 0$,*

$$\left(\frac{1}{t} - \frac{1}{t^3}\right)\frac{1}{\sqrt{2\pi}}e^{-t^2/2} \le \mathbb{P}(Z \ge t) \le \frac{1}{t\sqrt{2\pi}}e^{-t^2/2}.$$

We next show that the sampling procedure of the community assignment vector $\sigma^*$ in any of the models leads to communities with roughly $\rho$ and $(1 - \rho)$ fraction of vertices.

**Lemma B.3.** *Let $\sigma^* \in \{\pm 1\}^n$ be a vector whose coordinates are i.i.d. with $\mathbb{P}(\sigma_i^* = +1) = \rho$. Then, let $C_+ := \{i : \sigma_i^* = +1\}$ and $C_- := \{i : \sigma_i^* = -1\}$. Define the event $E$ as follow.*

$$E := \{||C_+| - \rho n| \le \rho n^{2/3} \quad \text{and} \quad ||C_-| - (1-\rho)n| \le \rho n^{2/3}\}. \tag{13}$$

*Then $\mathbb{P}(E) \ge 1 - o(1)$.*

*Proof.* For each $1 \le i \le n$, since $\mathbb{P}(\sigma_i^*) = \rho$ i.i.d., we have that $|C_+| = |\{i : \sigma_i^* = +1\}|$ follows $\text{Bin}(n, \rho)$. The Chernoff bound for binomial random variables (Mitzenmacher & Upfal, 2017, Theorem 4.4, Theorem 4.5) implies that for any $\delta \in (0,1)$,

$$\mathbb{P}(||C_+| - \mathbb{E}[|C_+|]| \le \delta n) \ge 1 - 2\exp\left(-\delta^2\mathbb{E}[|C_+|]/3\right).$$

Note that $\mathbb{E}[|C_+|] = \rho n$. Choosing $\delta = \rho n^{-1/3} = o(1)$ (as $\rho > 0$ is a constant)

$$\mathbb{P}(||C_+| - \rho n| \le \rho n^{2/3}) \ge 1 - 2\exp\left(-n^{-2/3}\rho^3 n/3\right) = 1 - 2\exp\left(-\rho^3 n^{1/3}/3\right).$$

Since $n^{1/3} = \omega(\log n)$ and $\rho > 0$ is a constant,

$$\mathbb{P}\left((1 - n^{-1/3})\rho n \le |C_+| \le \left(1 + n^{-1/3}\right)\rho n\right) \ge 1 - O(\exp(-10\log n/3)) = 1 - O(n^{-3}),$$

which implies the lemma. $\quad\square$

## C VERIFICATION FOR INFORMATION THEORETIC LIMITS

In this section, we will verify the function $L(t)$ is well defined and is strictly concave for both ROS and SBM and in the cases of no side information, or the special cases of GF, BEC and BSC side information channels. As a result, even $I^*$ is well-defined for these settings and sharply characterizes the information-theoretic limits in these settings, making our spectral algorithms in Theorem 1 and Theorem 2 optimal, when combined with the impossibility direction of Proposition 2.10. In order to verify these technical conditions, we will use a well-established closed form expression of Rényi divergence of two Gaussian random variables, e.g. see Gil et al. (2013).

**Lemma C.1.** *Let $\mathcal{A} \equiv \mathcal{N}(\mu_1, \sigma^2 I_d)$ and $\mathcal{B} \equiv \mathcal{N}(\mu_2, \sigma^2 I_d)$ for $\mu_1, \mu_2 \in \mathbb{R}^d$ and $\sigma^2 > 0$, then*

$$D_t(\mathcal{A}\|\mathcal{B}) = \frac{t}{2\sigma^2}\|\mu_1 - \mu_2\|_2^2.$$

**Side Information Terms.** We start by analyzing the additive terms that come in $L(t)$ due to the presence of each example of side information channels.

**Lemma C.2.** *Consider the side information laws $\mathcal{S}_+ \equiv \mathcal{N}(\upsilon_+, \sigma^2 I_d)$ and $\mathcal{S}_- \equiv \mathcal{N}(\upsilon_-, \sigma^2 I_d)$, for constant $\sigma^2 > 0$ and $\upsilon_+ = \boldsymbol{\beta}_+\sqrt{\log n}$ and $\upsilon_- = \boldsymbol{\beta}_-\sqrt{\log n}$ for $\boldsymbol{\beta}_+, \boldsymbol{\beta}_- \in \mathbb{R}^d$. Then*

$$\lim_{n\to\infty}\frac{n}{\log n} \cdot \frac{(1-t)}{n}D_t(\mathcal{S}_+\|\mathcal{S}_-) = \frac{t(1-t)\|\boldsymbol{\beta}_+ - \boldsymbol{\beta}_-\|_2^2}{2\sigma^2},$$

*which is well-defined. Note that this as a function of $t \in (0,1)$ is strictly concave when $\boldsymbol{\beta}_+ \ne \boldsymbol{\beta}_-$ or it is 0 otherwise.*

*Proof.* The lemma follows from a straightforward simplification and the use of Lemma C.1

$$\lim_{n\to\infty} \frac{n}{\log n} \cdot \frac{(1-t)}{n} D_t(\mathcal{S}_+\|\mathcal{S}_-) = \lim_{n\to\infty} \frac{(1-t)}{\log n} \frac{t\,\|v_+ - v_-\|_2^2}{2\sigma^2}$$

$$= \lim_{n\to\infty} \frac{t(1-t)}{\log n} \frac{\|\boldsymbol{\beta}_+ - \boldsymbol{\beta}_-\|_2^2 \log n}{2\sigma^2}$$

$$= \frac{t(1-t)\,\|\boldsymbol{\beta}_+ - \boldsymbol{\beta}_-\|_2^2}{2\sigma^2}.$$

$\square$

**Lemma C.3.** *Consider the side information laws* $(\mathcal{S}_+, \mathcal{S}_-)$ *as in the* BEC *channel with* $\epsilon$ *such that* $\lim_{n\to\infty} \frac{\log(1/\epsilon)}{\log n} = \beta$ *for* $\beta \geq 0$*, i.e.* $\epsilon = n^{-\beta+o(1)}$ *Then*

$$\lim_{n\to\infty} \frac{n}{\log n} \times \frac{(1-t)}{n} D_t(\mathcal{S}_+\|\mathcal{S}_-) = \beta.$$

*Proof.* By definition of Rényi divergence in (4)

$$\lim_{n\to\infty} \frac{n}{\log n} \cdot \frac{(1-t)}{n} D_t(\mathcal{S}_+\|\mathcal{S}_-) = \lim_{n\to\infty} \frac{(1-t)}{\log n} \frac{1}{(t-1)} \log \mathbb{E}_{y_i\sim\mathcal{S}_-}\left[\left(\frac{\mathcal{S}_+(y_i)}{\mathcal{S}_-(y_i)}\right)^t\right]$$

$$= \lim_{n\to\infty} -\frac{\log \epsilon}{\log n} = \lim_{n\to\infty} \frac{\log(1/\epsilon)}{\log n} = \beta.$$

$\square$

**Lemma C.4.** *Consider the side information laws* $(\mathcal{S}_+, \mathcal{S}_-)$ *as in the* BSC *channel with* $\alpha$ *such that* $\lim_{n\to\infty} \frac{\log(\frac{1-\alpha}{\alpha})}{\log n} = \beta$ *for* $\beta \geq 0$*, i.e.* $\alpha = n^{-\beta+o(1)}$ *Then*

$$\lim_{n\to\infty} \frac{n}{\log n} \times \frac{(1-t)}{n} D_t(\mathcal{S}_+\|\mathcal{S}_-) = \beta \min\{t, 1-t\}.$$

*Note that this is a concave function of* $t \in (0,1)$*.*

*Proof.* By definition of Rényi divergence in (4)

$$\lim_{n\to\infty} \frac{n}{\log n} \cdot \frac{(1-t)}{n} D_t(\mathcal{S}_+\|\mathcal{S}_-) = \lim_{n\to\infty} \frac{(1-t)}{\log n} \frac{1}{(t-1)} \log \mathbb{E}_{y_i\sim\mathcal{S}_-}\left[\left(\frac{\mathcal{S}_+(y_i)}{\mathcal{S}_-(y_i)}\right)^t\right]$$

$$= \lim_{n\to\infty} -\frac{\log(\alpha^t(1-\alpha)^{1-t} + \alpha^{1-t}(1-\alpha)^t)}{\log n}$$

$$= -\lim_{n\to\infty} \frac{-\min\{\beta t, \beta(1-t)\}\log n}{\log n} = \beta \min\{t, 1-t\}.$$

$\square$

**Deriving Threshold for** ROS **and** SBM   Finally, we will derive the expression of $L(t)$ and show that it is strictly concave.

**Lemma C.5.** *Fix* $\rho \in (0,1)$ *and* $a, b \in \mathbb{R}$ *such that* $\max\{|a|, |b|\} > 0$*. Consider the distribution* $(\mathcal{P}_+, \mathcal{P}_-, \mathcal{Q})$ *as defined in the* ROS *model (Definition 2.2). Then in the absence of side information,* $L(t)$ *is well-defined and given by*

$$L(t) = t(1-t)\frac{(a-b)^2(\rho a^2 + (1-\rho)b^2)}{2}.$$

*Moreover,* $L(t)$ *is strictly concave. Additionally, under* GF, BEC, BSC *channels described in Lemmas C.2, C.3, and C.4 respectively,* $L(t)$ *continues to be well-defined and strictly concave.*

*Proof.* In the absence of side information

$$L(t) = \lim_{n \to \infty} \frac{n}{\log n} \mathrm{CH}_t(+, -) = \lim_{n \to \infty} \frac{n(1-t)}{\log n} \left[ \rho \mathrm{D}_t(\mathcal{P}_+ \| \mathcal{Q}) + (1 - \rho) \mathrm{D}_t(\mathcal{Q} \| \mathcal{P}_-) \right]$$

$$= \lim_{n \to \infty} \frac{n(1-t)}{\log n} \left[ \rho \cdot \frac{t(a^2 - ab)^2}{2} \frac{\log n}{n} + (1 - \rho) \cdot \frac{t(ab - b^2)^2}{2} \frac{\log n}{n} \right]$$

$$\text{(using Lemma B.1)}$$

$$= t(1-t) \frac{\left( \rho(a^2 - ab)^2 + (1 - \rho)(ab - b^2)^2 \right)}{2} = t(1-t) \frac{(a - b)^2 (\rho a^2 + (1 - \rho)b^2)}{2}.$$

Note that when $\max\{|a|, |b|\} > 0$, the multiplicative coefficient of $t(1-t)$ is positive. Therefore, $L(t)$ is strictly concave function in $(0, 1)$.

Under side information, $L(t)$ is given by adding another term based on the channel as derived in Lemmas C.2, C.3, and C.4. Therefore, $L(t)$ continues to be well-defined. Moreover, $L(t)$ still remains strictly concave because the addition of a strictly concave function with another concave function (including just constant) is strictly concave. $\qquad\square$

**Lemma C.6.** *Fix $\rho \in (0, 1)$ and $a_1, a_2, b > 0$. Consider the distribution $(\mathcal{P}_+, \mathcal{P}_-, \mathcal{Q})$ as defined in the* SBM *model (Definition 2.3). Then in the absence of side information, $L(t)$ is well-defined and given by*

$$L(t) = t\rho a_1 + (1 - t)\rho b + t(1 - \rho)b + (1 - t)(1 - \rho)a_2 - \rho a_1^t b^{1-t} - (1 - \rho)b^t a_2^{1-t}.$$

*Moreover, except the degenerate case $a_1 = a_2 = b$, we have that $L(t)$ is strictly concave. Additionally, under* GF, BEC, BSC *channels described in Lemmas C.2, C.3, and C.4 respectively, $L(t)$ continues to be well-defined and strictly concave.*

*Proof.* This is just a special case of the expression derived by (Dreveton et al., 2024, Example 1) but for two communities. Note that $L(t)$ is twice differentiable, and

$$L''(t) = -\rho \left( \frac{a_1}{b} \right)^t b \log^2 \left( \frac{a_1}{b} \right) - (1 - \rho) \left( \frac{b}{a_2} \right)^t a_2 \log^2 \left( \frac{b}{a_2} \right) < 0,$$

unless $a_1 = a_2 = b$, and thus, $L(t)$ is strictly concave. Similar to Lemma C.5, even for SBM, after adding another side information channel-specific term to $L(t)$, derived in Lemmas C.2, C.3, and C.4, the function $L(t)$ is well-defined and strictly concave. $\qquad\square$

# D OMITTED PROOFS FROM SECTION 4

In this section, we prove all the genie-aided estimation related lemmas. We start with the proof of Lemma 4.1, which establishes that the failure of some genie-aided estimator implies that the global MAP estimator also fails.

*Proof of Lemma 4.1.* We first consider the case when side information $y$ (either BEC or BSC) is provided. Let $\mathcal{S}_1 = \{(\overline{A}, \overline{\sigma}, \overline{y}) : \exists i \in [n] \text{ such that } \hat{\sigma}_{\mathrm{Gen}, i} \text{ fails}\}$. Similarly, $\mathcal{S}_2 = \{(\overline{A}, \overline{\sigma}, \overline{y}) : \hat{\sigma}_{\mathrm{MAP}} \text{ fails}\}$. To show the desired claim, it suffices to show that $\mathcal{S}_1 \subseteq \mathcal{S}_2$.

To this end, fix any instance $(\overline{A}, \overline{\sigma}, \overline{y}) \in \mathcal{S}_1$ of $(A, \sigma^*, y)$. Then by definition of the failure of the genie-aided estimators give below Equation (7), there exists $i \in [n]$ such that

$$\mathbb{P}(\sigma_i^* = -\overline{\sigma}_i \mid \overline{A}, \overline{y}, \overline{\sigma}_{-i}) > \mathbb{P}(\sigma_i^* = \overline{\sigma}_i \mid \overline{A}, \overline{y}, \overline{\sigma}_{-i}). \tag{14}$$

We now consider the community assignment vector $\sigma' \in \{\pm 1\}^n$ whose labeling agrees with $\overline{\sigma}$ except for the $i^{\mathrm{th}}$ label, for which $\sigma_i' = -\overline{\sigma}_i$. Then

$$\mathbb{P}(\sigma^* = \sigma' \mid \overline{A}, \overline{y}) = \mathbb{P}(\sigma_{-i}^* = \sigma_{-i}', \sigma_i^* = \sigma_i' \mid \overline{A}, \overline{y})$$

$$= \mathbb{P}(\sigma_{-i}^* = \sigma_{-i}' \mid \overline{A}, \overline{y}) \cdot \mathbb{P}(\sigma_i^* = \sigma_i' \mid \overline{A}, \overline{y}, \sigma_{-i}^* = \sigma_{-i}')$$

$$= \mathbb{P}(\sigma_{-i}^* = \overline{\sigma}_{-i} \mid \overline{A}, \overline{y}) \cdot \mathbb{P}(\sigma_i^* = -\overline{\sigma}_i \mid \overline{A}, \overline{y}, \sigma_{-i}^* = \overline{\sigma}_{-i})$$

$$\text{(as } \sigma_{-i}' = \overline{\sigma}_{-i} \text{ but } \sigma_i' = -\overline{\sigma}_i)$$

$$> \mathbb{P}(\sigma_{-i}^* = \overline{\sigma}_{-i} \mid A, y) \cdot \mathbb{P}(\sigma^* = \overline{\sigma}_i \mid \overline{A}, \overline{y}, \sigma_{-i}^* = \overline{\sigma}_{-i}) \quad \text{(using (14))}$$

$$= \mathbb{P}(\sigma_{-i}^* = \overline{\sigma}_{-i}, \sigma_i^* = \overline{\sigma}_i \mid \overline{A}, \overline{y})$$

$$= \mathbb{P}(\sigma^* = \overline{\sigma} \mid \overline{A}, \overline{y}). \tag{15}$$

By the definition of the MAP estimator $\hat{\sigma}_{\text{MAP}} = \arg\max_{\sigma \in \{\pm 1\}^n} \mathbb{P}(\sigma^* = \sigma | \overline{A}, \overline{y}) \neq \overline{\sigma}$, which implies $(\overline{A}, \overline{\sigma}, \overline{y}) \in \mathcal{S}_2$.

Finally, we consider the case when there is no side information. Define $\mathcal{S}_1$ and $\mathcal{S}_2$ similarly, but after dropping $y$.

- Case $\mathcal{P}_+ \not\equiv \mathcal{P}_-$ or $\rho \neq 1/2$. Consider any $(\overline{A}, \overline{\sigma}) \in \mathcal{S}_1$ and follow exactly the same argument used in in deriving (15) (after dropping the conditioning on $y$). This will lead to the conclusion that $(\overline{A}, \overline{\sigma}) \in \mathcal{S}_2$, yielding $\mathcal{S}_1 \subseteq \mathcal{S}_2$.

- Case $\mathcal{P}_+ \equiv \mathcal{P}_-$ and $\rho = 1/2$. Consider any $(\overline{A}, \overline{\sigma}) \in \mathcal{S}_1$. Follow exactly the same argument as in (15) and conclude that $\mathbb{P}(\sigma^* = \sigma' \mid \overline{A}) > \mathbb{P}(\sigma^* = \overline{\sigma} \mid \overline{A})$. Additionally, due to the symmetry, $\mathbb{P}(\sigma^* = \overline{\sigma} \mid \overline{A}) = \mathbb{P}(\sigma^* = -\overline{\sigma} | \overline{A})$. Combining these two, we obtain that $\hat{\sigma}_{\text{MAP}} \notin \{\pm \sigma^*\}$, which implies $\hat{\sigma}_{\text{MAP}}$ fails by Definition 2.8. Conclude $(\overline{A}, \overline{\sigma}) \in \mathcal{S}_2$, as desired.

$\square$

We now derive the expressions for genie scores given by Lemma 4.2, without or with side information.

*Proof of Lemma 4.2.* We first recall the definition of genie scores from (8). For any $i \in [n]$, we do the following analysis in each model of side information.

No side information:

$$z_i^* = \log\left(\frac{\mathbb{P}(\sigma_i^* = +1 \mid A, \sigma_{-i}^*)}{\mathbb{P}(\sigma_i^* = -1 \mid A, \sigma_{-i}^*)}\right) = \log\left(\frac{\mathbb{P}(\sigma_i^* = +1) \cdot \mathcal{L}(A, \sigma_{-i}^* \mid \sigma_i^* = +1)}{\mathbb{P}(\sigma_i^* = -1) \cdot \mathcal{L}(A, \sigma_{-i}^* \mid \sigma_i^* = -1)}\right)$$

$$= \log\left(\frac{\mathbb{P}(\sigma_i^* = +1) \cdot \mathcal{L}(A \mid \sigma_i^* = +1, \sigma_{-i}^*)}{\mathbb{P}(\sigma_i^* = -1) \cdot \mathcal{L}(A \mid \sigma_i^* = -1, \sigma_{-i}^*)}\right) \qquad (\sigma_{-i}^* \text{ is independent of } \sigma_i^*)$$

$$= \log\left(\frac{\mathbb{P}(\sigma_i^* = +1) \cdot \mathcal{L}(A_{i\cdot} \mid \sigma_i^* = +1, \sigma_{-i}^*)}{\mathbb{P}(\sigma_i^* = -1) \cdot \mathcal{L}(A_{i\cdot} \mid \sigma_i^* = -1, \sigma_{-i}^*)}\right)$$

(the likelihood of all but the $i^{\text{th}}$ row is same conditioned under $\sigma_{-i}^*$ irrespective of $\sigma_i^*$)

$$= \log\left(\rho \cdot \prod_{j \in C_+^{-i}} \mathcal{P}_+(A_{ij}) \cdot \prod_{j \in C_-^{-i}} \mathcal{Q}(A_{ij}) \Big/ (1-\rho) \cdot \prod_{j \in C_+^{-i}} \mathcal{Q}(A_{ij}) \cdot \prod_{j \in C_-^{-i}} \mathcal{P}_-(A_{ij})\right)$$

(due to the conditional independence of the entries and the law of GBM)

$$= \sum_{j \in C_+^{-i}} \log\left(\frac{\mathcal{P}_+(A_{ij})}{\mathcal{Q}(A_{ij})}\right) + \sum_{j \in C_-^{-i}} \log\left(\frac{\mathcal{Q}(A_{ij})}{\mathcal{P}_-(A_{ij})}\right) + \log\left(\frac{\rho}{1-\rho}\right).$$

Side information: Again by (8), we have

$$z_i^* = \log\left(\frac{\mathbb{P}(\sigma_i^* = +1 \mid A, y, \sigma_{-i}^*)}{\mathbb{P}(\sigma_i^* = -1 \mid A, y, \sigma_{-i}^*)}\right) = \log\left(\frac{\mathbb{P}(\sigma_i^* = +1 \mid A, y_i, \sigma_{-i}^*)}{\mathbb{P}(\sigma_i^* = -1 \mid A, y_i, \sigma_{-i}^*)}\right)$$

$$\text{(conditioned on } \sigma_{-i}^*, \text{ we have } \sigma_i^* \perp y_{-i})$$

$$= \log\left(\frac{\mathbb{P}(\sigma_i^* = +1) \cdot \mathcal{L}(A, y_i, \sigma_{-i}^* \mid \sigma_i^* = +1)}{\mathbb{P}(\sigma_i^* = -1) \cdot \mathcal{L}(A, y_i, \sigma_{-i}^* \mid \sigma_i^* = -1)}\right)$$

$$= \log\left(\frac{\mathbb{P}(\sigma_i^* = +1) \cdot \mathcal{L}(A, \sigma_{-i}^* \mid \sigma_i^* = +1) \cdot \mathbb{P}(y_i \mid \sigma_i^* = +1)}{\mathbb{P}(\sigma_i^* = -1) \cdot \mathcal{L}(A, \sigma_{-i}^* \mid \sigma_i^* = -1) \cdot \mathbb{P}(y_i \mid \sigma_i^* = -1)}\right)$$

$$\text{(conditioned on } \sigma_i^*, \text{ we have } y_i \perp A \text{ and } y_i \perp \sigma_{-i}^*)$$

$$= \log\left(\frac{\mathbb{P}(\sigma_i^* = +1 \mid A, \sigma_{-i}^*) \cdot \mathbb{P}(y_i \mid \sigma_i^* = +1)}{\mathbb{P}(\sigma_i^* = -1 \mid A, \sigma_{-i}^*) \cdot \mathbb{P}(y_i \mid \sigma_i^* = -1)}\right)$$

$$= \log\left(\frac{\mathbb{P}(\sigma_i^* = +1 \mid A, \sigma_{-i}^*)}{\mathbb{P}(\sigma_i^* = -1 \mid A, \sigma_{-i}^*)}\right) + \log\left(\frac{\mathbb{P}(y_i \mid \sigma_i^* = +1)}{\mathbb{P}(y_i \mid \sigma_i^* = -1)}\right) \qquad (16)$$

Note that

$$\log\left(\frac{\mathbb{P}(y_i \mid \sigma_i^* = +1)}{\mathbb{P}(y_i \mid \sigma_i^* = -1)}\right) = \log\left(\frac{\mathcal{S}_+(y_i)}{\mathcal{S}_-(y_i)}\right).$$

Substituting this in (16) along with the definition of genie score without side information (8) and using the expression from the case without side information, we obtain

$$z_i^* = \sum_{j \in C_+^{-i}} \log\left(\frac{\mathcal{P}_+(A_{ij})}{\mathcal{Q}(A_{ij})}\right) + \sum_{j \in C_-^{-i}} \log\left(\frac{\mathcal{Q}(A_{ij})}{\mathcal{P}_-(A_{ij})}\right) + \log\left(\frac{\rho}{1-\rho}\right) + \log\left(\frac{\mathcal{S}_+(y_i)}{\mathcal{S}_-(y_i)}\right).$$

We now show the proof of Lemma 4.3, where we explicitly derive the closed form of the additive factor under side information for the special cases of Gaussian Features (GF), Binary Erasure Channel (BEC), and Binary Symmetric Channel (BSC).

*Proof of Lemma 4.3.* We break into cases.

- Gaussian Features (GF):

$$\log\left(\frac{\mathcal{S}_+(y_i)}{\mathcal{S}_-(y_i)}\right) = \log\left(\frac{e^{-\frac{\|y_i - v_+\|_2^2}{2\sigma^2}}}{e^{-\frac{\|y_i - v_-\|_2^2}{2\sigma^2}}}\right) = \log\left[\exp\left(\frac{\|y_i - v_-\|_2^2 - \|y_i - v_+\|_2^2}{2\sigma^2}\right)\right]$$

$$= \frac{\|y_i - v_-\|_2^2 - \|y_i - v_+\|_2^2}{2\sigma^2}.$$

- Binary Erasure Channel (BEC): If $y_i = 0$, then $\mathcal{S}_+(y_i) = \mathcal{S}_-(y_i) = \epsilon$. If $y_i = +1$, then $\mathcal{S}_+(y_i) = 1 - \epsilon$ but $\mathcal{S}_-(y_i) = 0$, and similarly, if $y_i = -1$, then $\mathcal{S}_+(y_i) = 0$ but $\mathcal{S}_-(y_i) = 1 - \epsilon$.

$$\log\left(\frac{\mathcal{S}_+(y_i)}{\mathcal{S}_-(y_i)}\right) = \begin{cases} +\infty, & \text{if } \sigma_i^* = +1; \\ -\infty, & \text{if } \sigma_i^* = -1; \\ 0, & \text{otherwise.} \end{cases}$$

- Binary Symmetric Channel (BSC):

$$\log\left(\frac{\mathcal{S}_+(y_i)}{\mathcal{S}_-(y_i)}\right) = \log\left(\frac{1-\alpha}{\alpha}\right)\mathbf{1}[y_i = +1] + \log\left(\frac{\alpha}{1-\alpha}\right)\mathbf{1}[y_i = -1] = \log\left(\frac{1-\alpha}{\alpha}\right)y_i.$$

$\square$

$\square$

## D.1   GENIE SUCCESS WITH MARGIN ABOVE THE IT THRESHOLD.

We now give a formal version of the claim that above the information-theoretic limit, all the genie scores succeed with a margin of $\Omega(\log n)$, formalized in the following lemma.

**Lemma D.1.** *Fix $\rho \in (0,1)$ and consider probability laws $(\mathcal{P}_+, \mathcal{P}_-, \mathcal{Q})$. Let $(A, \sigma^*) \sim \mathsf{GBM}_n(\rho, \mathcal{P}_+, \mathcal{P}_-, \mathcal{Q})$ and we observe A. Condition on $\sigma^*$ such that the event E from (13) holds. Optionally, for side information laws $(\mathcal{S}_+, \mathcal{S}_-)$, we observe $y \sim \mathsf{SI}(\sigma^*, \mathcal{S}_+, \mathcal{S}_-)$. Let $z^*$ be the genie score vector for the corresponding model given by (8). Let $I^*$ be defined according to (6). Then if $I^* > 1$ then there exists some constant $\delta > 0$ such that for any $i \in [n]$:*

$$\mathbb{P}(\sigma_i^* z_i^* < \delta \log n) = o(n^{-1}).$$

*Proof.* By Lemma 4.2, for any $i \in [n]$, the genie score $z_i^*$ is given by

$$z_i^* = \sum_{j \in C_+^{-i}} \log\left(\frac{\mathcal{P}_+(A_{ij})}{\mathcal{Q}(A_{ij})}\right) + \sum_{j \in C_-^{-i}} \log\left(\frac{\mathcal{Q}(A_{ij})}{\mathcal{P}_-(A_{ij})}\right) + \log\left(\frac{\rho}{1-\rho}\right) + \log\left(\frac{\mathcal{S}_+(y_i)}{\mathcal{S}_-(y_i)}\right).$$

For convenience, we define $X_i := z_i^* - \log\left(\frac{\rho}{1-\rho}\right)$. For any $i \in C_+$ and any $\varepsilon, t > 0$,

$$\mathbb{P}(\sigma_i^* z_i^* < \varepsilon \log n) = \mathbb{P}(z_i^* < \varepsilon \log n) = \mathbb{P}(X_i < (1 + o(1))\varepsilon \log n) = \mathbb{P}\left(e^{tX_i} < e^{t(1+o(1))\varepsilon \log n}\right)$$

$$\leq e^{t\varepsilon \log n}\, \mathbb{E}\left[e^{-tX_i}\right].$$

We now analyze

$$
\begin{aligned}
\mathbb{E}\left[e^{-tX_i}\right] &= \mathbb{E}\left[\exp\left(-t\left(\sum_{j\in C_+^{-i}}\log\left(\frac{\mathcal{P}_+(A_{ij})}{\mathcal{Q}(A_{ij})}\right) + \sum_{j\in C_-^{-i}}\log\left(\frac{\mathcal{Q}(A_{ij})}{\mathcal{P}_-(A_{ij})}\right) + \log\left(\frac{\mathcal{S}_+(y_i)}{\mathcal{S}_-(y_i)}\right)\right)\right)\right] \\
&= \mathbb{E}\left[\exp\left(t\left(\sum_{j\in C_+^{-i}}\log\left(\frac{\mathcal{Q}(A_{ij})}{\mathcal{P}_+(A_{ij})}\right) + \sum_{j\in C_-^{-i}}\log\left(\frac{\mathcal{P}_-(A_{ij})}{\mathcal{Q}(A_{ij})}\right) + \log\left(\frac{\mathcal{S}_-(y_i)}{\mathcal{S}_+(y_i)}\right)\right)\right)\right] \\
&= \mathbb{E}\left[\prod_{j\in C_+^{-i}} e^{t\log\left(\frac{\mathcal{Q}(A_{ij})}{\mathcal{P}_+(A_{ij})}\right)} \prod_{j\in C_-^{-i}} e^{t\log\left(\frac{\mathcal{P}_-(A_{ij})}{\mathcal{Q}(A_{ij})}\right)} \cdot e^{t\log\left(\frac{\mathcal{S}_-(y_i)}{\mathcal{S}_+(y_i)}\right)}\right] \\
&= \prod_{j\in C_+^{-i}} \mathbb{E}\left[\left(\frac{\mathcal{Q}(A_{ij})}{\mathcal{P}_+(A_{ij})}\right)^t\right] \prod_{j\in C_-^{-i}} \mathbb{E}\left[\left(\frac{\mathcal{P}_-(A_{ij})}{\mathcal{Q}(A_{ij})}\right)^t\right] \mathbb{E}\left[\left(\frac{\mathcal{S}_-(y_i)}{\mathcal{S}_+(y_i)}\right)^t\right] \\
&= e^{(1+o(1))(t-1)(\rho n\mathrm{D}_t(\mathcal{Q}\|\mathcal{P}_+)+(1-\rho)n\mathrm{D}_t(\mathcal{P}_-\|\mathcal{Q})+\mathrm{D}_t(\mathcal{S}_-\|\mathcal{S}_+))} \\
&\qquad\qquad\qquad \text{(using Lemma B.1 and using community sizes conditioned on } E) \\
&= e^{-(1+o(1))tn\left(\rho\mathrm{D}_{1-t}(\mathcal{P}_+\|\mathcal{Q})+(1-\rho)\mathrm{D}_{1-t}(\mathcal{Q}\|\mathcal{P}_+)+\frac{1}{n}\mathrm{D}_{1-t}(\mathcal{S}_+\|\mathcal{S}_-)\right)} \\
&\qquad\qquad\qquad \text{(since } (t-1)\mathcal{D}_t(\mathcal{A}\|\mathcal{B}) = -t\mathcal{D}_{1-t}(\mathcal{B}\|\mathcal{A})) \\
&= e^{-(1+o(1))n\mathrm{CH}_{1-t}(+,-)}
\end{aligned}
$$

Substituting this in our Chernoff-style bound, we obtain for any $i\in C_+$

$$
\mathbb{P}(\sigma_i^* z_i^* < \varepsilon\log n) \leq e^{-(1+o(1))(n\mathrm{CH}_{1-t}(+,-)-t\varepsilon\log n)}
$$

When $I^* > 1$, we have there exists $t^* \in (0,1)$ such that $L(1-t^*) = \lim_{n\to\infty}\frac{n}{\log n}\mathrm{CH}_{1-t^*}(+,-) > 1$. Thus, there exists a constant $\epsilon > 0$, and $t^* \in (0,1)$ such that for sufficient large $n$:

$$
\mathbb{P}(\sigma_i^* z_i^* < \varepsilon\log n) \leq e^{-((1+\epsilon)\log n - t^*\varepsilon\log n)}.
$$

Therefore, one can choose $\delta_1 := \delta_1(t^*,\epsilon) > 0$ small enough such that

$$
\mathbb{P}(\sigma_i^* z_i^* < \delta_1\log n) \leq e^{-((1+\epsilon/2)\log n)} = o(n^{-1}).
$$

We carry out exactly similar calculation for the community $C_-$. For any $i\in C_-$ and $t,\varepsilon > 0$,

$$
\mathbb{P}(\sigma_i^* z_i^* < \varepsilon\log n) = \mathbb{P}(z_i^* > -\varepsilon\log n) = \mathbb{P}\left(X_i > -(1+o(1))\varepsilon\log n\right) = \mathbb{P}\left(e^{tX_i} > e^{-t(1+o(1))\varepsilon\log n}\right)
$$
$$
\leq e^{t\varepsilon\log n}\,\mathbb{E}\left[e^{tX_i}\right].
$$

Simplifying

$$
\begin{aligned}
\mathbb{E}\left[e^{tX_i}\right] &= \mathbb{E}\left[\exp\left(t\left(\sum_{j\in C_+^{-i}}\log\left(\frac{\mathcal{P}_+(A_{ij})}{\mathcal{Q}(A_{ij})}\right) + \sum_{j\in C_-^{-i}}\log\left(\frac{\mathcal{Q}(A_{ij})}{\mathcal{P}_-(A_{ij})}\right) + \log\left(\frac{\mathcal{S}_+(y_i)}{\mathcal{S}_-(y_i)}\right)\right)\right)\right] \\
&= \prod_{j\in C_+^{-i}} \mathbb{E}\left[\left(\frac{\mathcal{P}_+(A_{ij})}{\mathcal{Q}(A_{ij})}\right)^t\right] \prod_{j\in C_-^{-i}} \mathbb{E}\left[\left(\frac{\mathcal{Q}(A_{ij})}{\mathcal{P}_-(A_{ij})}\right)^t\right] \mathbb{E}\left[\left(\frac{\mathcal{S}_+(y_i)}{\mathcal{S}_-(y_i)}\right)^t\right] \\
&= e^{(1+o(1))(t-1)(\rho n\mathrm{D}_t(\mathcal{P}_+\|\mathcal{Q})+(1-\rho)n\mathrm{D}_t(\mathcal{Q}\|\mathcal{P}_-)+\mathrm{D}_t(\mathcal{S}_+\|\mathcal{S}_-))} \\
&\qquad\qquad\qquad \text{(using Lemma B.1 and using community sizes conditioned on } E) \\
&= e^{-(1+o(1))n\mathrm{CH}_t(+,-)}
\end{aligned}
$$

Overall, we obtain for any $i\in C_+$

$$
\mathbb{P}(\sigma_i^* z_i^* < \varepsilon\log n) \leq e^{-(1+o(1))(n\mathrm{CH}_t(+,-)-t\varepsilon\log n)}
$$

If $I^* > 1$, then there exists $t^* \in (0,1)$ such that $L(t^*) = \lim_{n \to \infty} \frac{n}{\log n} \mathrm{CH}_t(+, -) > 1$ and thus, there exists a constant $\epsilon > 0$, and $t^* \in (0,1)$ such that for sufficient large $n$:

$$\mathbb{P}(\sigma_i^* z_i^* < \varepsilon \log n) \leq e^{-((1+\epsilon) \log n - t^* \varepsilon \log n)}.$$

Therefore, choosing $\delta_2 := \delta_2(t^*, \epsilon) > 0$ small enough such that

$$\mathbb{P}(\sigma_i^* z_i^* < \delta_2 \log n) \leq e^{-((1+\epsilon/2) \log n)} = o(n^{-1}).$$

Finally, choosing $\delta = \min\{\delta_1, \delta_2\} > 0$, we obtain for $i \in [n]$

$$\mathbb{P}(\sigma_i^* z_i^* < \delta \log n) = o(n^{-1}),$$

concluding the proof of the lemma. $\qquad\square$

## E  ENTRYWISE BEHAVIOR OF EIGENVECTORS.

Abbe, Fan, Wang, and Zhong Abbe et al. (2020) showed the powerful entrywise behavior of eigenvectors for a general ensemble of random matrices under certain assumptions. Their result (Abbe et al., 2020, Theorem 2.1) applies more generally to eigenspaces; below we note a special case of their result when the eigenspace has a single eigenvalue.

Suppose $A \in \mathbb{R}^{n \times n}$ is a symmetric random matrix and $A^* = \mathbb{E}[A]$. Let the eigenvalues of $A$ be $|\lambda_1| \geq \cdots \geq |\lambda_n|$, and their associated eigenvectors be $\{u_j\}_{j \in [n]}$ (defined up to rotation if eigenvalues are repeated). Analogously for $A^*$, the eigenvalues and eigenvectors are $|\lambda_1^*| \geq \cdots \geq |\lambda_n^*|$ and $\{u_j^*\}_{j \in [n]}$, respectively. For any fixed $(\lambda_i^*, u_i^*)$, define the eigengap quantity

$$\Delta^* := |\lambda_i^*| \wedge \min_{j \in [n] \setminus \{i\}} |\lambda_i^* - \lambda_j^*|. \tag{17}$$

Here we define the eigengap for the special case of (Abbe et al., 2020, Theorem 2.1) applied to a single eigenvector, rather than for an eigenspace associated with consecutive eigenvalues. For more general definition when the eigenspace contains multiple eigenvalues, see (Abbe et al., 2020, Equation (2.1)). We define $\kappa := |\lambda_i^*|/\Delta^*$, which is always bounded from below by 1. For a parameter $\gamma \geq 0$, consider the following four assumptions.

**A1** (Incoherence). $\|A^*\|_{2 \to \infty} \leq \gamma \Delta^*$.

**A2** (Row- and column-wise independence). *For any $m \in [n]$, the entries in the $m^{\mathrm{th}}$ row and column of $A$ are independent from others, i.e. $\{A_{ij} : i = m \text{ or } j = m\} \perp \{A_{ij} : i \neq m, j \neq m\}$.*

**A3** (Spectral norm concentration). *For some $\delta_0 \in (0,1)$, suppose $\mathbb{P}(\|A - A^*\|_2 \leq \gamma \Delta^*) \geq 1 - \delta_0$.*

**A4** (Row concentration). *Suppose $\varphi(x)$ is continuous and non-decreasing in $\mathbb{R}_+$ and $\varphi(x)/x$ is non-increasing for $x > 0$. Additionally $\varphi(0) = 0$ and $32\kappa \max\{\gamma, \varphi(\gamma)\} \leq 1$. Let there be some $\delta_1 \in (0,1)$ such that for any $m \in [n]$ and $w \in \mathbb{R}^n$*

$$\mathbb{P}\left( |(A - A^*)_{m \cdot} w| \leq \Delta^* \|w\|_\infty \varphi\left( \frac{\|w\|_2}{\sqrt{n} \|w\|_\infty} \right) \right) \geq 1 - \frac{\delta_1}{n}.$$

**Lemma E.1** (Theorem 2.1 Abbe et al. (2020)). *Under Assumptions 1 to 4, with probability at least $1 - \delta_0 - 2\delta_1$, we have*

$$\min_{s \in \{\pm 1\}} \left\| su_i - \frac{Au_i^*}{\lambda_i^*} \right\|_\infty \lesssim \kappa(\kappa + \varphi(1))(\gamma + \varphi(\gamma)) \|u_i^*\|_\infty + \frac{\gamma \|A^*\|_{2 \to \infty}}{\Delta^*}.$$

### E.1  ENTRYWISE ANALYSIS FOR EIGENVECTORS FOR ROS.

In this subsection, we will show that the top eigenvector of $A$ sampled from ROS exhibits the entrywise behavior discussed above. More formally, we show the following lemma.

**Lemma E.2.** *Fix $\rho \in (0,1)$ and $a, b \in \mathbb{R}$ such that $\max\{|a|, |b|\} > 0$. Let $(A, \sigma^*) \sim \mathsf{ROS}_n(\rho, a, b)$. Condition on $\sigma^*$ satisfying $E$ from (13). Let $A^* := \mathbb{E}[A \mid \sigma^*]$. Define $(\lambda_1, u_1)$ and $(\lambda_1^*, u_1^*)$ as above. Then with probability $1 - o(1)$*

$$\min_{s \in \{\pm 1\}} \left\| su_1 - \frac{Au_1^*}{\lambda_1^*} \right\|_\infty \leq \frac{C}{\sqrt{n \log n}},$$

*for some constant $C := C(\rho, a, b) > 0$.*

According to the definition of the ROS model, we have $A = \mathsf{zd}\left(\sqrt{\frac{\log n}{n}}\, v^* v^{*\top} + W\right)$. The entire analysis is done conditioned on $\sigma^*$, so the only randomness in this analysis is from the added noise matrix $W$. We verify Assumptions 1-4 required to apply Lemma E.1 using similar ideas as (Abbe et al., 2020, Theorem 3.1).

First, observe that $A^* = \mathsf{zd}(v^* v^{*\top} \sqrt{\log n/n})$. Let $(\lambda_1^*, u_1^*)$ be the top eigenpair. The corresponding eigengap quantity defined in (17) is $\Delta^* := |\lambda_1^*| \wedge \min_{2 \le i \le n} |\lambda_1^* - \lambda_i^*|$. We begin by characterizing $u_1^*$, $\lambda_1^*$, and $\Delta^*$.

**Lemma E.3.** *Let $(\lambda_1^*, u_1^*)$ be the top eigenpair of $A^*$. Then*

$$u_1^* = \frac{(1 + o(1))v^*}{\|v^*\|_2} \quad \lambda_1^* = (1 + o(1))\sqrt{\frac{\log n}{n}}\,\|v^*\|_2^2, \quad \Delta^* \approx |\lambda_1^*| = \Theta(\sqrt{n \log n}).$$

*Proof.* Note that $v^* v^{*\top} \sqrt{\log n/n}$ is a rank-1 matrix. Let $|\tilde{\lambda}_1| \ge \cdots \ge |\tilde{\lambda}_n|$ be its eigenvalues. Then we have that only non-zero eigenvalue is $\tilde{\lambda}_1 = \sqrt{\frac{\log n}{n}}\,\|v^*\|_2^2 = \Theta(\sqrt{n \log n})$ and the corresponding normalized eigenvector is $v^*/\|v^*\|_2$ and $\tilde{\lambda}_2 = \cdots = \tilde{\lambda}_n = 0$. After zeroing out the diagonal, the entries of the corresponding eigenvector $v^*/\|v^*\|_2$ will be perturbed by a factor of $(1 + o(1))$ since the diagonal correction is of the order of $O(\sqrt{\log n/n})$. Hence, we obtain $u_1^* = (1 + o(1))v^*/\|v^*\|_2$. By Weyl's inequality, we calculate the effect of zeroing out the diagonal on the eigenvalue:

$$|\lambda_1^* - \tilde{\lambda}_1| \le \left\| v^* v^{*\top} \sqrt{\log n/n} - \mathsf{zd}\left(v^* v^{*\top} \sqrt{\log n/n}\right) \right\|_2 = O\left(\sqrt{\frac{\log n}{n}}\right).$$

Therefore,

$$\lambda_1^* = \tilde{\lambda}_1 + O(\sqrt{\log n/n}) = \sqrt{\frac{\log n}{n}}\,\|v^*\|_2^2 + O(\sqrt{\log n/n}) = (1 + o(1))\sqrt{\frac{\log n}{n}}\,\|v^*\|_2^2 \asymp \sqrt{n \log n}.$$

Applying Weyl's inequality, for $2 \le i \le n$, we get $|\lambda_i^*| = O(\sqrt{\log n/n})$. Hence, $\Delta^* \approx |\lambda_1^*| \asymp \sqrt{n \log n}$. $\square$

*Proof of Lemma E.2.* We will let $\gamma := \frac{3\sqrt{n}}{\Delta^*} = 1/\Theta(\sqrt{\log n})$, due to Lemma E.3. Let us now verify Assumption 1. For any $i \in C_+$

$$\|A_{i\cdot}^*\|_2 = \sqrt{|C_+^{-i}| \cdot a^4 \frac{\log n}{n} + |C_-| \cdot a^2 b^2 \frac{\log n}{n}}$$

$$= \sqrt{(1 + o(1))\rho n \cdot a^4 \frac{\log n}{n} + (1 + o(1))(1 - \rho)n \cdot a^2 b^2 \frac{\log n}{n}} = \Theta(\sqrt{\log n}),$$

where the second step follows from using Lemma B.3. Similarly, also for any $i \in C_-$

$$\|A_{i\cdot}^*\|_2 = \sqrt{|C_+| \cdot a^2 b^2 \frac{\log n}{n} + |C_-^{-i}| \cdot b^4 \frac{\log n}{n}} = \Theta(\sqrt{\log n}).$$

Overall, combining these two we obtain $\|A^*\|_{2 \to \infty} = \Theta(\sqrt{\log n}) \le 3\sqrt{n} = \gamma \Delta^*$, verifying Assumption 1. Assumption 2 on row and column-wise independence trivially holds due to the i.i.d. noise matrix $W$ (up to symmetry).

To verify Assumption 3 on spectral norm concentration, first observe that $A - A^* = W$, where $W$ is the zero diagonal symmetric matrix with i.i.d. $\mathcal{N}(0, 1)$ entries. Applying (Bandeira et al., 2017, Proposition 3.3), we have that with probability at least $1 - e^{-n/2}$,

$$\|A - A^*\|_2 = \|W\|_2 \le 3\sqrt{n} = \gamma \Delta^*,$$

Therefore, Assumption 3 holds with $\delta_0 = e^{-n/2}$. We now turn our attention to Assumption 4. Let us choose $\varphi(x) = cx$ for some constant $c > 0$ which we will decide later. Clearly, $\varphi$ is continuous, non-decreasing in $\mathbb{R}_+$ with $\varphi(0) = 0$, and $\varphi(x)/x = c$ is also non-increasing in $(0, \infty)$. Letting $\kappa = 1$, it is straightforward to see that $32\kappa \max\{\gamma, \varphi(\gamma)\} = 32\gamma \max\{1, c\} = o(1) \le 1$, as $\gamma = o(1)$.

We now verify the row concentration part of the assumption. Using Lemma E.3, we have $\Delta^* \approx |\lambda_1^*| \geq \max\{\rho a^2, (1-\rho)b^2\}\sqrt{n \log n}$. Therefore, it holds that for any $\epsilon > 0$, there is a sufficiently large $n$ such that $\Delta^* \geq (1-\epsilon)\max\{\rho a^2, (1-\rho)b^2\}\sqrt{n \log n}$. Moreover, for any fixed $w \in \mathbb{R}^n$, one can say that

$$\Delta^* \|w\|_\infty \, \varphi\left(\frac{\|w\|_2}{\sqrt{n}\|w\|_\infty}\right) \geq \frac{(1-\epsilon)\max\{\rho a^2, (1-\rho)b^2\}\sqrt{n \log n} \cdot \|w\|_\infty \cdot c\|w\|_2}{\sqrt{n}\|w\|_\infty}$$
$$= (1-\epsilon)c\max\{\rho a^2, (1-\rho)b^2\}\sqrt{\log n}\|w\|_2.$$

Additionally, for any fixed $m \in [n]$, $(A - A^*)_{m.}w \sim \mathcal{N}(0, \|w_{-m}\|_2^2)$. Therefore,

$$\mathbb{P}\left(|(A-A^*)_{m.}w| \leq \Delta^* \|w\|_\infty \, \varphi\left(\frac{\|w\|_2}{\sqrt{n}\|w\|_\infty}\right)\right)$$

$$\geq \mathbb{P}\left(|(A-A^*)_{m.}w| \leq (1-\epsilon)c\max\{\rho a^2, (1-\rho)b^2\}\sqrt{\log n}\|w\|_2\right)$$

$$= \mathbb{P}\left(|\mathcal{N}(0, \|w_{-m}\|_2^2)| \leq (1-\epsilon)c\max\{\rho a^2, (1-\rho)b^2\}\sqrt{\log n}\|w\|_2\right)$$

$$\geq \mathbb{P}\left(|\mathcal{N}(0, \|w\|_2^2)| \leq (1-\epsilon)c\max\{\rho a^2, (1-\rho)b^2\}\sqrt{\log n}\|w\|_2\right)$$

$$= \mathbb{P}\left(|\mathcal{N}(0, 1)| \leq (1-\epsilon)c\max\{\rho a^2, (1-\rho)b^2\}\sqrt{\log n}\right)$$

$$\geq 1 - \frac{2e^{-(1-\epsilon)^2 c^2 \max\{\rho a^2, (1-\rho)b^2\}^2 \log n/2}}{(1-\epsilon)c\max\{\rho a^2, (1-\rho)b^2\}\sqrt{\log n}\sqrt{2\pi}} \qquad \text{(using Lemma B.2)}$$

$$= 1 - \frac{2n^{-(1-\epsilon)^2 c^2 \max\{\rho a^2, (1-\rho)b^2\}^2/2}}{(1-\epsilon)c\max\{\rho a^2, (1-\rho)b^2\}\sqrt{2\pi \log n}}.$$

Therefore, letting

$$\delta_1 = \frac{2n^{1-(1-\epsilon)^2 c^2 \max\{\rho a^2, (1-\rho)b^2\}^2/2}}{(1-\epsilon)c\max\{\rho a^2, (1-\rho)b^2\}\sqrt{2\pi \log n}}, \quad \text{and setting} \quad c = \frac{2}{(1-\epsilon)\max\{\rho a^2, (1-\rho)b^2\}},$$

we get $\delta_1 = o(1)$. Finally, applying Lemma E.1, we obtain that with probability $1 - \delta_0 - 2\delta_1 = 1 - o(1)$

$$\min_{s \in \{\pm 1\}}\left\|u_1 - \frac{Au_1^*}{\lambda_1^*}\right\|_\infty \lesssim \kappa(\kappa + \varphi(1))(\gamma + \varphi(\gamma))\|u_1^*\|_\infty + \frac{\gamma\|A^*\|_{2\to\infty}}{\Delta^*}$$

$$\leq (1+c)(1+c)\gamma\|u_1^*\|_\infty + \frac{\gamma\|A^*\|_{2\to\infty}}{\Delta^*} \quad \text{(since } \kappa = 1 \text{ and } \varphi(x) = cx)$$

$$= \frac{1}{\Theta(\sqrt{n \log n})}.$$

We used $\gamma = \frac{1}{\Theta(\sqrt{\log n})}$, $\|u_1^*\|_\infty = O(\frac{1}{\sqrt{n}})$, $\|A^*\|_{2\to\infty} = \sqrt{\log n}$, and $\Delta^* = \sqrt{n \log n}$. $\qquad \square$

## E.2 Entrywise Analysis of Eigenvectors for SBM.

In this subsection, we show that the similar behavior also holds for eigenvectors of $A$ sampled from the SBM. More specifically, we restrict ourselves to the case when the expectation $A^*$ (after the appropriate diagonal correction) has rank 2. This is achieved when

$$\frac{a_1}{b} \neq \frac{b}{a_2}.$$

In this case, the eigenvectors that correspond to the top two leading eigenvalues (in magnitude) exhibit the entrywise behavior, which is formalized in the following lemma.

**Lemma E.4.** *Let* $\rho \in (0,1)$ *and* $a_1, a_2, b > 0$ *such that* $a_1 a_2 \neq b^2$. *Let* $(A, \sigma^*) \sim \text{SBM}_n(\rho, a_1, a_2, b)$. *Condition on* $\sigma^*$ *such that the event E from* (13) *holds and let* $A := \mathbb{E}[A \mid \sigma^*]$. *Define* $\{(\lambda_i, u_i)\}_{i \in [n]}$ *and* $\{(\lambda_i^*, u_i^*)\}_{i \in [n]}$ *as above. Then with probability* $1 - O(n^{-3})$

$$\min_{s_1 \in \{\pm 1\}}\left\|s_1 u_1 - \frac{Au_1^*}{\lambda_1^*}\right\|_\infty \leq \frac{C}{\sqrt{n}\log\log n} \quad \text{and} \quad \min_{s_2 \in \{\pm 1\}}\left\|s_2 u_2 - \frac{Au_2^*}{\lambda_2^*}\right\|_\infty \leq \frac{C}{\sqrt{n}\log\log n},$$

*for some constant* $C := C(\rho, a_1, a_2, b)$.

This again requires verifying Assumptions 1-4 for the top two eigenpairs. We note that Dhara et al. (2022b) showed a similar lemma for the special case of Planted Dense Subgraph (PDS), and our proof just generalizes their results. In order to do this, we first note down a couple of important lemmas. The first one directly establishes the spectral norm concentration (Assumption 3).

**Lemma E.5.** *Let $\rho \in (0, 1)$ and $a_1, a_2, b > 0$. Sample $(A, \sigma^*) \sim \mathsf{SBM}_n(\rho, a_1, a_2, b)$. Condition on $\sigma^*$ such that the event $E$ from* (13) *holds. Let $A^* := \mathbb{E}[A \mid \sigma^*]$, then there exists a constant $c_1 = c_1(\rho, a_1, a_2, b) > 0$ such that*

$$\mathbb{P}(\|A - A^*\|_2 \leq c_1 \sqrt{\log n}) \geq 1 - n^{-3}.$$

*Proof.* The lemma is a special case of (Hajek et al., 2016, Theorem 5), invoking the theorem with $c = 3$. □

The next lemma establishes that the leading two eigenvalues of $A^*$, in the rank-2 case, are different in the following sense.

**Lemma E.6.** *Consider $\rho \in (0, 1)$ and $a_1, a_2, b > 0$ such that $a_1 a_2 \neq b^2$. Let $(A, \sigma^*) \sim \mathsf{SBM}_n(\rho, a_1, a_2, b)$. Condition on a labelling $\sigma^*$ such that the event $E$ from* (13) *holds. Let $A^* := \mathbb{E}[A \mid \sigma^*]$. Then the top two eigenvalues in magnitude are given by $\lambda_1^* = (1 + o(1))\theta_1 \log n$ and $\lambda_2^* = (1 + o(1))\theta_2 \log n$, for some non-zero constants $\theta_1 \neq \theta_2$ in terms of $(\rho, a_1, a_2, b)$. As a consequence, $|\lambda_1^* - \lambda_2^*| = \Theta(\log n)$.*

*Proof.* The proof follows similar arguments as the proof of (Dhara et al., 2022b, Lemma 3.2). We note that they prove the special case of the PDS when $a_2 = b$, but the same argument directly generalizes as long as $a_1 a_2 \neq b^2$. □

*Proof of Lemma E.4.* The entire analysis is done conditioned on $\sigma^*$ such that $E$ holds. We will verify Assumptions 1-4 for the leading two eigenpairs. First note that after adding a diagonal matrix $D$, whose entries are $O(\frac{\log n}{n})$, the matrix $A^* + D$ has rank 2, and its remaining eigenvalues satisfy $\tilde{\lambda}_3 = \cdots = \tilde{\lambda}_n = 0$, where $|\tilde{\lambda}_1| \geq \ldots |\tilde{\lambda}_n|$. Applying Weyl's inequality for $3 \leq i \leq n$,

$$|\lambda_i^* - \tilde{\lambda}_i| = |\lambda_i^*| \leq \|D\|_2 = O(\log n / n). \tag{18}$$

By the definition of the eigengap quantity in (17) for both the eigenvalues respectively

$$\Delta_1^* := |\lambda_1^*| \wedge \min_{i \neq 1} |\lambda_i^* - \lambda_1^*| = \Theta(\log n) \text{ and } \Delta_2^* := |\lambda_2^*| \wedge \min_{i \neq 2} |\lambda_i^* - \lambda_2^*| = \Theta(\log n),$$

where we used (18) and Lemma E.6. We also define $\kappa_1 := \frac{|\lambda_1^*|}{\Delta_1^*}$ and $\kappa_2 := \frac{|\lambda_2^*|}{\Delta_2^*}$. We first make an inportant observation, that to verify Assumptions 1-4 for both eigenpairs separately, it suffices to just verify them with $\Delta^*$ and $\kappa$ such that

$$\Delta^* := \min\{\Delta_1^*, \Delta_2^*\} = \Theta(\log n) \text{ and } \kappa := \max\{\kappa_1, \kappa_2\}$$

To verify Assumption 1, first let $\tau = 2\max\{a_1, a_2, b\}$. Fixing any $i \in C_+$,

$$\|A_{i\cdot}\|_2 = \sqrt{|C_+^{-i}|\left(\frac{a_1 \log n}{n}\right)^2 + |C_-|\left(\frac{b \log n}{n}\right)^2}$$

$$= \sqrt{(1 + o(1))\rho\left(\frac{a_1^2 \log^2 n}{n}\right) + (1 + o(1))(1 - \rho)\left(\frac{b^2 \log^2 n}{n}\right)} \leq \frac{\tau \log n}{\sqrt{n}}.$$

Similarly, even for $i \in C_-$

$$\|A_{i\cdot}\|_2 = \sqrt{|C_+|\left(\frac{b \log n}{n}\right)^2 + |C_-^{-i}|\left(\frac{a_2 \log n}{n}\right)^2} \leq \frac{\tau \log n}{\sqrt{n}}$$

Combining both bounds, we obtain

$$\|A^*\|_{2 \to \infty} \leq \frac{\tau \log n}{\sqrt{n}}.$$

We now define the parameter $\gamma$ in terms of $\Delta^*$ and the constant $c_1$ from Lemma E.5:

$$\gamma \triangleq \frac{c_1 \sqrt{\log n}}{\Delta^*} = \frac{c_1 \sqrt{\log n}}{\Theta(\log n)} = o(1).$$

Then $\gamma \Delta^* = c_1 \sqrt{\log n} = \Omega(\sqrt{\log n})$, which dominates $\tau \log n / \sqrt{n}$. This implies $\|A^*\|_{2\to\infty} \le \gamma \Delta^*$, verifying Assumption 1. Assumption 2 trivially holds due to the conditional independence of the entries of $A$, conditioned on $\sigma^*$. By Lemma E.5, Assumption 3 holds with $\delta_0 = n^{-3}$

$$\mathbb{P}(\|A - A^*\|_2 \le \gamma \Delta^*) \ge 1 - n^{-3}.$$

To verify Assumption 4, we let

$$\varphi(x) \triangleq \frac{(2\tau + 4) \log n}{\Delta^* (1 \vee \log(1/x))} \text{ for } x > 0 \text{ and } \varphi(0) = 0.$$

It is straightforward to verify that $\varphi$ satisfies the desired property stated in Assumption 4 and $\varphi(\gamma) = O\left(1/\log\log n\right)$. Also, $\kappa = O(1)$ since both $\Delta_1^* \asymp \Delta_2^* \asymp \log n$, and by Lemma E.6, also $|\lambda_1^*| \asymp |\lambda_2^*| \asymp \log n$. This implies $32\kappa \max\{\gamma, \varphi(\gamma)\} = o(1)$ verifying the first part of the assumption.

To verify the row concentration part, we simply apply (Abbe et al., 2020, Lemma 7) with $p = \tau \log n / n$ and $\alpha = 4/\tau$. We obtain that for a fixed vector $w \in \mathbb{R}^n$ and $m \in [n]$,

$$\mathbb{P}\left(|(A - A^*)_{m\cdot} w| \le \frac{(2\tau + 4)\log n}{\max\left\{1, \log\left(\frac{\sqrt{n}\|w\|_\infty}{\|w\|_2}\right)\right\}} \|w\|_\infty\right) \ge 1 - 2n^{-4}.$$

Substituting the definition of $\Delta^*$ and $\varphi(\cdot)$,

$$\mathbb{P}\left(|(A - A^*)_{m\cdot} w| \le \Delta^* \|w\|_\infty \varphi\left(\frac{\|w\|_2}{\sqrt{n}\|w\|_\infty}\right)\right) \ge 1 - 2n^{-4},$$

which verifies Assumption 4 with $\delta_1 = 2n^{-3}$. Finally, applying Lemma E.1, with probability $1 - \delta_0 - 2\delta_1 = 1 - O(n^{-3})$,

$$\min_{s_1 \in \{\pm 1\}} \left\| s_1 u_1 - \frac{A u_1^*}{\lambda_1^*} \right\|_\infty \lesssim \kappa(\kappa + \varphi(1))(\gamma + \varphi(\gamma)) \|u_1^*\|_\infty + \frac{\gamma \|A^*\|_{2\to\infty}}{\Delta^*} = O\left(\frac{1}{\sqrt{n}\log\log n}\right).$$

We used $\gamma = \frac{1}{\Theta(\sqrt{\log n})}$, $\varphi(\gamma) = O(\frac{1}{\log\log n})$, $\|u_1^*\|_\infty = O(\frac{1}{\sqrt{n}})$, $\|A^*\|_{2\to\infty} = O\left(\frac{\log n}{\sqrt{n}}\right)$, and $\Delta^* = \Theta(\log n)$.

Similarly, with probability $1 - O(n^{-3})$, we have

$$\min_{s_2 \in \{\pm 1\}} \left\| s_2 u_2 - \frac{A u_2^*}{\lambda_2^*} \right\|_\infty = O\left(\frac{1}{\sqrt{n}\log\log n}\right).$$

The proof is complete by a union bound. $\qquad \square$

## F  PROOFS AND ALGORITHMS FOR ROS.

In Appendix F.1, we first derive the form of genie scores. In Appendices F.2 we give our spectral algorithm formally and prove Theorem 1. Finally, in Appendix F.3, we provide degree-profiling algorithm under enormous BEC and BSC channel.

### F.1  GENIE SCORES' FORMULA WHEN NO SIDE INFORMATION

We start by noting the form of genie scores when no side information is present.

**Lemma F.1.** *Fix $\rho \in (0,1)$ and $a, b \in \mathbb{R}$ such that $\max\{|a|, |b|\} > 0$. Let $(A, \sigma^*) \sim \mathsf{ROS}_n(\rho, a, b)$. Then for any $i \in [n]$*

$$z_i^* = (a - b)\sqrt{\frac{\log n}{n}}\left(a \sum_{j \in C_+^{-i}} A_{ij} + b \sum_{j \in C_-^{-i}} A_{ij}\right) + \frac{\log n}{2n}(|C_+^{-i}|(a^2 b^2 - a^4) + |C_-^{-i}|(b^4 - a^2 b^2))$$

$$+ \log\left(\frac{\rho}{1 - \rho}\right).$$

*Moreover, conditioned on the event $E$ from (13), the genie score vector $z^* \in \mathbb{R}^n$ can be written as*

$$z^* = (a-b)\sqrt{\frac{\log n}{n}} Av^* + \left(\gamma + \log\left(\frac{\rho}{1-\rho}\right)\right) \mathbf{1}_n + o(1),$$

*where $\gamma = \left(\rho(a^2b^2 - a^4) + (1-\rho)(b^4 - a^2b^2)\right)\log n/2$ and $v^*$ is given by (2).*

*Proof.* For ease of notation, we denote $f(n) = \sqrt{\log n/n}$. First, note that ROS is a special case of GBM with $\mathcal{P}_+ = \mathcal{N}(a^2 f(n), 1), \mathcal{P}_- = \mathcal{N}(b^2 f(n), 1)$ and $\mathcal{Q} = \mathcal{N}(abf(n), 1)$. Applying Lemma 4.2 for this special case, we obtain the Genie score expressions; for any $i \in [n]$,

$$
\begin{aligned}
z_i^* &= \sum_{j \in C_+^{-i}} \log\left( e^{-\frac{(A_{ij} - a^2 f(n))^2}{2}} \Big/ e^{-\frac{(A_{ij} - abf(n))^2}{2}} \right) \\
&\quad + \sum_{j \in C_-^{-i}} \log\left( e^{-\frac{(A_{ij} - abf(n))^2}{2}} \Big/ e^{-\frac{(A_{ij} - b^2 f(n))^2}{2}} \right) + \log\left(\frac{\rho}{1-\rho}\right) \\
&= \sum_{j \in C_+^{-i}} \left( \frac{(A_{ij} - abf(n))^2}{2} - \frac{(A_{ij} - a^2 f(n))^2}{2} \right) \\
&\quad + \sum_{j \in C_-^{-i}} \left( \frac{(A_{ij} - b^2 f(n))^2}{2} - \frac{(A_{ij} - abf(n))^2}{2} \right) + \log\left(\frac{\rho}{1-\rho}\right) \\
&= (a^2 - ab)f(n)\sum_{j \in C_+^{-i}} A_{ij} + (ab - b^2)f(n)\sum_{j \in C_-^{-i}} A_{ij} \\
&\quad + \frac{f^2(n)}{2}(|C_+^{-i}|(a^2b^2 - a^4) + |C_-^{-i}|(b^4 - a^2b^2)) + \log\left(\frac{\rho}{1-\rho}\right) \\
&= (a-b)\sqrt{\frac{\log n}{n}}\left( a\sum_{j \in C_+^{-i}} A_{ij} + b\sum_{j \in C_-^{-i}} A_{ij} \right) \\
&\quad + \frac{\log n}{2n}(|C_+^{-i}|(a^2b^2 - a^4) + |C_-^{-i}|(b^4 - a^2b^2)) + \log\left(\frac{\rho}{1-\rho}\right).
\end{aligned}
\tag{19}
$$

Conditioned on $E$, simplifying a term from (19)

$$
\begin{aligned}
\frac{\log n}{2n}(|C_+^{-i}|(a^2b^2 &- a^4) + |C_-^{-i}|(b^4 - a^2b^2)) \\
&= \frac{\log n}{2n}\left(\rho n(a^2b^2 - a^4) + (1-\rho)n \cdot (b^4 - a^2b^2)\right) + o(1) \\
&= \left(\rho(a^2b^2 - a^4) + (1-\rho)(b^4 - a^2b^2)\right)\log n/2 + o(1) \\
&= \gamma + o(1).
\end{aligned}
$$

Therefore, substituting this in (19), we obtain that for any $i \in [n]$:

$$
\begin{aligned}
z_i^* &= (a-b)\sqrt{\frac{\log n}{n}}\left( a\sum_{j \in C_+^{-i}} A_{ij} + b\sum_{j \in C_-^{-i}} A_{ij} \right) + \gamma + \log\left(\frac{\rho}{1-\rho}\right) + o(1). \\
&= (a-b)\sqrt{\frac{\log n}{n}} A_{i.}v^* + \gamma + \log\left(\frac{\rho}{1-\rho}\right) + o(1).
\end{aligned}
$$

Writing the same for all $i \in [n]$ in vector notation, we obtain

$$z^* = (a-b)\sqrt{\frac{\log n}{n}} Av^* + \left(\gamma + \log\left(\frac{\rho}{1-\rho}\right)\right) \mathbf{1}_n + o(1).$$

$\square$

Roughly speaking, these genie scores in absolute value are on a $\log n$ scale for both ROS and SBM. This is when the exact recovery becomes statistically possible and explains the scaling choices for both models.

### F.2 Spectral Algorithm and Proof of Theorem 1

Below is our spectral algorithm which takes $A$ (and optionally $y$ when available) as input along with the parameters and returns an estimator $\hat{\sigma}_{\text{spec}}$. One of the (two) score vectors formed by the algorithm approximates the genie score $z^*$.

---

**Algorithm 2** Spectral recovery algorithm for ROS, without or with side information.

---

**Input:** An $n \times n$ observation matrix $A$ and parameters $(\rho, a, b)$. Optionally, side information $y \in \mathcal{Y}^n$ such that we can compute likelihoods of laws $\mathcal{S}_+$ and $\mathcal{S}_-$.

**Output:** An estimate of community assignments $\hat{\sigma}_{\text{spec}}$.

1: **Compute leading eigenpairs.** Compute the top eigenpair of $A$, denoted by $(\lambda_1, u_1)$, where $|\lambda_1| \geq \cdots \geq |\lambda_n|$.
2: **Compute coefficients of linear combination.**

$$c_1 := \sqrt{n} \log n \cdot (a-b) \cdot (\rho a^2 + (1-\rho)b^2)^{3/2} \text{ and } \gamma := \left( \rho(a^2 b^2 - a^4) + (1-\rho)(b^4 - a^2 b^2) \right) \frac{\log n}{2}.$$

3: **Compute spectral scores.** For any $s \in \{\pm 1\}$, prepare the spectral score vectors as follows.
   - No side information:
   $$z^{(s)} = s c_1 u_1 + \gamma \mathbf{1}_n.$$
   - Side information:
   $$z^{(s)} = s c_1 u_1 + \gamma \mathbf{1}_n + \log \left( \frac{\mathcal{S}_+}{\mathcal{S}_-}(y) \right)$$

4: **Remove sign ambiguity.** For each $s \in \{\pm 1\}$, let $\hat{\sigma}^{(s)} = \text{sgn}(z^{(s)})$.
   - No side information: Return $\hat{\sigma}_{\text{spec}} = \arg\max_{\{\hat{\sigma}^{(s)}:s\in\{\pm 1\}\}} \mathbb{P}(\sigma^* = \hat{\sigma}^{(s)} \mid A)$.
   - BEC or BSC side information: Return $\hat{\sigma}_{\text{spec}} = \arg\max_{\{\hat{\sigma}^{(s)}:s\in\{\pm 1\}\}} \mathbb{P}(\sigma^* = \hat{\sigma}^{(s)} \mid A, y)$.

---

The values $c_1$ and $\gamma$ are carefully designed to emulate the genie score. Since the eigenvectors are only recovered up to a global direction flip, we need to keep both candidates in the algorithm. One of them is approximating the genie score well. Finally, whichever one has the higher posterior probability is picked in step 4. To show the proof of the score approximation guarantee, we need the following lemma, whose proof is included at the end of this subsection.

**Lemma F.2.** *Fix $\rho \in (0,1)$ and $a, b \in \mathbb{R}$ such that $\max\{|a|, |b|\} > 0$. Let $(A, \sigma^*) \sim \text{ROS}_n(\rho, a, b)$. Condition on $\sigma^*$ satisfying $E$ from (13) holds for it. Then there exists a constant $c := c(\rho, a, b)$ such that with probability $1 - O(n^{-3})$, the following event holds*

$$E_1 := \left\{ \sqrt{\frac{\log n}{n}} \|Av^*\|_\infty \leq c \log n \right\}. \tag{20}$$

Below is our primary lemma which shows the spectral and genie score vector approximation in $\ell_\infty$ norm.

**Lemma F.3.** *Fix $\rho \in (0,1)$ and $a, b$ such that $\max\{|a|, |b|\} > 0$. Let $(A, \sigma^*) \sim \text{ROS}_n(\rho, a, b)$ and condition on $\sigma^*$ such that $E$ from (13) holds. Optionally, let $y \sim \text{SI}(\sigma^*, \mathcal{S}_+, \mathcal{S}_-)$ for the channel laws $(\mathcal{S}_+, \mathcal{S}_-)$. Let $z^*$ and $z^{(s)}$ be the genie score and the spectral score vectors respectively for the corresponding model. Then with probability $1 - o(1)$,*

$$\min_{s \in \{\pm 1\}} \left\| z^* - z^{(s)} \right\|_\infty = o(\log n).$$

*Proof.* First of all, note that conditioned on $E$, we have $\lambda_1^* = (1 + o(1))\sqrt{\frac{\log n}{n}} \|v^*\|_2^2$, and $u_1^* = (1 + o(1)) \frac{v^*}{\|v^*\|_2}$. Additionally, $\|v^*\|_2 = \sqrt{|C_+|a^2 + |C_-|b^2} = (1 + o(1))\sqrt{n}\sqrt{\rho a^2 + (1-\rho)b^2}$.

Using these, one can simplify

$$\frac{c_1 A u_1^*}{\lambda_1^*} = \frac{(1 + o(1))c_1 A v^*}{\lambda_1^* \|v^*\|_2} \approx \frac{\sqrt{n} \log n (a - b)(\rho a^2 + (1 - \rho)b^2)^{3/2} A v^*}{\sqrt{\log n/n} \|v^*\|_2^3} \approx \sqrt{\frac{\log n}{n}}(a - b) A v^*, \tag{21}$$

where in the last step we substitute $\|v^*\|_2$.

Now the high probability event in this lemma is such that (i) the behavior of eigenvectors as stated in Lemma E.4 and (ii) the event $E_1$ from (20) hold. By Lemma E.2 and F.2, they both happen with probability $1 - o(1)$. Additionally, let $s$ be the sign for which the conclusion of Lemma E.2 holds. We now analyze the three models separately.

- No side information: For every $i \in [n]$

$$|z^{(s)} - z_i^*| = |c_1 s(u_1)_i + \gamma - z_i^*|, \qquad \text{(recall Algorithm 2)}$$

$$= \left| \frac{c_1 (A u_1^*)_i}{\lambda_1^*} + \gamma - z_i^* \right| + O\left( \frac{c_1}{\sqrt{n} \log n} \right)$$
$$\text{(by Lemma E.4 and the triangle inequality)}$$

$$= \left| (1 + o(1))\sqrt{\frac{\log n}{n}}(a - b)(A v^*)_i + \gamma - z_i^* \right| + O(\sqrt{\log n})$$
$$\text{(using (21) and } c_1 \asymp \sqrt{n} \log n)$$

$$= \left| (1 + o(1))\sqrt{\frac{\log n}{n}}(a - b)(A v^*)_i + \gamma - \sqrt{\frac{\log n}{n}}(a - b) A_{i.} v^* - \gamma - O(1) \right|$$
$$+ O(\sqrt{\log n}) \qquad \text{(putting } z_i^* \text{ from Lemma F.1)}$$

$$= o(1)\sqrt{\frac{\log n}{n}}|(A v^*)_i| + O(\sqrt{\log n}) = o(\log n), \qquad \text{(recall } E_1 \text{ from (20))}$$

- Side information: By Lemma 4.2 and Algorithm 2 (step 3), when side information is provided, both the genie score vector $z^*$ and the spectral score vector $z^{(s)}$ are achieved by adding $\ln\left( \frac{S_+}{S_-}(y) \right)$ to their counterpart when no side information is provided. Therefore, the triangle inequality along with the analysis in the no side information case gives us that for every $i \in [n]$, we have $|z_i^{(s)} - z_i^*| = o(\log n)$.

In either case, one can equivalently write conclusions for all $i \in [n]$ together in vector notation

$$\min_{s \in \{\pm 1\}} \left\| z^{(s)} - z^* \right\|_\infty = o(\log n).$$

$\square$

We are finally set to prove our first main result in Theorem 1.

*Proof of Theorem 1.* We note that step 4 of Algorithm 2 keeps two candidates $\{\hat{\sigma}^{(s)} : s \in \{\pm 1\}\}$ and chooses the one which has maximum posterior probability. Therefore, to show that $\hat{\sigma}_{\text{spec}}$ achieves exact recovery above the IT threshold, it suffices to show that one of the two candidates achieves exact recovery, and $\hat{\sigma}_{\text{MAP}}$ also succeeds above the IT threshold, which ensures that the algorithm selects the correct vector by maximizing the posterior probability. The MAP estimator achieves exact recovery whenever $I^* >$ is already shown in Proposition 2.10. It remains to show that one of $\{\hat{\sigma}^{(s)} : s \in \{\pm 1\}\}$ succeeds. To this end, recall Lemma F.3 that with probability $1 - o(1)$,

$$\min_{s \in \{\pm 1\}} \left\| z^* - z^{(s)} \right\|_\infty = o(\log n).$$

Moreover, whenever $I^* > 1$, by Lemma D.1 and union bound over $i \in [n]$, there exists $\delta > 0$ such that

$$\mathbb{P}\left( \min_{i \in [n]} \sigma_i^* z_i^* > \delta \log n \right) = 1 - o(1).$$

Taking a union bound over these two events, there exists $\varsigma > 0$ and $s^* \in \{\pm 1\}$ such that

$$\mathbb{P}\left(\min_{i\in[n]} \sigma_i^* z_i^{(s^*)} > \varsigma \log n\right) = 1 - o(1).$$

Since $\hat{\sigma}^{(s^*)} = \mathrm{sgn}(z^{(s^*)})$ in step 4, we obtain $\hat{\sigma}^{(s^*)}$ achieves exact recovery. As a consequence, even $\hat{\sigma}_{\mathrm{spec}}$ achieves exact recovery above the IT threshold. In other words,

$$\lim_{n\to\infty} \mathbb{P}\left(\hat{\sigma}_{\mathrm{spec}} \text{ succeeds}\right) = 1.$$

$\square$

We finally return to the proof of the lemma already mentioned.

*Proof of Lemma F.2.* For each $i \in [n]$, first define $Y_i = \sqrt{\frac{\log n}{n}}(Av^*)_i$. We first note that $Y_i$ is a Gaussian random variable as it is the sum of at most $n$ independent Gaussian random variables. Therefore, we have $Y_i \sim \mathcal{N}(\mu_i, \sigma_i^2)$ for certain $\mu_i$ and $\sigma_i^2$, which we will calculate later. Applying Lemma B.2 for any $Y_i$ (after normalizing) yields

$$\mathbb{P}\left(\frac{|Y_i - \mu_i|}{\sigma_i} \geq 4\sqrt{\log n}\right) \leq e^{-8\log n} = n^{-8}.$$

Rearrangement of the terms using the triangle inequality along with a union bound over all $i \in [n]$ gives us

$$\mathbb{P}\left(\forall i \in [n] : |Y_i| \leq |\mu_i| + 4\sigma_i\sqrt{\log n}\right) \geq 1 - n^{-7}.$$

Therefore it simply suffices to show that for every $i \in [n]$, the quantity $|\mu_i| + 4\sigma_i\sqrt{\log n} = O(\log n)$. To this end, we first observe that $(Av^*)_i$ is the sum of $n-1$ independent Gaussian random variables all with means whose absolute values are $O\left(\sqrt{\frac{\log n}{n}}\right)$. Thus,

$$|\mu_i| = \sqrt{\frac{\log n}{n}}\,\mathbb{E}[(Av^*)_i] = \sqrt{\frac{\log n}{n}}(n-1) \cdot O\left(\sqrt{\frac{\log n}{n}}\right) = O(\log n).$$

Similarly, $(Av^*)_i$ is the sum of $n-1$ independent Gaussian random variables with variances $O(1)$. This gives us

$$\sigma_i^2 = \mathrm{Var}\left[\sqrt{\frac{\log n}{n}}(Av^*)_i\right] = \frac{\log n \cdot \mathrm{Var}\left[(Av^*)_i\right]}{n} = \frac{\log n \cdot O(n)}{n} = O(\log n),$$

which also implies

$$4\sigma_i\sqrt{\log n} = O(\log n).$$

$\square$

### F.3 Degree-Profiling Algorithm for BEC and BSC Channels

The following is a simple degree-profiling algorithm that tries to mimic the genie naïvely and achieves exact recovery if side information is substantial to shift the thresholds of exact recovery.

---

**Algorithm 3** Degree-Profiling algorithm for ROS in the presence of BEC or BSC side information.

---

**Input:** An $n \times n$ observation matrix $A$ and parameters $(\rho, a, b)$. The BEC side information $y$ with parameter $\epsilon$ *or* BSC side information $y$ with parameter $\alpha$.

**Output:** An estimate of community assignments $\hat{\sigma}_{\mathrm{dp}}$.

1: Let $S_+ := \{i : y_i = +1\}, S_- := \{i : y_i = -1\}$, and

$$\gamma := \left(\rho(a^2b^2 - a^4) + (1-\rho)(b^4 - a^2b^2)\right)\log n / 2.$$

Compute $z \in \mathbb{R}^n$ such that, for every $i \in [n]$

$$z_i = a(a-b)\sqrt{\frac{\log n}{n}}\sum_{j\in S_+} A_{ij} + b(a-b)\sqrt{\frac{\log n}{n}}\sum_{j\in S_-} A_{ij} + \gamma.$$

2: Prepare the degree-profile score vector $z^{\mathrm{dp}}$ as follows.

- BEC side information: For any $i \in [n]$,

$$z_i^{\mathrm{dp}} = \begin{cases} z_i & \text{if } y_i = 0; \\ +\infty, & \text{if } y_i = +1; \\ -\infty & \text{if } y_i = -1; \end{cases}$$

- BSC side information:

$$z^{\mathrm{dp}} = z + \ln\left(\frac{1-\alpha}{\alpha}\right) y$$

3: Return $\hat{\sigma}_{\mathrm{dp}} = \mathrm{sgn}(z^{\mathrm{dp}})$.

The following is our formal theorem.

**Theorem 3.** *Fix $\rho \in (0,1)$ and $a, b \in \mathbb{R}$ such that $\max\{|a|, |b|\} > 0$. Let $(A, \sigma^*) \sim \mathrm{ROS}_n(\rho, a, b)$. Let $y \sim \mathrm{BEC}(\sigma^*, \epsilon)$ or $y \sim \mathrm{BSC}(\sigma^*, \alpha)$, where*

$$\lim_{n \to \infty} \frac{\log(1/\epsilon)}{\log n} = \beta \text{ and } \lim_{n \to \infty} \frac{\log(\frac{1-\alpha}{\alpha})}{\log n} = \beta$$

*for some $\beta > 0$. Then $I^*$ from (6) is well-defined and there is a degree-profiling algorithm (Algorithm 3) that returns the estimator $\hat{\sigma}_{\mathrm{dp}}$ which achieves exact recovery whenever $I^* > 1$.*

The following lemma plays a crucial role in the analysis of our degree profiling algorithm which formalizes the notion of receiving most of the labels correct.

**Lemma F.4.** *Let $\sigma^* \in \{\pm 1\}^n$ be sampled such that each entry is i.i.d. with $\mathbb{P}(\sigma_i^* = +1) = \rho$. Condition on $\sigma^*$ such that the event $E$ from (13) holds. For any $\beta > 0$, we let $y \sim \mathrm{BEC}(\sigma^*, \epsilon)$ or $y \sim \mathrm{BSC}(\sigma^*, \alpha)$ for $\epsilon$ and $\alpha$ scales as described in Theorem 3. Define $S_+ = \{i : y_i = +1\}$ and $S_- = \{i : y_i = -1\}$. Then with probability $1 - o(1)$*

$$\max\{|C_+ \setminus S_+|, |C_- \setminus S_-|\} = O\left(\frac{n}{\log^{10} n}\right).$$

*Proof.* First, recall that conditioned on the even $E$ about $\sigma^*$, we have $|C_+| = \Theta(n)$ and $|C_-| = \Theta(n)$. We now consider the two types of side information.

- BEC side information: Observe that

$$\mathbb{E}[|C_+ \setminus S_+|] = \sum_{i \in C_+} \mathbb{P}(y_i = 0) = |C_+|\epsilon = n^{-\beta}|C_+| \leq n^{1-\beta}.$$

Then Markov's inequality immediately implies that, with probability $1 - O(n^{-\beta/2})$, we have

$$|C_+ \setminus S_+| \leq n^{1-\beta/2} = O\left(n/\log^{10} n\right).$$

Similarly, we also have $\mathbb{E}[|C_- \setminus S_-|] = n^{-\beta}|C_-| \leq n^{1-\beta}$. Thus, applying Markov's inequality again implies $|C_- \setminus S_-| = O(n/\log^{10} n)$ with probability $1 - O(n^{-\beta/2})$. A simple union bound over these two events implies, with probability $1 - O(n^{-\beta/2}) = 1 - o(1)$

$$\max\{|C_+ \setminus S_+|, |C_- \setminus S_-|\} = O\left(\frac{n}{\log^{10} n}\right).$$

- BSC side information: Under BSC side information, $\mathbb{E}[|C_+ \setminus S_+|] = \alpha|C_+| \leq n^{1-\beta}$ and $\mathbb{E}[|C_- \setminus S_-|] = \alpha|C_-| \leq n^{1-\beta}$. Therefore, using the Markov's inequality for both of these sets along with a union bound immediately implies that with probability $1 - O(n^{-\beta/2}) = 1 - o(1)$,

$$\max\{|C_+ \setminus S_+|, |C_- \setminus S_-|\} = O\left(\frac{n}{\log^{10} n}\right).$$

$\square$

To bound the effect of the error terms when we make $z^{\mathrm{dp}}$ approximation to $z^*$, we need another technical lemma whose proof we include at the end of this section.

**Lemma F.5.** *Consider $\rho \in (0,1)$ and $a, b \in \mathbb{R}$ such that $\max\{|a|, |b|\} > 0$ Let $(A, \sigma^*) \sim \mathsf{ROS}_n(\rho, a, b)$. Condition on $\sigma^*$ such that the event $E$ holds. Fix any set $T \subset C_+$ or $T \subset C_-$ such that $|T| = O(n/\log^{10} n)$. Then for any $i \in [n]$, let us define $Y_i = \sqrt{\frac{\log n}{n}} \sum_{j \in T} A_{ij}$.*

$$\mathbb{P}(\forall i \in [n] : |Y_i| \leq 1) \geq 1 - O(n^{-3}).$$

Using this lemma, we now show that the degree profiling vector $z^{\mathrm{dp}}$ is a good approximation to the genie score vector $z^*$ in $\ell_\infty$ norm.

**Lemma F.6.** *Fix $\rho \in (0,1)$ and $a, b$ such that $\max\{|a|, |b|\} > 0$. Let $(A, \sigma^*) \sim \mathsf{ROS}_n(\rho, a, b)$ and condition on $\sigma^*$ such that $E$ from (13) holds. For $\beta > 0$, let $y \sim \mathsf{BEC}(\sigma^*, \epsilon)$ or $y \sim \mathsf{BSC}(\sigma^*, \alpha)$ where $\epsilon$ and $\alpha$ scales as described in Theorem 3. Let $z^*$ and $z^{\mathrm{dp}}$ respectively be the genie score vector and the degree-profiling score vector produced by Algorithm 3 for the corresponding model of side information. Then (irrespective of the parameter values), with probability $1 - o(1)$,*

$$\left\| z^* - z^{\mathrm{dp}} \right\|_\infty = O(1).$$

*Proof.* We first start by observing, in the case of BEC side information $z^{\mathrm{dp}}$ is just formed by overriding the entries of $z$ from step 1 of Algorithm 3 with $+\infty$ or $-\infty$ depending on the side information label being $+1$ or $-1$. Also, for BSC side information, $z^{\mathrm{dp}} = z + \log\left(\frac{1-\alpha}{\alpha}\right) y$. By Lemmas 4.2, this is precisely how the genie score vector $z^*$ in the respective model of side information relates to the genie score vector without side information which we denote by $z'$.

Therefore, to show the lemma, it suffices to show that, with probability $1 - o(1)$,

$$\|z' - z\|_\infty = O(1).$$

- **BEC side information:**

$$\|z' - z\|_\infty = \max_{i \in [n]} |z'_i - z_i|$$

$$= \max_{i \in [n]} \left| a(a-b)\sqrt{\frac{\log n}{n}} \sum_{j \in C_+ \backslash S_+} A_{ij} + b(a-b)\sqrt{\frac{\log n}{n}} \sum_{j \in C_- \backslash S_-} A_{ij} + \log\left(\frac{\rho}{1-\rho}\right) + o(1) \right|$$

$$\text{(substituting } z' \text{ from Lemma F.1 and } z \text{ from Algorithm 3)}$$

$$\leq \max_{i \in [n]} \left| a(a-b)\sqrt{\frac{\log n}{n}} \sum_{j \in C_+ \backslash S_+} A_{ij} \right| + \left| b(a-b)\sqrt{\frac{\log n}{n}} \sum_{j \in C_- \backslash S_-} A_{ij} \right| + O(1), \qquad (22)$$

where the last step follows from the triangle inequality. By Lemma F.4, both $|C_+ \backslash S_+|$ and $|C_- \backslash S_-|$ are bounded by $O(n/\log^{10} n)$ with probability $1 - o(1)$. Moreover, these sets are chosen only based on the side information $y$ and hence independent of $A$, conditioned on $\sigma^*$. Using Lemma F.5 for these set $C_+ \backslash S_+$ and $C_- \backslash S_-$ as $T$, and using a union bound, we obtain that with probability $1 - o(1)$

$$\max_{i \in [n]} \left| a(a-b)\sqrt{\frac{\log n}{n}} \sum_{j \in C_+ \backslash S_+} A_{ij} \right| = O(1) \quad \text{and} \quad \max_{i \in [n]} \left| b(a-b)\sqrt{\frac{\log n}{n}} \sum_{j \in C_- \backslash S_-} A_{ij} \right| = O(1).$$

Substituting these bounds in (22), with probability $1 - o(1)$

$$\|z' - z\|_\infty = O(1).$$

- BSC side information:

$$\|z' - z\|_\infty = \max_{i \in [n]} |z'_i - z_i|$$

$$= \max_{i \in [n]} \left| a(a-b)\sqrt{\frac{\log n}{n}} \sum_{j \in C_+ \setminus S_+} A_{ij} - a(a-b)\sqrt{\frac{\log n}{n}} \sum_{j \in S_+ \setminus C_+} A_{ij} \right.$$
$$\left. + b(a-b)\sqrt{\frac{\log n}{n}} \sum_{j \in C_- \setminus S_-} A_{ij} - b(a-b)\sqrt{\frac{\log n}{n}} \sum_{j \in S_- \setminus C_-} A_{ij} + \log\left(\frac{\rho}{1-\rho}\right) + o(1) \right|$$

(substituting $z'$ from Lemma F.1 and $z$ from Algorithm 3)

$$= \max_{i \in [n]} \left| (a-b)^2 \sqrt{\frac{\log n}{n}} \sum_{j \in C_+ \setminus S_+} A_{ij} - (a-b)^2 \sqrt{\frac{\log n}{n}} \sum_{j \in C_- \setminus S_-} A_{ij} + \log\left(\frac{\rho}{1-\rho}\right) + o(1) \right|$$

(since $S_+ \setminus C_+ = C_- \setminus S_-$ and $S_- \setminus C_- = C_+ \setminus S_+$)

$$\leq \max_{i \in [n]} \left| (a-b)^2 \sqrt{\frac{\log n}{n}} \sum_{j \in C_+ \setminus S_+} A_{ij} \right| + \left| (a-b)^2 \sqrt{\frac{\log n}{n}} \sum_{j \in C_- \setminus S_-} A_{ij} \right| + O(1), \qquad (23)$$

where in the last step, we used the triangle inequality. Again by similar arguments, first using Lemma F.4, both $|C_+ \setminus S_+|$ and $|C_- \setminus S_-|$ is $O(n/\log^{10} n)$ with probability $1 - o(1)$. Using Lemma F.5 for these set $C_+ \setminus S_+$ and $C_- \setminus S_-$ further implies that, with probability $1 - o(1)$

$$\max_{i \in [n]} \left| (a-b)^2 \sqrt{\frac{\log n}{n}} \sum_{j \in C_+ \setminus S_+} A_{ij} \right| = O(1) \quad \text{and} \quad \max_{i \in [n]} \left| (a-b)^2 \sqrt{\frac{\log n}{n}} \sum_{j \in C_- \setminus S_-} A_{ij} \right| = O(1).$$

Substituting these bounds in (23), with probability $1 - o(1)$

$$\|z' - z\|_\infty = O(1).$$

$\square$

Finally, we prove Theorem 3.

*Proof of Theorem 3.* We have already verified in Appendix C that $I^*$ well-defined for BEC and BSC channels where $\epsilon$ and $\alpha$ scales as described in the theorem statements. When $\beta > 0$, by Lemma F.6 that with probability $1 - o(1)$,

$$\left\| z^* - z^{\mathrm{dp}} \right\|_\infty = O(1).$$

Above the IT threshold by Lemma D.1 and union bound over $i \in [n]$, there exists $\delta > 0$ such that

$$\mathbb{P}\left( \min_{i \in [n]} \sigma_i^* z_i^* > \delta \log n \right) = 1 - o(1).$$

Taking a union bound, there exists $\varsigma > 0$ such that

$$\mathbb{P}\left( \min_{i \in [n]} \sigma_i^* z_i^{\mathrm{dp}} > \varsigma \log n \right) = 1 - o(1).$$

Observing $\hat{\sigma}_{\mathrm{dp}} = \mathrm{sgn}(z^{\mathrm{dp}})$, we obtain $\hat{\sigma}_{\mathrm{dp}}$ achieves exact recovery, i.e.

$$\lim_{n \to \infty} \mathbb{P}\left( \hat{\sigma}_{\mathrm{dp}} \text{ succeeds} \right) = 1.$$

$\square$

We now return to the deferred proof.

*Proof of Lemma F.5.* First of all, observe that $Y_i$ is a Gaussian random variable. Let us say $Y_i \sim \mathcal{N}(\mu_i, \sigma_i^2)$, for some $\mu_i$ and $\sigma_i^2 > 0$. Applying Lemma B.2 for any $Y_i$ (after renormalizing) yields

$$\mathbb{P}\left(\frac{|Y_i - \mu_i|}{\sigma_i} \geq 4\sqrt{\log n}\right) \leq e^{-8\log n} = n^{-8}.$$

Rearrangement of the terms using the triangle inequality along with a union bound over all $i \in [n]$ gives us

$$\mathbb{P}\left(\exists i \in [n] : |Y_i| \geq |\mu_i| + 4\sigma_i\sqrt{\log n}\right) \leq n^{-7}.$$

Therefore it simply suffices to show that for every $i \in [n]$, we have $|\mu_i| + 4\sigma_i\sqrt{\log n} \leq 1$. Indeed, we will show that these terms are $o(1)$. To this end, first consider the term $4\sigma_i\sqrt{\log n}$ for any $i \in [n]$. Recall that $Y_i$ is the sum of at most $|T|$ i.i.d. Gaussian random variables, all with variance 1, scaled by $\sqrt{\log n / n}$. Therefore,

$$\sigma_i^2 \leq \frac{\log n}{n}|T| = O\left(\frac{1}{\log^9 n}\right) \implies 4\sigma_i\sqrt{\log n} = O\left(\frac{1}{\log^2 n}\right) = o(1),$$

where we used $|T| = O(n/\log^{10} n)$. We now show that $|\mu_i| = o(1)$ too for all $i \in [n]$, which requires some casework. First consider $T \subset C_+$, then for any $i \in C_+$:

$$|\mu_i| \leq \sqrt{\frac{\log n}{n}}|T|a^2\sqrt{\frac{\log n}{n}} = O\left(\frac{1}{\log^9 n}\right),$$

Similarly, for any $i \in C_-$:

$$|\mu_i| = \sqrt{\frac{\log n}{n}}|T||ab|\sqrt{\frac{\log n}{n}} = O\left(\frac{1}{\log^9 n}\right).$$

Exactly following the same arguments, we also get the same bounds on $\mu_i$ even when $T \subset C_-$. Overall, we established that

$$\mathbb{P}(\forall i \in [n] : |Y_i| \leq 1) \geq 1 - O(n^{-7}).$$

$\square$

# G  PROOFS AND ALGORITHMS FOR SBM.

We follow the same structure: in Appendix G.1, we derive the formula for genie scores when no side information is available. In Appendix G.2, we present our spectral algorithm with the optimality proof. Finally, in Appendix G.3, we do the degree profiling algorithm.

## G.1  GENIE SCORES' FORMULA WHEN NO SIDE INFORMATION

We begin by showing that, with high probability, all the vertices have degrees logarithmic in $n$.

**Lemma G.1.** *Let $\rho \in (0,1)$ and $a_1, a_2, b > 0$. Let $(A, \sigma^*) \sim \mathsf{SBM}_n(\rho, a_1, a_2, b)$. Condition on $\sigma^*$ such that the event $E$ from (24) holds then. For $c = 6\max\{1, a_1, a_2, b\}$ let*

$$E_1 = \left\{\forall i : \sum_{j \in [n]} A_{ij} \leq c\log n\right\}; \tag{24}$$

*then with $\mathbb{P}(E_1) = 1 - O(n^{-3})$.*

*Proof.* Note that the entries of $A$ (up to symmetry) are independent conditioned on $\sigma^*$. Therefore, for any $i \in [n]$, the $i^{\text{th}}$ row has independent Bernoulli entries with means either $p_1, p_2$ or $q$, where $(p_1, p_2, q) = (a_1, a_2, b)\log n/n$. Therefore, defining $X \sim \mathrm{Binom}(n, \tau\log n/n)$, where $\tau = \max\{a_1, a_2, b\}$, we have that $X$ stochastically dominates $\sum_{j \in [n]} A_{ij}$, for any $i \in [n]$. Then applying the Chernoff bound for Binomial random variables (Mitzenmacher & Upfal, 2017, Theorem 4.4, Equation 4.3) we get, for any $i \in [n]$

$$\mathbb{P}\left(\sum_{j \in [n]} A_{ij} > 6\max\{1, \tau\}\log n\right) \leq \mathbb{P}\left(X > 6\max\{1, \tau\}\log n\right) \leq 2^{-6\log n} = O(n^{-4}).$$

Taking a union bound over all $i \in [n]$ yields the desired claim. $\square$

We next analyze the form of genie scores without side information.

**Lemma G.2.** *Let $\rho \in (0,1)$ and $a_1, a_2, b > 0$. Let $(A, \sigma^*) \sim \mathsf{SBM}_n(\rho, a_1, a_2, b)$. Denote $(p_1, p_2, q) := (a_1, a_2, b) \log n / n$. Then for any $i \in [n]$, the genie score can be written as*

$$z_i^* = \log\left(\frac{p_1(1-q)}{q(1-p_1)}\right) \sum_{j \in C_+^{-i}} A_{ij} + \log\left(\frac{q(1-p_2)}{p_2(1-q)}\right) \sum_{j \in C_-^{-i}} A_{ij} + \log\left(\frac{\rho}{1-\rho}\right)$$

$$+ |C_+^{-i}| \log\left(\frac{1-p_1}{1-q}\right) + |C_-^{-i}| \log\left(\frac{1-q}{1-p_2}\right).$$

*Moreover, conditioned on the event $E$ from (13) and $E_1$ from (24),*

$$\|z^* - Aw^* - \gamma \mathbf{1}_n\|_\infty = O(1),$$

*where $w^* \in \mathbb{R}^n$ is a vector with entries $(w_+, w_-) := (\log(a_1/b), \log(b/a_2))$ on locations of $C_+$ and $C_-$ respectively and $\gamma := (\rho(b - a_1) + (1 - \rho)(a_2 - b)) \log n$.*

*Proof.* First of all, note that the SBM is a special case of the GBM model with $\mathcal{P}_+ \equiv \mathrm{Bern}(p_1)$, $\mathcal{P}_- \equiv \mathrm{Bern}(p_2)$ and $\mathcal{Q} \equiv \mathrm{Bern}(q)$. Using Lemma 4.2 for this special case, for any $i \in [n]$

$$z_i^* = \log\left(\frac{\rho}{1-\rho}\right) + \sum_{i \in C_+^{-i}} \log\left(\frac{p_1^{A_{ij}}(1-p_1)^{(1-A_{ij})}}{q^{A_{ij}}(1-q)^{(1-A_{ij})}}\right) + \sum_{j \in C_-^{-i}} \log\left(\frac{q^{A_{ij}}(1-q)^{(1-A_{ij})}}{p_2^{A_{ij}}(1-p_2)^{(1-A_{ij})}}\right)$$

$$= \log\left(\frac{p_1(1-q)}{q(1-p_1)}\right) \sum_{j \in C_+^{-i}} A_{ij} + \log\left(\frac{q(1-p_2)}{p_2(1-q)}\right) \sum_{j \in C_-^{-i}} A_{ij} + \log\left(\frac{\rho}{1-\rho}\right)$$

$$+ |C_+^{-i}| \log\left(\frac{1-p_1}{1-q}\right) + |C_-^{-i}| \left(\frac{1-q}{1-p_2}\right). \tag{25}$$

To show the second part of the lemma, we further simplify

$$\left|\log\left(\frac{1-q}{1-p_1}\right)\right| = \left|\log\left(1 + \frac{p_1 - q}{(1 - p_1)}\right)\right| = \left|\log\left(1 + \frac{(a_1 - b)\log n}{(1 - p_1)n}\right)\right| = O\left(\frac{\log n}{n}\right). \tag{26}$$

In the last inequality, we used $\frac{x}{x+1} \leq \log(1 + x) \leq x$ for $x > -1$. Similarly,

$$\left|\log\left(\frac{1-p_2}{1-q}\right)\right| = \left|\log\left(1 + \frac{q - p_2}{(1 - q)}\right)\right| = \left|\log\left(1 + \frac{(b - a_2)\log n}{(1 - q)n}\right)\right| = O\left(\frac{\log n}{n}\right). \tag{27}$$

Recall the definition of event $E$ from (13) and $E_1$ from (24). Conditioned on $E \cap E_1$, we simplify (25) using (26) and (27).

$$\log\left(\frac{p_1(1-q)}{q(1-p_1)}\right) \sum_{j \in C_+^{-i}} A_{ij} + \log\left(\frac{q(1-p_2)}{p_2(1-q)}\right) \sum_{j \in C_-^{-i}} A_{ij}$$

$$= \log\left(\frac{a_1}{b}\right) \sum_{j \in C_+^{-i}} A_{ij} + \log\left(\frac{b}{a_2}\right) \sum_{j \in C_-^{-i}} A_{ij} + O\left(\frac{\log n}{n}\right) \sum_{j \in [n]} A_{ij}$$

$$= \log\left(\frac{a_1}{b}\right) \sum_{j \in C_+^{-i}} A_{ij} + \log\left(\frac{b}{a_2}\right) \sum_{j \in C_-^{-i}} A_{ij} + o(1), \tag{28}$$

where the last equality followed by conditioning on $E_1$. We also simplify

$$|C_+^{-i}| \log\left(\frac{1-p_1}{1-q}\right) = |C_+^{-i}| \log\left(1 + \frac{q - p_1}{(1 - q)}\right) = |C_+^{-i}| \log\left(1 + \frac{(b - a_1)\log n}{(1 - q)n}\right)$$

$$= |C_+^{-i}| \left(\frac{(b - a_1)\log n}{(1 - q)n} + O\left(\frac{\log^2 n}{n^2}\right)\right)$$

$$\text{(using a Taylor expansion of } \log(1 + x))$$

$$= (1 + O(n^{-1/3}))\rho n \left(\frac{(b - a_1)\log n}{(1 - q)n} + O\left(\frac{\log^2 n}{n^2}\right)\right)$$

$$= \rho(b - a_1) \log n + o(1) \tag{29}$$

Similarly,

$$|C_-^{-i}| \log\left(\frac{1-q}{1-p_2}\right) = |C_-^{-i}| \log\left(1 + \frac{p_2 - q}{(1-p_2)}\right) = (1-\rho)(a_2 - b)\log n + o(1) \qquad (30)$$

Substituting (28), (29) and (30) into (25),

$$z_i^* = \log\left(\frac{a_1}{b}\right)\sum_{j \in C_+^{-i}} A_{ij} + \log\left(\frac{b}{a_2}\right)\sum_{j \in C_-^{-i}} A_{ij} + \rho(b - a_1) + (1-\rho)(a_2 - b)\log n + O(1)$$

$$= w_+ \sum_{j \in C_+^{-i}} A_{ij} + w_- \sum_{j \in C_-^{-i}} A_{ij} + \gamma + O(1) = A_i. w^* + \gamma + O(1).$$

Writing the above for all $i \in [n]$ in a vector notation, we obtain

$$\|z^* - (Aw^* + \gamma \mathbf{1}_n)\|_\infty = O(1).$$

$\square$

## G.2 Spectral Algorithm and Proof of Theorem 2

In this section, we present our spectral algorithm for the SBM that can emulate the genie. As discussed in Section 5, this requires taking an appropriate linear combination of eigenvectors such that the top two eigenvectors such that $c_1 u_1^* + c_2 u_2^*$ approximates $w^*$ in the $\ell_\infty$ norm. The vector $w^*$ is a block vector with entries $(w_+, w_-)$ on the locations of $C_+$ and $C_-$. Recall by Lemma G.2 that

$$w_+ = \log\left(\frac{a_1}{b}\right) \quad \text{and} \quad w_- = \log\left(\frac{b}{a_2}\right).$$

We first present a subroutine that finds these coefficients $(c_1, c_2)$. We will introduce the formal correctness of the subroutine later, but first we provide informal discussion as to how these coefficients are computed. Roughly speaking, both $u_1^*$ and $u_2^*$ also have a block structure, and therefore, finding $(c_1, c_2)$ just corresponds to solving a system of $2 \times 2$ linear equations. Also, the coefficients do not depend on the locations of $\sigma^*$ with $+1$ or $-1$ labels, so exploiting this fact we just do calculation as if $C_+$ is on the first $\lfloor \rho n \rfloor$ vertices and compute the proxy for actual $A^*$. This results into the following subroutine.

---

**Algorithm 4** Find Linear Combination Coefficients

**Input:** The parameter set $(\rho, a_1, a_2, b)$ such that $a_1 a_2 \neq b^2$ (Rank-2) and the graph size $n$.

**Output:** The desired linear combination $(c_1, c_2)$.

1: Let $S \subseteq [n]$ such that $S = \{i : i \leq \rho n\}$ and compute the matrix $B \in \mathbb{R}^{n \times n}$ such that

$$B_{ij} = \begin{cases} a_1 \log n/n & \text{if } i, j \in S; \\ b \log n/n & \text{if } i \in S \text{ but } j \notin S; \\ a_2 \log n/n & \text{if } i \notin S \text{ and } j \notin S. \end{cases}$$

2: Compute the two eigenpairs $(\tilde{\lambda}_1, \tilde{v}_1)$ and $(\tilde{\lambda}_2, \tilde{v}_2)$ of $B$ (note that $B$ is rank-2).
3: Let $w^* \in \mathbb{R}^n$ be the block vector such that:

$$w_i^* = \begin{cases} w_+ = \log(a_1/b), & \text{if } i \in S; \\ w_- = \log(b/a_2) & \text{if } i \notin S. \end{cases}$$

4: Return $(c_1, c_2) \in \mathbb{R}^2$ that satisfies

$$c_1\left(\frac{\tilde{v}_1}{\tilde{\lambda}_1}\right) + c_2\left(\frac{\tilde{v}_2}{\tilde{\lambda}_2}\right) = w^*. \qquad (31)$$

Both $\tilde{v}_1$ and $\tilde{v}_2$ are block-vectors and they are linearly independent. Thus, Finding $(c_1, c_2)$ corresponds to solving a system of $2 \times 2$ linear equations.

---

We note that in the rank-2 case when $a_1 a_2 \neq b^2$, the vectors $\tilde{v}_1$ and $\tilde{v}_2$ have block structure and are linearly independent. Therefore, it is possible to span any vector with block structure, and in particular, even $w^*$. We now propose our spectral algorithm in the rank-2 case.

---

**Algorithm 5** Spectral recovery algorithm for SBM (Rank-2)

---

**Input:** An $n \times n$ observation matrix $A$ and parameters $(\rho, a_1, a_2, b)$ such that $a_1 a_2 \neq b^2$. Optionally, side information $y \in \mathcal{Y}^n$ such that we can compute the likelihood laws $\mathcal{S}_+$ and $\mathcal{S}_-$.

**Output:** An estimate of community assignments $\hat{\sigma}_{\text{spec}}$.

1: **Compute leading eigenpairs.** Compute the top eigenpair of $A$, denoted by $(\lambda_1, u_1)$, where $|\lambda_1| \geq \cdots \geq |\lambda_n|$.
2: **Compute coefficients of linear combination.** Run Algorithm 4 to find $(c_1, c_2)$ and
$$\gamma := (\rho(b - a_1) + (1 - \rho)(a_2 - b)) \log n.$$
3: **Compute spectral scores.** For any $s = (s_1, s_2) \in \{\pm 1\}^2$, prepare the spectral score vectors
   - No side information:
   $$z^{(s)} = s_1 c_1 u_1 + s_2 c_2 u_2 + \gamma \mathbf{1}_n.$$
   - Side information:
   $$z^{(s)} = s_1 c_1 u_1 + s_2 c_2 u_2 + \gamma \mathbf{1}_n + \log\left(\frac{\mathcal{S}_+}{\mathcal{S}_-}(y)\right).$$
4: **Remove sign ambiguity.** For each $s \in \{\pm 1\}^2$, let $\hat{\sigma}^{(s)} = \text{sgn}(z^{(s)})$.
   - No side information: Return $\hat{\sigma}_{\text{spec}} = \arg\max_{\{\hat{\sigma}^{(s)}: s \in \{\pm 1\}\}} \mathbb{P}(\sigma^* = \hat{\sigma}^{(s)} \mid A)$.
   - BEC or BSC side information: Return $\hat{\sigma}_{\text{spec}} = \arg\max_{\{\hat{\sigma}^{(s)}: s \in \{\pm 1\}\}} \mathbb{P}(\sigma^* = \hat{\sigma}^{(s)} \mid A, y)$.

---

Finally, when $a_1 a_2 = b^2$, from Lemma G.2 we have $w_+ = w_- = \log\left(\frac{a_1}{b}\right)$. In this case, we can just use the deterministic vector along $\mathbf{1}_n$ to emulate the genie-score vector. Strictly speaking, in this degenerate case, we do not even need a spectral strategy to achieve optimality. Despite this, we just refer to this as a spectral algorithm in the rest of the analysis for simplicity of exposition.

---

**Algorithm 6** (Spectral) Recovery algorithm for SBM (Rank-1)

---

**Input:** An $n \times n$ observation matrix $A$ and parameters $(\rho, a_1, a_2, b)$ such that $a_1 a_2 = b^2$. Optionally, side information $y \in \mathcal{Y}^n$ such that we can compute the likelihood laws $\mathcal{S}_+$ and $\mathcal{S}_-$.

**Output:** An estimate of community assignments $\hat{\sigma}_{\text{spec}}$.

1: Let
$$c := \log\left(\frac{a_1}{b}\right), \quad \gamma := (\rho(b - a_1) + (1 - \rho)(a_2 - b)) \log n.$$
2: Prepare the spectral score vector $z^{\text{spec}}$ as follows.
   - No side information:
   $$z^{\text{spec}} = cA\mathbf{1}_n + \gamma \mathbf{1}_n.$$
   - Side information:
   $$z^{\text{spec}} = cA\mathbf{1}_n + \gamma \mathbf{1}_n + \log\left(\frac{\mathcal{S}_+}{\mathcal{S}_-}(y)\right).$$
3: Return $\hat{\sigma}_{\text{spec}} = \text{sgn}(z^{\text{spec}})$.

---

We now show that the score vectors formed by these algorithms approximate the genie score vector $z^*$ in each case.

**Lemma G.3.** *Consider $\rho \in (0, 1)$ and $a_1, a_2, b > 0$. Let $(A, \sigma^*) \sim \text{SBM}_n(\rho, a_1, a_2, b)$ and condition on $\sigma^*$ such that $E$ from (13) holds. Optionally, let $y \sim \text{SI}(\sigma^*, \mathcal{S}_+, \mathcal{S}_-)$ for the channel laws $(\mathcal{S}_+, \mathcal{S}_-)$. Let $z^*$ be the genie score vector for the corresponding model. Then with probability $1 - o(1)$*

   - *Case $a_1 a_2 \neq b^2$: for some $s = (s_1, s_2) \in \{\pm 1\}^2$*
   $$\left\| z^* - z^{(s)} \right\|_\infty = o(\log n).$$

- *Case $a_1 a_2 = b^2$:*

$$\|z^* - z^{\text{spec}}\|_\infty = o(\log n)$$

*Proof.* Recall from Lemma 4.2, how the genie score changes in the presence of side information. In Algorithm 5 (Step 3) and Algorithm 6 (Step 2), this is precisely how the spectral score vectors are updated from the case when no side information is present. Therefore, it suffices to show that the score approximation holds in the case when no side information is present, which now will be the focus of the proof. The argument is exactly analogous to the one done used in Lemma F.3. We now discuss the rank 1 and rank 2 cases one by one.

**Rank-1 case**: $a_1 a_2 = b^2$. In this case, when no side information is present

$$\|z^* - z^{\text{spec}}\|_\infty = \|z^* - \log(a_1/b) A \mathbf{1}_n - \gamma \mathbf{1}_n\|_\infty = O(1),$$

where the last equation follows from Lemma G.2 and using that $\frac{a_1}{b} = \frac{b}{a_2}$.

**Rank-2 case**: $a_1 a_2 \neq b^2$. Fix $(s_1, s_2) \in \{\pm 1\}^n$ to be the signs for which the high probability event in Lemma E.4 holds. Define $w$ to be the vector from Lemma G.2 with entries $(w_+, w_-) = (\log(a_1/b), \log(b/a_2))$ on the locations of $C_+$ and $C_-$ respectively. In this case, using Lemma E.4, we will show that with probability $1 - o(1)$, we have

$$\|Aw^* - s_1 c_1 u_1 - s_2 c_2 u_2\|_\infty = o(\log n). \tag{32}$$

Combing this (32) along with Lemma G.2 implies the desired result: with probability $1 - o(1)$,

$$\begin{aligned}
\left\|z^* - z^{(s)}\right\|_\infty &\leq \|z^* - Aw^* - \gamma \mathbf{1}_n\|_\infty + \left\|Aw^* + \gamma \mathbf{1}_n - z^{(s)}\right\|_\infty \\
&= \|z^* - Aw^* - \gamma \mathbf{1}_n\|_\infty + \|Aw^* - s_1 c_1 u_1 - s_2 c_2 u_2\|_\infty \quad \text{(Step 3 of Algorithm 5)} \\
&= o(\log n).
\end{aligned}$$

It remains to show (32). Note that, we calculate $(c_1, c_2)$ in Algorithm 4 using the matrix $B$ where community sizes are exactly $\rho n$. But, condition on $E$, we have community sizes $(1 + o(1))\rho n$ and $(1 + o(1))(1 - \rho)n$. Also, in $A^*$, we have zero diagonal, where as, the matrix $B$ has diagonal entries of the order of $O(\log n/n)$. These changes only affect the eigenvalues by the multiplicative factor of $(1 + o(1))$ by Weyl's inequality. Therefore,

$$\lambda_1^* = (1 + o(1))\tilde{\lambda}_1 \text{ and } \lambda_2^* = (1 + o(1))\tilde{\lambda}_2.$$

Moreover, the entries of $u_1^*$ in the locations of $C_+$ are $(1 + o(1))$ factor of the entries of $\tilde{v}_1$ in the locations in $S$. By the same argument, one can say the same for $u_1^*$ in locations of $C_-$ and $\tilde{v}_1$ in locations of $[n] \setminus S$, and also for $u_2^*$ and $\tilde{v}_2$. From the way we calculate $(c_1, c_2)$ in (31), we have

$$c_1 \left(\frac{\tilde{v}_1}{\tilde{\lambda}_1}\right) + c_2 \left(\frac{\tilde{v}_2}{\tilde{\lambda}_2}\right) = w^*.$$

Then the above discussion implies

$$w_+ = (1 + o(1)) \left(\frac{c_1 u_{1,i}^*}{\lambda_1^*} + \frac{c_2 u_{2,i}^*}{\lambda_2^*}\right), \text{ for } i \in C_+ \text{ and } w_- = (1 + o(1)) \left(\frac{c_1 u_{1,i}^*}{\lambda_1^*} + \frac{c_2 u_{2,i}^*}{\lambda_2^*}\right), \text{ for } i \in C_-.$$

Finally using Lemma E.4, with probability $1 - o(1)$

$$\begin{aligned}
\|Aw^* - s_1 c_1 u_2 - s_2 c_2 u_2\|_\infty &\leq \left\|Aw^* - c_1 \frac{Au_1^*}{\lambda_1^*} - c_2 \frac{Au_2^*}{\lambda_2^*}\right\|_\infty + o(1) \\
&= \left\|(1 + o(1)) \left(c_1 \frac{Au_1^*}{\lambda_1^*} + c_2 \frac{Au_2^*}{\lambda_2^*}\right) - c_1 \frac{Au_1^*}{\lambda_1^*} - c_2 \frac{Au_2^*}{\lambda_2^*}\right\|_\infty + o(1) \\
&= o(1) \left(\max_{i \in [n]} \|A_{i\cdot}\|_1\right) \cdot \left(\left\|\frac{c_1 u_1^*}{\lambda_1^*}\right\|_\infty \vee \left\|\frac{c_2 u_2^*}{\lambda_2^*}\right\|_\infty\right) \\
&= o(\log n),
\end{aligned}$$

where the last Step follows by using Lemma G.1 and $(c_1, c_2)$ are chosen in such a way that the entries of vectors $\frac{c_1 u_1^*}{\lambda_1^*}$ and $\frac{c_2 u_2^*}{\lambda_2^*}$ are $O(1)$ by (31). $\qquad\square$

We finally combine all the pieces and give the proof of Theorem 2.

*Proof of Theorem 2.* We break into cases.

1. Rank 1, i.e. $a_1 a_2 = b^2$: First note that Algorithm 6 creates a score vector $z^{\text{spec}} \in \mathbb{R}^n$ such that $\|z^* - z^{\text{spec}}\|_\infty = o(\log n)$ by Lemma G.3. Using Lemma D.1, whenever $I^* >$, there exists $\varsigma > 0$ such that

$$\mathbb{P}(\sigma_i^* z_i^{\text{spec}} \leq \varsigma \log n) = o(n^{-1}).$$

Taking a union bound,

$$\mathbb{P}(\forall i \in [n],\, \sigma_i^* z_i^{\text{spec}} > \varsigma \log n) = 1 - o(1).$$

Finally, the algorithm outputs $\hat{\sigma}_{\text{spec}} = \text{sgn}(z^{\text{spec}})$ (Step 3), this immediately implies

$$\mathbb{P}(\hat{\sigma}_{\text{spec}} = \sigma^*) = 1 - o(1).$$

2. Rank 2, i.e. $a_1 a_2 \neq b^2$: We first note that whenever $I^* > 1$, the estimator $\hat{\sigma}_{\text{MAP}}$ achieves exact recovery by Proposition 2.10. That is with high probability, we have $\hat{\sigma}_{\text{MAP}} = \sigma^*$, unless $a_1 = a_2$ and no side information, in which case $\hat{\sigma}_{\text{MAP}} \in \{\pm\sigma^*\}$. Recall that Algorithm 5 creates four candidates for $\sigma^*$ and chooses the one with maximum posterior probability. Due to statistical achievability, it suffices to show that one of the $\{\sigma^{(s)} : s = (s_1, s_2) \in \{\pm 1\}^2\}$ maintained by the algorithm is $\sigma^*$ with high probability. To this end, recall Lemma G.3 that with probability $1 - o(1)$,

$$\min_{s^* \in \{\pm 1\}} \left\| z^* - z^{(s)} \right\|_\infty = o(\log n).$$

Combining this with Lemma D.1, there exists $s^* \in \{\pm 1\}^2$ and $\varsigma > 0$ such that

$$\mathbb{P}\left( \min_{i \in [n]} \sigma_i^* z_i^{(s^*)} > \varsigma \log n \right) = 1 - o(1),$$

after taking a union bound. As $\hat{\sigma}^{(s^*)} = \text{sgn}(z^{(s^*)})$ in Step 4, we obtain $\hat{\sigma}^{(s^*)} = \sigma^*$ with high probability. Overall, we established that $\hat{\sigma}_{\text{spec}}$ achieves exact recovery above the IT threshold.

$\square$

### G.3  DEGREE-PROFILING ALGORITHM FOR BEC AND BSC CHANNELS

**Algorithm 7** Degree-Profiling algorithm for SBM in the presence of BEC or BSC side information.

**Input:** An $n \times n$ observation matrix $A$ and parameters $(\rho, a_1, a_2, b)$. The BEC side information $y$ with parameter $\epsilon$ *or* BSC side information $y$ with parameter $\alpha$.

**Output:** An estimate of community assignments $\hat{\sigma}_{\text{dp}}$.

1: Let $S_+ := \{i : y_i = +1\}, S_- := \{i : y_i = -1\}$, and $\gamma := (\rho(b - a_1) + (1 - \rho)(a_2 - b)) \log n$. Compute $z \in \mathbb{R}^n$ such that, for every $i \in [n]$

$$z_i = \log\left(\frac{a_1}{b}\right) \sum_{j \in S_+} A_{ij} + \log\left(\frac{b}{a_2}\right) \sum_{j \in S_-} A_{ij} + \gamma.$$

2: Prepare the degree-profile score vector $z^{\text{dp}}$ as follows.
   - BEC side information: For any $i \in [n]$,

$$z_i^{\text{dp}} = \begin{cases} z_i & \text{if } y_i = 0; \\ +\infty, & \text{if } y_i = +1; \\ -\infty & \text{if } y_i = -1; \end{cases}$$

   - BSC side information:

$$z^{\text{dp}} = z + \log\left(\frac{1 - \alpha}{\alpha}\right) y$$

3: Return $\hat{\sigma}_{\text{dp}} = \text{sgn}(z^{\text{dp}})$.

The success of the degree-profiling algorithm is formalized in the following lemma.

**Theorem 4.** *Let $\rho \in (0,1)$ and $a_1, a_2, b > 0$. Let $(A, \sigma^*) \sim \mathsf{SBM}_n(\rho, a_1, a_2, b)$. Let $y \sim \mathsf{BEC}(\sigma^*, \epsilon)$ or $y \sim \mathsf{BSC}(\sigma^*, \alpha)$, where*

$$\lim_{n \to \infty} \frac{\log(1/\epsilon)}{\log n} = \beta \text{ and } \lim_{n \to \infty} \frac{\log(\frac{1-\alpha}{\alpha})}{\log n} = \beta$$

*for some $\beta > 0$. Then $I^*$ from (6) is well-defined and there is a degree-profiling algorithm (Algorithm 7) that returns the estimator $\hat{\sigma}_{\mathrm{dp}}$ which achieves exact recovery whenever $I^* > 1$.*

Again to prove that our approximation is good enough when using $S_+$ and $S_-$ as proxies for the actual communities, we will need a technical lemma to bound the error terms, whose proof we include the at the end of this section.

**Lemma G.4.** *Consider $\rho \in (0,1)$ and $a_1, a_2, b > 0$. Let $(A, \sigma^*) \sim \mathsf{SBM}_n(\rho, a_1, a_2, b)$. Condition on $\sigma^*$ such that the event $E$ from (13) holds. Fix any set $T \subset C_+$ or $T \subset C_-$ such that $|T| = O(n/\log^{10} n)$. Then for any $i \in [n]$, define $Y_i := \sum_{j \in T} A_{ij}$. Then*

$$\mathbb{P}\left(\forall i \in [n] : Y_i \leq \frac{\log n}{\log \log n}\right) \geq 1 - O(n^{-3}).$$

Using this, we now show that the degree-profiling scores are indeed good approximations of the actual genie scores.

**Lemma G.5.** *Fix $\rho \in (0,1)$ and $a_1, a_2, b > 0$. Let $(A, \sigma^*) \sim \mathsf{SBM}_n(\rho, a_1, a_2, b)$ and condition on $\sigma^*$ such that $E$ from (13) holds. For $\beta > 0$, let $y \sim \mathsf{BEC}(\sigma^*, \epsilon)$ or $y \sim \mathsf{BSC}(\sigma^*, \alpha)$ where $\epsilon$ and $\alpha$ scales as described in Theorem 4. Let $z^*$ and $z^{\mathrm{dp}}$ respectively be the genie score vector and the degree-profiling score vector produced by Algorithm 7 for the corresponding model of side information. Then with probability $1 - o(1)$,*

$$\left\| z^* - z^{\mathrm{dp}} \right\|_\infty = O\left(\frac{\log n}{\log \log n}\right).$$

*Proof.* The proof has similar calculations as in Lemma F.6 for ROS. We again first start by noting that $z^{\mathrm{dp}}$ is just formed by overriding the entries of $z$ from Step 1 of Algorithm 7 with $+\infty$ or $-\infty$ depending on the side information label being $+1$ or $-1$ under BEC channel. Also, in Step 2 under the BSC channel, we have $z^{\mathrm{dp}} = z + \log\left(\frac{1-\alpha}{\alpha}\right) y$. Recall Lemma 4.2 as to how the genie scores change under both types of side information. It suffices to show that, with probability $1 - o(1)$,

$$\| z' - z \|_\infty = O\left(\frac{\log n}{\log \log n}\right),$$

where $z'$ is the genie score vector without side information and $z$ is formed in Step 1. Recall that $E_1$ holds with probability $1 - O(n^{-3})$ from (24). Using Lemma G.2 along with the triangle inequality we obtain the following.

- BEC side information:

$$\| z' - z \|_\infty = \max_{i \in [n]} |z'_i - z_i|$$

$$\leq \max_{i \in [n]} \left| \log\left(\frac{a_1}{b}\right) \sum_{j \in C_+ \backslash S_+} A_{ij} + \log\left(\frac{b}{a_2}\right) \sum_{j \in C_- \backslash S_-} A_{ij} \right| + O(1)$$

where we substitute $z'$ from Step 1. From Lemma F.4, both $|C_+ \backslash S_+|$ and $|C_- \backslash S_-|$ are bounded by $O(n/\log^{10} n)$ with probability $1 - o(1)$. These sets are independent of $A$ and are chosen based on $y$. Using Lemma G.4 for these sets, with probability $1 - o(1)$

$$\| z' - z \|_\infty \leq \max_{i \in [n]} \left| \log\left(\frac{a_1}{b}\right) \sum_{j \in C_+ \backslash S_+} A_{ij} \right| + \left| \log\left(\frac{b}{a_2}\right) \sum_{j \in C_- \backslash S_-} A_{ij} \right| + O(1) = O\left(\frac{\log n}{\log \log n}\right).$$

- BSC side information: There are additional error terms caused by the sets $S_+ \setminus C_+$ and $S_- \setminus C_-$:

$$\|z' - z\|_\infty = \max_{i \in [n]} |z'_i - z_i| = \max_{i \in [n]} \left| \log\left(\frac{a_1}{b}\right) \sum_{j \in C_+ \setminus S_+} A_{ij} - \log\left(\frac{a_1}{b}\right) \sum_{j \in S_+ \setminus C_+} A_{ij} \right.$$

$$\left. + \log\left(\frac{b}{a_2}\right) \sum_{j \in C_- \setminus S_-} A_{ij} - \log\left(\frac{b}{a_2}\right) \sum_{j \in S_- \setminus C_-} A_{ij} \right| + O(1)$$

(substituting $z'$ from Lemma G.2 and $z$ from Step 1)

$$= O(1) \cdot \max_{i \in [n]} \left| \sum_{j \in C_+ \setminus S_+} A_{ij} + \sum_{j \in C_- \setminus S_-} A_{ij} \right| + O(1).$$

(since $S_+ \setminus C_+ = C_- \setminus S_-$ and $S_- \setminus C_- = C_+ \setminus S_+$)

Exactly the same argument of using Lemma F.4, both $|C_+ \setminus S_+|$ and $|C_- \setminus S_-|$ is $O(n / \log^{10} n)$ with probability $1 - o(1)$. Using Lemma G.4 for these sets, with probability $1 - o(1)$

$$\|z' - z\|_\infty = O\left(\frac{\log n}{\log \log n}\right).$$

$\square$

We finally prove Theorem 4.

*Proof of Theorem 4.* We already discussed in Appendix C that $I^*$ is well-defined. When $\beta > 0$, by Lemma G.5, with probability $1 - o(1)$,

$$\left\| z^* - z^{\mathrm{dp}} \right\|_\infty = O(1).$$

Above the IT threshold by Lemma D.1 and union bound over $i \in [n]$, there exists $\delta > 0$ such that

$$\mathbb{P}\left( \min_{i \in [n]} \sigma_i^* z_i^* > \delta \log n \right) = 1 - o(1).$$

Taking a union bound of these two events, there exists $\varsigma > 0$ such that

$$\mathbb{P}\left( \min_{i \in [n]} \sigma_i^* z_i^{\mathrm{dp}} > \varsigma \log n \right) = 1 - o(1).$$

Observing $\hat{\sigma}_{\mathrm{dp}} = \mathrm{sgn}(z^{\mathrm{dp}})$ in Step 3 of Algorithm 7, we obtain $\hat{\sigma}_{\mathrm{dp}}$ achieves exact recovery, i.e.

$$\lim_{n \to \infty} \mathbb{P}\left( \hat{\sigma}_{\mathrm{dp}} \text{ succeeds} \right) = 1.$$

$\square$

We finally return to the skipped proof of a technical lemma.

*Proof of Lemma G.4.* Fix any set $T$ according to the lemma and define $\{Y_i : i \in [n]\}$. Let $\tau = \max\{a_1, a_2, b\}$ and $Y \sim \mathrm{Binom}(|T|, \frac{\tau \log n}{n})$. Observe that for any $i \in [n]$, we have $Y_i$ is stochastically dominated by $Y$. Applying the following Chernoff bound for Binomial random variable (Mitzenmacher & Upfal, 2017, Thereom 4.4): for any $r > 1$ and $X \sim \mathrm{Binom}(N, p)$, $\mathbb{P}(X \geq rnp) \leq (e/r)^{rnp}$, we obtain for any $i \in [n]$

$$\mathbb{P}\left( Y_i \leq \frac{\log n}{\log \log n} \right) \leq \mathbb{P}\left( Y \leq \frac{\log n}{\log \log n} \right) \leq \left( \frac{e|T|\tau \log n / n}{\log n / \log \log n} \right)^{\frac{\log n}{\log \log n}}$$

$$= O\left( \frac{\log \log n}{\log^{10} n} \right)^{\frac{\log n}{\log \log n}} = n^{-10 + o(1)}.$$

A simple union bound over all $i \in [n]$ gives us

$$\mathbb{P}\left( \forall i \in [n] : Y_i \leq \frac{\log n}{\log \log n} \right) \geq 1 - O(n^{-3}).$$

$\square$

