**Definition 2.9** (MAP Estimator). *Consider the observation matrix $A$ and the side information $y$ (either $\mathsf{BSC}$ or $\mathsf{BEC}$). We define the Maximum A Posteriori (MAP) estimator as*

$$\hat{\sigma}_{\mathrm{MAP}} = \arg\max_{\sigma \in \{\pm 1\}^n} \mathbb{P}(\sigma^* = \sigma \mid A, y).$$

*When no side information is present, define $\hat{\sigma}_{\mathrm{MAP}} = \arg\max_{\sigma \in \{\pm 1\}^n} \mathbb{P}(\sigma^* = \sigma \mid A)$.*

## 3 MAIN RESULTS

Before stating our algorithmic result, we describe the information-theoretic limit due to Dreveton et al. (2024) in our notation.

### 3.1 Information Theoretic Threshold from Dreveton et al. (2024)

Define the *Chernoff coefficient* across the pair of communities as

$$\text{CH}_t(+, -) = (1 - t) \left[ \rho\, \text{D}_t(\mathcal{P}_+ \| \mathcal{Q}) + (1 - \rho)\text{D}_t(\mathcal{Q} \| \mathcal{P}_-) + \frac{1}{n}\text{D}_t(\mathcal{S}_+ \| \mathcal{S}_-). \right]$$

Here $\text{D}_t(\mathcal{A} \| \mathcal{B})$ is the is the Rényi divergence of order $t$ between any two laws $(\mathcal{A}, \mathcal{B})$, such that they are both continuous or discrete, is given by

$$\text{D}_t(\mathcal{A} \| \mathcal{B}) := \frac{1}{(t - 1)} \log \mathbb{E}_{x \sim \mathcal{B}} \left[ \left( \frac{\mathcal{A}(x)}{\mathcal{B}(x)} \right)^t \right]. \tag{3}$$

Define the limit $L : (0, 1) \to \mathbb{R}_{\geq 0} \cup \{+\infty\}$ by

$$L(t) = \lim_{n \to \infty} \frac{n}{\log n} \text{CH}_t(+, -). \tag{4}$$

We restrict ourselves to the laws such that $L(t)$ is well defined. Then the information theoretic limit is characterized by

$$I^* = \sup_{t \in (0,1)} L(t), \tag{5}$$

where by convention, we consider the supremum to be $+\infty$ if $L(t)$ is unbounded in $t \in (0, 1)$ or there exists $t \in (0, 1)$ such that $L(t) = +\infty$. Then following proposition is a minor variant of the two community case of (Dreveton et al., 2024, Theorem 1), characterizing the fundamental limit for exact recovery for the GBM.

**Proposition 3.1** (Dreveton et al. (2024)). *Let $\rho \in (0, 1)$ and $(\mathcal{P}_+, \mathcal{P}_-, \mathcal{Q})$ be probability laws. Let $(A, \sigma^*) \sim \text{GBM}_n(\rho, \mathcal{P}_+, \mathcal{P}_-, \mathcal{Q})$ and we observe $A$. Optionally, for side information laws $(\mathcal{S}_+, \mathcal{S}_-)$, we observe $y \sim \text{SI}(\sigma^*, \mathcal{S}_+, \mathcal{S}_-)$. We assume the laws are such that $I^*$ in (5) is well-defined. Then*

1. *If $I^* > 1$, then the Maximum A Posteriori estimator $\hat{\sigma}_{\text{MAP}}$ achieves exact recovery.*

2. *Additionally, if $I^* < 1$ and the function $L(t)$ is strictly concave then no estimator $\hat{\sigma}$ achieves exact recovery.*

This variant follows from (Dreveton et al., 2024, Theorem 1) by observing that the success of MAP proof does not rely on the strict concavity of $L(t)$ and the proof goes through even if $I^* = +\infty$. The impossibility result is completely equivalent to theirs. We discuss the necessary proof changes in Appendix A.3. We note that the case of no side information can be realized by considering side information laws $(\mathcal{S}_+, \mathcal{S}_-)$ that always deterministically output 0; in this case $\text{D}_t(\mathcal{S}_+ \| \mathcal{S}_-) = 0$.

### 3.2 Algorithmic Achievability for ROS and SBM

The main results of the paper is to design an optimal spectral algorithm for ROS and SBM.

**Theorem 1.** *Fix $\rho \in (0, 1)$ and $a, b \in \mathbb{R}$ such that $\max\{|a|, |b|\} > 0$. Let $(A, \sigma^*) \sim \text{ROS}_n(\rho, a, b)$. Optionally, consider a side information channel $y \sim \text{SI}(\sigma^*, \mathcal{S}_+, \mathcal{S}_-)$ such that $I^*$ in (5) is well-defined. Then there is a spectral algorithm (Algorithm 2) that returns the estimator $\hat{\sigma}_{\text{spec}}$ which achieves exact recovery, whenever $I^* > 1$.*

**Theorem 2.** *Let $\rho \in (0, 1)$ and $a_1, a_2, b > 0$. Let $(A, \sigma^*) \sim \text{SBM}_n(\rho, a_1, a_2, b)$. Optionally, consider a side information channel $y \sim \text{SI}(\sigma^*, \mathcal{S}_+, \mathcal{S}_-)$ such that $I^*$ in (5) is well-defined. Then there is a spectral algorithm (Algorithms 5 and 6) that returns the estimator $\hat{\sigma}_{\text{spec}}$ which achieves exact recovery, whenever $I^* > 1$.*

In Appendix C, we will verify that for both SBM and ROS, the function $L(t)$ (and thus also $I^*$) is well-defined and $L(t)$ is strictly concave for each (i) no side information (ii) GF, BEC, and BSC channels. This along with Theorems 1, 2 and Proposition 3.1 establishes that the spectral algorithm succeeds up to the information-theoretic limit in these settings.

# 4 GENIE-AIDED ESTIMATORS

We begin our analysis by defining the framework of genie-aided estimation (see e.g., Abbe (2017)). Our contribution is a systematic method of connecting spectral algorithms to genie estimators. In the genie-aided setting, we suppose that all labels but the $i^{\text{th}}$ are known, and the goal is to determine the $i^{\text{th}}$ label. More formally, let $\sigma^*_{-i}$ denote the true labels, apart from $\sigma^*_i$. The optimal estimator for the $i^{\text{th}}$ label is given by

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

It is known that the asymptotic information-theoretic threshold does not shift for the BEC and BSC channels unless $\epsilon = O(n^{-\beta})$ and $\alpha = O(n^{-\beta})$ for some $\beta > 0$ (Dreveton et al., 2024; Saad & Nosratinia, 2018). Therefore, the side information $y$ already satisfies almost exact recovery criterion, recovering $(1 - o(1))$ labels correctly. Hence, we have $|C_+ \Delta S_+|, |C_- \Delta S_-| = o(n)$ with high probability. We then show that for this proxy choice of scores $z^{\mathrm{dp}}$, we indeed have $\|z^{\mathrm{dp}} - z^*\|_\infty = o(\log n)$, and thus, the degree-profiling algorithm succeeds down to the shifted threshold. The formal theorems and algorithms can be found in Appendix F.3 and G.3.

**Remark 5.1.** *We emphasize that the degree profiling algorithm has an important caveat that it would fail to recover labels exactly from a tuple $(A, y)$, when side information strength is "weak" or completely absent, even though the recovery was possible just from $A$. To overcome this, one has to rely on the signal from $A$ to get preliminary almost exactly correct labels, rather than just trusting the side information $y$. This exactly corresponds to the two-stage strategies described in Section 1.2. Our spectral algorithm in just one stage recover all the labels correctly from $(A, y)$ (including no side information), whenever it is possible to do so, and that too for any side information channel (under the technical assumption that $I^*$ in (5) is well-defined which is also needed for the MAP success).*

## 5.2 SPECTRAL ALGORITHM

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

- **More general settings:** To design spectral algorithms for more than two communities and more general observation distributions $(\mathcal{P}_+, \mathcal{P}_-, \mathcal{Q})$ from a class of exponential families (also see details in Appendix A.1).

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

# Appendices

## A DEFERRED DISCUSSIONS

### A.1 FUTURE EXTENSIONS AND PRIOR WORKS ON SPECTRAL ALGORITHMS

**Future Extensions.** We now describe, how we can generalize the spectral algorithm to more than two communities and more general distribution families. The entire framework of genie-aided estimation naturally generalizes to the multi-community case. We note that the entrywise eigenvectors behaviors of Abbe et al. (2020) for top eigenvectors continue to hold. Despite this, (Abbe et al., 2020, Appendix C.4) noted difficulties in designing spectral algorithms for more than two blocks due to the multiplicity of eigenvalues and eigenvectors being only computed up to a rotation. But we note that, except for these degenerate cases (some measure zero subset of parameters where exact recovery is possible), the algorithm should be able to emulate the genie-aided estimation and achieve optimality. For the degenerate cases noted in Abbe et al. (2020), it remains an interesting open question if we can design a spectral estimator.

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

$\square$