# OpenReview forum: "Exact Community Recovery under Side Information: Optimality of Spectral Algorithms"
_ICLR.cc/2025/Conference — ICLR 2025 Poster_

### Official Review · Reviewer_Zdp2 · 2024-10-17

**Soundness:** 3
**Presentation:** 3
**Contribution:** 3
**Rating:** 6
**Confidence:** 4

**Summary:**

This paper addresses the problem of exact community recovery in two-community block models with node-attributed side information. The primary contribution is the design of an optimal spectral algorithm that achieves the information-theoretic limits in various settings. The general block model in this paper encompass  rank-one spike Gaussian model and SBM, and the side information incorporates GMM and two types of "noisy'' labels.

**Strengths:**

1. The paper is well-organized, and the arguments flow logically.

2. The claims are solid, and the proposed framework covers a broad range of interesting models.

3. The results provide theoretical insights into community detection with side information and demonstrate the optimality of spectral methods.

**Weaknesses:**

1. Some related literature is missing, e.g., [1]. Specifically, the SBM model with $\rho = 1/2$ and Gaussian Features aligns with Section 4 in [1] and should be cited and compared. Additionally, for the SBM with GF where $\rho = 1/2$ and $a_1 = a_2 = a$, the expression for $I^*$ takes a simple form in Equation (4.4) of [1]. Providing explicit forms of $I^*$ for other models would enhance interpretability of the results.

2. For SBM with BEC/BSC, the algorithms assume prior knowledge of parameters $(\rho,\varepsilon, \alpha)$. The paper should mention this clearly and discuss why it's reasonable to assume this prior knowledge, or suggest how these parameters could be estimated. From a theory standpoint, achieving optimality might require paying a cost for adapting to these parameters, especially in regimes $\varepsilon = n^{-\beta}$ and $\alpha = n^{-\beta}$ for $\beta>0$.

3. The two-community assumption limits the algorithm’s practical use. While I understand that the optimality of the spectral algorithm depends on this assumption, the statement in Appendix A, "The entire framework of genie-aided estimation naturally generalizes to the multi-community case," is confusing. Can the authors elaborate on how the spectral algorithm could be modified, for example, to handle $K=3$ communities?

4. There are no experiments. It would be helpful to include numerical experiments demonstrating the algorithm’s performance on the specific models considered.

[1] Abbe, E., Fan, J., & Wang, K. (2022). An $\ell_p$ theory of PCA and spectral clustering. The Annals of Statistics, 50(4), 2359-2385.

**Questions:**

See weaknesses above. Also, the paper could benefit from a careful proofreading. For example:

1. Gaussain -> Gaussian on page 3.
2. In Algorithm 7,  $\alpha_n$ is not defined. Should it be $\alpha$?
3. In Algorithm 6, the input section ends with two periods.

---

> ### Author Response · Authors · 2024-11-22
> **Response to the review**
>
> We thank the reviewer for a thoughtful review and their constructive feedback on the work! Below we address the weaknesses pointed out in order.
> 1. Indeed, the reference of [Abbe, Fan, Wang 2022] is an important one, which we missed. We will include it and compare our work. As mentioned in the common response, we will include the simplified expressions of $I^*$ whenever possible to enhance interpretability.
> 2. Please see the common response where we elaborate how the distribution parameters can be estimated from $A$ and $y$. It is not clear how to estimate $\alpha$ in the BSC channel if it is not already given.
> 3. Please see the common response for details on the multi-community extension.
> 4. Yes, as mentioned in the common response, we will include numerical simulations verifying the optimality of spectral algorithms.
>
> Thank you for catching some typos! We will fix all of them and carefully proof-read the final version

---

### Official Review · Reviewer_dGJq · 2024-11-03

**Soundness:** 2
**Presentation:** 3
**Contribution:** 2
**Rating:** 3
**Confidence:** 4

**Summary:**

The paper studies block models with two communities, where each community has a general form of edge distribution. Side information on nodes is also considered. The authors specifically focuses on the case of Bernoulli and Gaussian distributions. The paper then proposes a variant of spectral algorithm to recover the community structure in the model, and provides statistical bounds for exact recovery of the signal.

**Strengths:**

- The paper is well organized and easy to follow. The notations and definitions are mostly self contained.
- The authors provide the statistical bounds for exact recovery in the case of the Rank One Spike (ROS) model and the Stochastic Blockmodel (SBM).

**Weaknesses:**

My biggest concern is that some of the language in the paper seems overclaiming. For example, the authors state that the paper proposes a "unified model" and "unified proof framework". The amount of novelty and contribution is unclear to me. Can the authors specify, preferably in bullet points, their new results, algorithm, bounds, proof techniques, or experiments?

Beyond that, I have some more detailed questions:
- The paper aims to provide a unified framework of analysis for at-most-two-community block models, but the results in Theorem 1 and 2 only target two specific and arguably "simpler" classic models. Why is that? Is it possible to give a bound based on the definition of P+, P-, and Q, for instance?
- Section 3.1: Does the information-theoretic bounds in Prop 3.1 contain anything novel, or just cited as is from [Dreveton et al., 2024]? If it is the latter, I strongly believe it should be moved to the preliminary section and not put in the "Main Results".
- Theorem 1 and 2: The exact recovery condition depends on I* > 1. Is there any way to calculate this efficiently? From a reader's perspective, I'd love to see many examples of two community models and know how their parameters are related to this I*. For example, can the authors relate I* to p and q in the SBM?
- The proofs like the concentration inequalities and the spectral analysis in the paper seem standard. In fact the union bound strategy feels similar to the ones in E. Abbe's and Dreveton's work. What are the technical contribution of this paper? Is there any proof technique that can be used in the future?

Minor:
- Bern() used but not defined on the first page.

**Questions:**

- I understand this is a theoretical work, but have the authors verified their algorithm on any dataset?
- What is the computational complexity in terms of n?
- Is it possible to generalize this work to the case of multiple communities? If not, what are the technical difficulties?

---

> ### Author Response · Authors · 2024-11-21
> **Response to the review**
>
> We thank the reviewer for their time in reviewing our submission and are happy to address their concerns!
>
> **Summary and Contributions**
> ---
> ---
> **New Results:** The main results of this work is to provide *optimal* spectral algorithms for the exact recovery problem under node-attributed side information in the two community block models with *Bernoulli* and *Gaussian* edge observations (lines 17-19, 84-89). The work of Dreveton et al. has characterized the IT limits and their proposed algorithm (requires edge observations are from exponential families) has no theoretical guarantees.
>
> ---
> **Algorithm:** The main challenge here is that there could be many ways to use side information, and we are interested in the one that optimally combines the signal from $A$ and the side information $y$, to achieve the new IT threshold. Our focus here is to design a spectral algorithm in particular. See lines 90-122 as to how different setting-specific choices have been adopted in designing spectral algorithms to achieve optimality e.g. (a) just sign thresholding 2nd eigenvectors for symmetric SBM [Abbe et al. 2020] (with adjacency matrix) and [Deng et al. 2021] (with laplacian) Vs carefully designing linear combinations of eigenvectors for PDS [Dhara et al. 2022b] (b) the special choice of encoding in the censored variant of the problem [Dhara el al. 2022a] and (c) combining eigenvectors of the two matrix representations of the same network to handle general censored SBM [Dhara et al. 2023]. Notably for the uncensored variant as we study here, even under no side information, the optimal spectral strategy was not clear; in particular it was not apparent whether one matrix representation suffices or whether we need two as in the censored counterpart [Dhara et al. 2023]?
>
> Designing optimal spectral algorithms is a delicate issue and the availability of side information makes it an interesting problem as to what is the optimal strategy to use it. See lines 116-122 for the auxiliary goal of this work. That brings us to our main technical novelty of the work.
>
> ---
>
> **The genie - spectral connections and proof technique:** The main novelty of the work is in rigorously establishing the connection between the spectral estimators and the genie aided estimators (the entire Sections 4 & 5). Note that these genie-aided estimators in Eq 6 are very powerful due to the availability of a genie that reveals all but the label that we are trying to estimate. We are able to design spectral strategies that can mimic the genie by approximating the genie scores in Eq (7), which in words is the log likelihood ratio of the $i^\mathrm{th}$ label being +1 and -1 given the edge and node information, and also the remaining labels (due to the genie). The spectral algorithm sketched in Algorithm 1 computes the statistics $z_{\textnormal{spec}}$ that approximates $z^*$ in $\ell_\infty$ norm (without the access of the genie) directly from the eigenvectors of $A$.
>
> As a consequence of this connection, we immediately have an optimal strategy to incorporate side information, mentioned in Eq (1). See also Lemma 4.2 and lines 368-377.
>
> ----
>
> We continue the response as a new comment below. We address the specific points raised by the reviewer in order.

---

> ### Author Response · Authors · 2024-11-21
> **Continuation of the response**
>
> * **Regarding ''unification'' and generality of the results:**
>
> As for the “unified proof framework” language choice: we mean the entire framework of genie-aided estimation (done generally) and its connection with the spectral algorithm (done for SBM and ROS, and general side information). See Figure 2 for a summary. We believe this is still very general compared to the prior works with algorithmic results. We further expand on the analysis of spectral algorithms to elaborate on the difficulty involved in further generalizing. The analysis has the following crucial steps.
>
> 1. **Genie scores and its success with margin:** We define the notion of genie scores (Section 4) and show that they have the correct sign with sufficient ``margin” above the IT threshold (Appendix D, D.1). This step has nothing to do with the spectral algorithm and hence derived more generally for any block model GBM (Definition 2.1), i.e. in terms of the definitions of the laws $P_{+}, P_{-}, Q$.
>
> 2. **Design of spectral algorithm and genie-spectral score approximation:** This is the step where we specialize in the SBM and ROS. In order to handle other models, we discuss a roadmap in Appendix A.1 (future extensions). There should be a possibility of handling more general distributions from the exponential families, where the log-likelihood is linear. (Note that even the algorithm of Dreveton et al. without any provable guarantees is also only for exponential families). However, the main bottleneck is the entrywise behavior of eigenvectors [Abbe et al. 20], which is crucial for the spectral algorithm and holds under some set of abstract assumptions A1-A4 (lines 1242-1252) in Appendix E. We verify these assumptions in Appendix E for ROS and SBM. As these assumptions are not simple and intuitive, we decide to just focus on ROS and SBM because (i) it already brings across the key idea behind the genie-spectral connection (ii) these already include important cases that have received their own special treatment in the literature: general two community SBM, $\mathbb{Z}_2$-synchronization, Submatrix Localization. The interesting thing is that we can handle general side information channels.
>
> ---
> * **Section 3.1:** That’s a good suggestion. The information-theoretic limit in Section 3.1 is stated in a slightly different way than from Dreveton et al., having made some simple observations. We do not consider them as the main results and will exclude them from Section 3.
>
> ----
> * **IT threshold expressions:** The expressions for $I^*$ when simplified, have simpler intuitions of SNR. Under no side information, according to the parameterization in Def 2.3, it is well-known that for SBM(½,$ a,a,b)$, the threshold $I^*=(\sqrt{a}-\sqrt{b})^2 /2$. For ROS($\rho, a, b$), the threshold is given by $I^*=(a-b)^2 (\rho a^2+(1-\rho)b^2)/8$ and also can be seen as SNR. Under side information channels, the additional term introduced $D_t (S_{+} \Vert S_{-})$ has an intuitive formula for GF, BEC, and BSC (derived in Appendix C). As alluded to before in the common response, we will include simplified expressions whenever possible to enhance the interpretability of the results. The algorithm is not generally concerned with calculating $I^*$. For the specific cases of models and side information considered, the formula of $I^*$ has closed-form expressions and hence can be computed. The algorithm is agnostic to the threshold and operates on the input $(A,y,$ model-parameters$)$; only whether the algorithm succeeds or not is characterized by the recovery threshold.
>
> ---
> * **Proof technique:** The main novelty in the proof technique is in the genie-spectral connection. Overall, we believe this insight should help us in the design and analysis of spectral algorithms going forward. See future extensions in Appendix A.1 and the discussion in Section 6.
>
> ----
> **Questions:** Below we answer the questions in order.
> 1. Please see the common response---we will include numerical simulations for SBM and ROS, with three side information channels. We have not performed experiments on real-world datasets. As the real-world networks may not really be satisfying the modeling assumptions of SBM, this may require delving into the robustness of the algorithm to the modeling assumptions, which is beyond the scope of this work. For synthetic networks that are really sampled according to our models, the algorithm should indeed succeed up to the sharp recovery threshold as we have theoretical guarantees.
> 2. Runtime: Please see the common response. Our algorithm has a ``nearly linear” runtime.
> 3. Multi-community extension: We refer to the common response for the detailed answer on this.

---

### Official Review · Reviewer_T3f8 · 2024-11-03

**Soundness:** 3
**Presentation:** 3
**Contribution:** 3
**Rating:** 6
**Confidence:** 3

**Summary:**

This paper addresses the problem of exact community recovery for general two-community block models with node-attributed side information. The authors consider both Gaussian and Bernoulli matrix models for the edge observations, as well as a general side information channel, including Gaussian Features and Binary Erasure Channel. Based on the recent work of Dreveton et al. (2024) on the information-theoretic limit on this problem, the authors demonstrate the algorithmic achievability using a spectral algorithm that incorporates side information with the eigenvectors of the observed edge matrix. The main technical novelty lies in establishing a rigorous connection between the spectral estimator and the genie-aided estimator by the results on the first-order approximation of the eigenvectors in Abbe et al. (2020).

**Strengths:**

The paper is well-structured, with clearly stated objectives and supported arguments. It provides background definitions and results on related models, which greatly aids in understanding the topic. The novel approach of connecting the eigenvectors to the genie-aided estimator is particularly intriguing, contributing to the design of an efficient spectral algorithm. Additionally, the rigorous mathematical derivations are well-organized in the appendix, offering solid theoretical support.

**Weaknesses:**

Due to the page limit, I feel that some important details have been deferred to the appendix. For instance, I would like to see a more rigorous discussion in the main context on the special form of the genie score vector, $z^* \approx Aw + \gamma I_n$; how $w$ can be derived from the model, any regularity conditions needed for this approximation, and whether this form extends to models beyond Gaussian and Bernoulli. I suggest moving part of the theoretical discussion into the main text and possibly relocating some of the preliminaries to the appendix.

 Additionally, I am uncertain about the robustness of this algorithm, as the coefficients $c_i$ relating $w$ to eigenvectors must be computed from model parameters. However, in practice, we typically observe only the adjacency matrix and side information, without access to underlying parameters like $a_1, a_2, b$ in ${\rm SBM}_n(\rho, a_1,a_2, b)$. A discussion or clarification on this would be beneficial.

  Another limitation of this paper is the lack of a numerical study. I suggest including simulation examples that compare the proposed algorithm’s performance (in terms of accuracy and computation cost) with the existing two-stage algorithms. This would strengthen the paper by demonstrating the practical advantages of this approach.

**Questions:**

1. On Page 6, $L(t)$ and $I^*$ are introduced to characterize the information theoretic limit. Are there any general sufficient conditions that ensure $L(t)$, and consequently $I^*$, are well-defined?

2. In Section 5.1, degree-profiling algorithm is mentioned as having certain caveat. However, it is unclear to me what this algorithm refers to. Perhaps adding a brief summary of the key points of this algorithm would improve clarity.

3. On page 9, it is stated that an approximate linear combination coefficients $c_i$ such that $w\approx \sum_{i=1}^K c_i/\lambda_i^* u_i^*$. Is there any intuitive explanation for why $w$ is approximately in the span of $u_i^*$? Is this result purely from the computations of the specific model (Gaussian and Bernoulli), or is there a broader rationale suggesting that this holds across more robust models?

4. Regarding the algorithm, do the parameters $c_i$ and $\gamma$ need be explicitly computed from the model parameters, or is there a possibility to estimate them using only the adjacency matrix $A$ and the side information?

5. This final question may not be directly related to the problem considered in the paper. Dreveton et al (2024) provides  the information theoretic limit of exact recovery in terms of $I^*$.  I am curious whether the regime $I^*<1$ can be further split. Specifically, could we identify a sub-regime where exact recovery is achievable even without the side information, and another sub-regime where side information is essential with some specific conditions?

---

> ### Author Response · Authors · 2024-11-22
> **Response to the review**
>
> We thank the reviewer for their time and the thorough review of the submission! Below we provide some clarifications on the weaknesses.
>
> ----
> * **Derivation of the genie score:** As we mentioned in the common response, we will include important details in the main text, in particular the derivation of $ z^* \approx Aw+ \gamma \mathbf{1}$. The expressions of genie scores will take this form for the distributions from the exponential families more generally. Roughly speaking, (i) $A$ corresponds to the sufficient statistic, (ii) the vector $w$ is a vector with block structure whose entries are the ratio of canonical parameters for $(P_+, Q)$ and $(Q, P_-)$ respectively (iii) the scalar $\gamma$ is some weighted combination (depends on $\rho$) of log-ratio of normalizing constants of $(P+,Q)$ and $(Q,P_{-})$. Therefore, the vector $w$ and $\gamma$ can be explicitly computed from the model parameters. We will include this discussion in the final version.
>
> ---
> * **Estimating model parameters:** See the common response where we describe a method to estimate the parameters when they are unknown. Specifically, for SBM and ROS, the estimated model parameters (calculated in this way from $A$) will be close to the true value with an additive error of $o(1)$ with high probability. This suffices for the spectral algorithm to still continue to succeed when we use the plugins.
>
> ----
> * **Numerical Study:** Please see the common response. We will include a simulation verifying that the spectral algorithm achieves the IT threshold. The comparison with two-stage approaches is beyond the scope of this work because: (i) Even the two-stage algorithm has nearly linear runtime and is a good algorithm. Our work does not aim to establish a practical advantage of the spectral algorithm over it (ii) Even if there is an advantage, the two-stage algorithm so far has been only presented for SBM and PDS with BEC and BSC channels [by Saad and Nosratinia 2018,20]. It is not too difficult to design the algorithm more generally given our framework. However, we strongly feel this should be done with more careful attention to the details along with theoretical analysis of its optimality, which is beyond the scope.
>
> ---
> Questions
> --
> 1. **Well-defined L(t)**: Note that the way we define $P_+, P_-, Q$ are the sequences of distributions that vary with $n$, e.g. this is the case in both SBM and ROS. We explicitly mention $L(t)$ is well-defined for rigor, to avoid having "weird" pathological sequences of distributions i.e. different distributions for odd and even $n$ s, etc; otherwise, we expect it to hold more generally, whenever the $P_+, P_-,Q$ converges in distribution as $n \to \infty$.
> 2. **The degree-profiling algorithm:** this algorithm is just using the side information $y$ as proxy for the genie given labels $\sigma_{-i}^*$. This is briefly described in lines 413-417. The caveat in Remark 5.1 essentially says that this strategy is not going to succeed in recovering labels, if side information is absent or "weak" even though the recovery is possible just from $A$, because it just relies on the signal from $y$ to attain the first preliminary labeling.
> 3. **w in the span of eigenvectors:**  This is a good question and the answer is subtle. When the rank of $A^*$ is the same as the number of communities, it is always possible to express $w \approx c_i u_i^*/\lambda_i^*$. This is because $w$ is a block vector with $K$ blocks and so are $u_i^*$ s. And due to the linear independence of $u_i^*$ s, all $K$-block vectors are in the span, including $w$. However, when rank$(A^*)<($# communities$)$, there is no such guarantee. In our cases, we were still able to achieve this though.
>
> (a) ROS: This is a rank-1 model. However, the genie linear combination vector $w$ is along the direction of $u_1^*$, and thus we are able to emulate the genie.
>
> (b) SBM rank-1: The case when $a_1a_2=b^2$. In this case, the genie vector $w$ is not along the direction of $u_1^*$ (contrary to ROS) but it turns out that $w$ is a deterministic vector that does not depend on the locations of $\sigma^*$. This allows us to still approximate the genie scores, though without a "spectral approach" strictly speaking i.e. without using the eigenvectors. See Algorithm 6 and lines 2077-2080, where we handle the rank-1 case separately.
>
> When rank$(A^*)<($# communities$)$, we are not sure if there is any general condition that ensures this happens. Thus, we critically put this in Requirement 2 in line 775-776 in future extensions in Appendix A.1.
>
> 4. Yes, as specified before, the parameters $\gamma, a_1, a_2, b $ can be estimated from the adjacency matrix with an additive error of $o(1)$, which suffices for the spectral algorithm to continue to succeed when used as a proxy.

---

> ### Author Response · Authors · 2024-11-22
> **Continuation of the response**
>
> 5. The region of $I^*<1$ corresponds to the impossibility region. Did you mean whether we can split the region $I^*>1$? Yes, we can let $I_0^*$ as the expression of IT threshold under no side information, and divide $I^*>1$ into two subregions $I^*_{0}>1$ and $\lim_{n \to \infty} \left( D_t(S_+ \Vert S_-)/\log n \right)  > 1-I_0^*$. More specifically, for ROS with BEC and BSC channels, there is a nice phase diagram which we will include in the updated version.

---

### Official Review · Reviewer_Appg · 2024-11-04

**Soundness:** 4
**Presentation:** 2
**Contribution:** 4
**Rating:** 8
**Confidence:** 3

**Summary:**

This paper considers the problem of recovery of two-community block
models. It sits in the context of a broad literature on detectability
limits, exact algorithms, etc. The particular setting considered by
the authors is one of a general sort of block-model random matrix
based on a given partition/distribution model, along with a vector of
"side-information" that is aligned with the block structure in some
way. The authors study the information theoretic limit of this
detection problem via simple spectral methods, and by studying
"genie-aided estimators."

**Strengths:**

This is a well-written paper -- there are quite a few moving parts
involved in this work, but the explanation was clear and easy to
follow. The authors ought to be commended for clear technical
communication. I do have onemajor misgiving about this, but that
is saved for the weaknesses section.

I am admittedly not too familiar with the literature on exact recovery
of SBMs beyond the paper of Abbe (2018), but the results here seem
quite good, and fit into the surveyed prior results nicely. I
appreciate how the recovery guarantees are accompanied by specific
algorithms, rather than proving detectability in merely an abstract
way.

**Weaknesses:**

My critique of this paper is not one aimed at the strength of the
results -- indeed, I found the results to be nice, and the exposition
to be quite good. However, I have the sense that the main body of the
paper was merely a rough overview of the "real paper" hidden in the
appendix. I understand that for space reasons, lots of ML papers have
to put substantial content in the supplementary material. I don't have
a deep argument for why this is the case, but I get the feeling that
this work is better-suited for a journal format. For instance, one of
the main constructed algorithms that attains exact recovery
(Algorithm 2) is only given in the supplementary material. It would be
more appropriate to structure the paper so that the algorithm is given
and explained in the body of the paper, with other variants possibly left
to the supplement.

I should note that although I am a reader of theory papers of this
sort, I have not reviewed many of them in the context of ML
conferences, so I am ultimately willing to defer to the other
reviewers on this point.

Other notes:

(I am not holding the following against you in any way -- deadlines
can be tough!)

Please check the paper for spleling mistakes -- I spotted a few such
as "Gaussain Features" and "shifing the eigenvector combination."

**Questions:**

My only concern with this paper has to do with the formatting issue raised in the weaknesses section. If this is a typical approach that is commonplace and accepted in ML conferences, then I am willing to cede that point.

---

> ### Author Response · Authors · 2024-11-21
> **Response to the review**
>
> We thank the reviewer for their quality feedback and acknowledging the key contributions of our work! We indeed faced some difficult choices due to the page limit and had to delegate some important points on the design of spectral algorithms to the appendix. As mentioned in the common response, we will adjust the presentation such that the key ideas involved in designing the spectral algorithms therein are well-highlighted in the main text. We also appreciate the feedback on the spelling errors, and will make sure to resolve them in the final version.

---

### Author Response · Authors · 2024-11-21
**Common Response**

We thank all the reviewers for their thorough feedback on our submission! Following their combined suggestions, we will improve our presentation. We will (i) include some crucial ideas involved in designing the spectral algorithms in the main text (ii) discuss the simplified expressions for $I^*$ for ROS and the symmetric SBM, and side information channels (iii) carefully resolve all the spelling errors.

Some common and important questions are clarified now, and we will expand on them in the updated version.

----
 **Runtime of the algorithm:** Our algorithm has nearly linear runtime for the respective model. Generally, $\tilde{O}(n^2)$ for ROS, and if the network is sparse as it is the case in log-degree regime SBM, then runtime is $\tilde{O}($ # non-zero entries of $A)$. The main implementation idea is already laid out in [Dhara et al. 2022b, Remark 2.1] and [Gaudio & Joshi 2023]; the idea is to compute top eigenpairs sequentially using the faster eigenvector computation of Garber et al. 2016.

----
**Estimating model parameters:** If the distribution parameters for $P_+$, $P_-$, and $Q$ are unknown, one can perform almost exact recovery to find a preliminary labeling and use the preliminary labeling to estimate the distributional parameters. One can then use these estimates as a proxy in (a) either the spectral algorithm or even (b) in the refinement step of the two-stage algorithm similar to [Saad & Nosratinia 2018] which requires knowing the model parameters. As a result, the practical advantage of the spectral algorithm over two-stage approaches is not clear if the model parameters are not known, however, the spectral algorithm’s analysis has still been of great interest due to its elegance and for theoretical investigations. The plug-in estimates created this way for SBM and ROS parameters suffice for the spectral algorithm to continue to succeed.
We can estimate the parameters for $S_+$ and $S_- $ similarly from preliminary labelings if they are parameterized distributions like BEC, BSC, GF. However, whether the estimates would be good enough for the spectral algorithm to continue to succeed depends on the channel. For our examples:

BEC: the algorithm does not use $\varepsilon$, so no need to estimate.

BSC: it remains unclear whether the estimated $\alpha$ would be good enough.

GF: the estimated means from preliminary labelings suffice for the spectral algorithm to continue to work.

----
**Multi-community extension:** We have some discussion on this in Appendix A.1 lines 758-761. We would further like to expand on this at a technical level. The genie-aided framework generalizes to the $K$-community case where each vertex now has $K$ score values $\{z^*_{i,(k)}: k \in [K], i \in [n] \}$. The score for the $k^\mathrm{th}$ community for the vertex $i$ is $$z^*_{i,(k)}= \log\left(\mathbb{P}( \sigma_{i}^*=k \mid A,y, \sigma_{-i}^*))\right).$$
The optimal genie-based estimator is then simply given by $\hat\sigma_{i,\\, Gen}=\arg \max_{k \in [K]} z^*_{i,(k)}$.
The spectral algorithm now also computes $K$ different score vectors to approximate the statistics of the genie scores and takes the entrywise arg max to come up with the spectral estimator of the labeling.

We emphasize that Abbe et al. 2020 noted some crucial issues in the balanced and symmetric multi-community case. We note that this issue occurs only when eigenvalues of $A^*=\mathbb{E}[A \mid \sigma^*]$ have multiplicities, and thus eigenvectors are only recovered up to rotation. However, these degenerate cases only happen for a measure zero subset of IT-feasible parameters. Otherwise, for most  IT-feasible parameters, the above strategy should be able to approximate the genie scores from eigenvectors. It remains an interesting open question to handle these degenerate cases by a spectral strategy, which may require new ideas.

-----
**Experiments:** Some reviewers have pointed out that having numerical simulations verifying the spectral algorithm’s performance would strengthen the paper. We will include simulations verifying empirically that the algorithms attain the information-theoretic thresholds.

---
References:

[1] Souvik Dhara, Julia Gaudio, Elchanan Mossel, and Colin Sandon. Spectral algorithms optimally recover (censored) planted dense subgraphs. arXiv preprint arXiv:2203.11847, 2022b

[2] Julia Gaudio and Nirmit Joshi. Community detection in the hypergraph SBM: Exact recovery given the similarity matrix. In The Thirty Sixth Annual Conference on Learning Theory, pp. 469–510. PMLR, 2023.

[3] Dan Garber, Elad Hazan, Chi Jin, Cameron Musco, Praneeth Netrapalli, Aaron Sid- ford, et al. Faster eigenvector computation via shift-and-invert preconditioning. In International Conference on Machine Learning, pages 2626–2634. PMLR, 2016.

---

### Meta-Review · Area_Chair_C9tB · 2024-12-14

**Metareview:**

Three of the reviews are positive, commenting favorably on the contributions, writing, and other aspects, with some minor reservations such as missing references and no experiments.  The authors said they will add some experiments, and it's not ideal that those would not be reviewed, but overall I'm OK with taking this as a theory paper.

One reviewer gave a much lower score, but from what I can tell, the comments themselves would be more aligned with at most 'borderline reject' rather than such a strong rejection.  This reviewer did not acknowledge the rebuttal and did not respond even after repeated requests.  From my own viewing of the rebuttal, I think the concerns are addressed adequately.  But I ask the authors to very carefully revise the final version to address the feedback from all 4 reviewers.

**Additional Comments On Reviewer Discussion:**

As noted above, the reviewer with (by far) the lowest score was unresponsive so I had to use my own judgment.

---

### Decision · Program_Chairs · 2025-01-22

Accept (Poster)